# Dynamically Stable Infinite-Width Limits of Neural Classifiers

## Abstract

Recent research has been focused on two different approaches to studying neural networks training in the limit of infinite width (1) a mean-field (MF) and (2) a constant neural tangent kernel (NTK) approximations. These two approaches have different scaling of hyperparameters with the width of a network layer and as a result, different infinite-width limit models. Restricting ourselves to single hidden layer nets with zero-mean initialization trained for binary classification with SGD, we propose a general framework to study how the limit behavior of neural models depends on the scaling of hyperparameters with network width. Our framework allows us to derive scaling for existing MF and NTK limits, as well as an uncountable number of other scalings that lead to a dynamically stable limit behavior of corresponding models. However, only a finite number of distinct limit models are induced by these scalings. Each distinct limit model corresponds to a unique combination of such properties as boundedness of logits and tangent kernels at initialization or stationarity of tangent kernels. Existing MF and NTK limit models, as well as one novel limit model, satisfy most of the properties demonstrated by finite-width models. We also propose a novel initialization-corrected mean-field limit that satisfies all properties noted above, and its corresponding model is a simple modification for a finite-width model.

## 1 Introduction

For a couple of decades neural networks have proved to be useful in a variety of applications. However, their theoretical understanding is still lacking. Several recent works have tried to simplify the object of study by approximating a training dynamics of a finite-width neural network with its limit counterpart in the limit of a large number of hidden units; we refer it as an "infinite-width" limit. The exact type of the limit training dynamics depends on how hyperparameters of the training dynamics scale with width. In particular, two different types of limit models have been already extensively discussed in the literature: an NTK model (Jacot et al., 2018) and a mean-field limit model (Mei et al., 2018; 2019; Rotskoff & Vanden-Eijnden, 2019; Sirignano & Spiliopoulos, 2020; Chizat & Bach, 2018; Yarotsky, 2018). A recent work (Golikov, 2020) attempted to provide a link between these two different types of limit models by building a framework for choosing a scaling of hyperparameters that lead to a "well-defined" limit model. Our work is the next step in this direction. We study infinite-width limits for networks with a single hidden layer trained to minimize cross-entropy loss with gradient descent. Our contributions are following.

1. We develop a framework for reasoning about scaling of hyperparameters, which allows one to infer scaling parameters that allow for a dynamically stable model evolution in the limit of infinite width. This framework allows us to derive both mean-field and NTK limits that have been extensively studied in the literature, as well as the "intermediate limit" introduced in Golikov (2020).

2. Our framework demonstrates that there are only 13 distinct stable model evolution equations in the limit of infinite width that can be induced by scaling hyperparameters of a finite-width model. Each distinct limit model corresponds to a region (two-, one-, or zero-dimensional) of a green band of the Figure 1, left.

3. We consider a list of properties that are statisfied by the evolution of finite-width models, but not generally are for its infinite-width limits. We demonstrate that mean-field and NTK

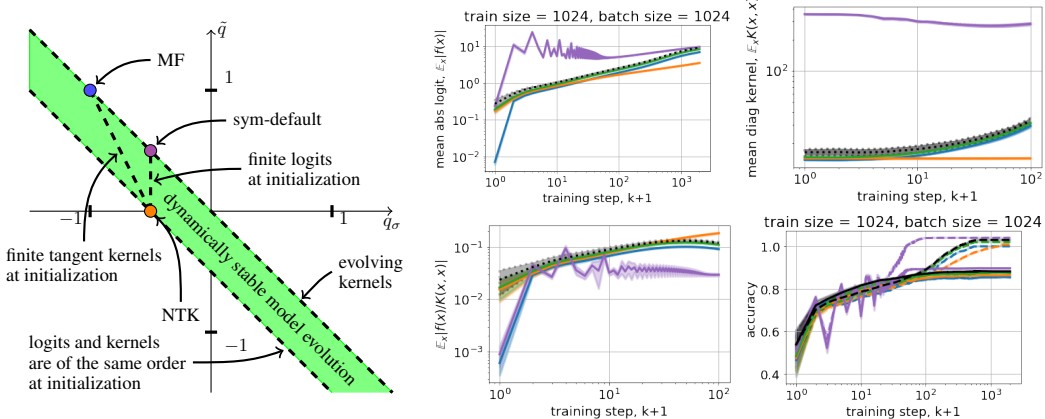

Figure 1: **A diagram on the left specifies several properties demonstrated by finite-width models. As plots on the right demonstrate, our novel IC-MF limit model satisfy all of these properties, while MF and NTK limit models, as well as sym-default limit model presented in the paper violate some of them.** *Left:* A band of scaling exponents $(q_\sigma, \tilde{q})$ that lead to dynamically stable model evolutions in the limit of infinite width, as well as dashed lines of special properties that corresponding limits satisfy. Three colored points correspond to limit models that satisfy most of these properties. *Right:* Training dynamics of three models that correspond to color points on the left plot, as well as of initialization-corrected mean-field model (IC-MF), which does not correspond to any point of the left plot. These models are results of scaling of a reference model of width $d = 2^7$ (black line) up to width $d = 2^{16}$ (colored lines). Solid lines correspond to the test set, while dashed lines are for the train set. Note that we have added a small vertical displacement to all curves in order to make them visually distinguishable. See Appendix F for details.

> limit models, as well as "sym-default" limit model which was not discussed in the literature previously, are special in the sense that they satisfy most of these properties among all limit models induced by hyperparameter scalings. We propose a model modification that allows for all of these properties in the limit of infinite width and call the corresponding limit "initialization-corrected mean-field limit (IC-MF)".

4. We discuss the ability of limit models to approximate the training dynamics of finite-width ones. We show that our proposed IC-MF limiting model is the best among all other possible limit models.

While our present analysis is restricted to networks with a single hidden layer, we discuss a high-level plan for generalizing it to deep nets, as well as an expected outcome of this research program, in App. H.

## 2 TRAINING A ONE HIDDEN LAYER NET WITH SGD

Here we consider training a one hidden layer net $f_d$ with $d$ hidden units with SGD. We assume the hyperparameters, namely, initialization variances and learning rates, are scaled as power-laws of $d$. Each scaling induces a limit model $f_\infty = \lim_{d \to \infty} f_d$. We present a notion of dynamical stability, which states that the change of logits after a single gradient step is comparable to logits themselves. We derive a necessary condition for dynamical stability in terms of the power-law exponents of hyperparameters. We then present a list of conditions that divide the class of scalings into 13 subclasses; each subclass corresponds to a unique distinct limit model.

Consider a one hidden layer network:

$$f(\mathbf{x}; \mathbf{a}, W) = \mathbf{a}^T \phi(W^T \mathbf{x}) = \sum_{r=1}^{d} a_r \phi(\mathbf{w}_r^T \mathbf{x}), \tag{1}$$

where $\mathbf{x} \in \mathbb{R}^{d_\mathbf{x}}$, $W = [\mathbf{w}_1, \ldots, \mathbf{w}_d] \in \mathbb{R}^{d_\mathbf{x} \times d}$, and $\mathbf{a} = [a_1, \ldots, a_d]^T \in \mathbb{R}^d$. We assume a nonlinearity to be real analytic and asymptotically linear: $\phi(z) = \Theta_{z \to \infty}(z)$. Such a nonlinearity

can be, e.g. "leaky softplus": $\phi(z) = \ln(1 + e^z) - \alpha \ln(1 + e^{-z})$ for $\alpha > 0$. This is a technical assumption introduced to simplify proofs. Note that we have used traditional leaky ReLUs in our experiments: see App. F for details. We assume the loss function $\ell(y, z)$ to be the standard binary cross-entropy loss: $\ell(y, z) = \ln(1 + e^{-yz})$, where labels $y \in \{-1, 1\}$. The data distribution loss is defined as $\mathcal{L}(\mathbf{a}, W) = \mathbb{E}_{\mathbf{x}, y \sim \mathcal{D}} \ell(y, f(\mathbf{x}; \mathbf{a}, W))$. We assume that the data distribution $\mathcal{D}$ does not depend on width $d$.

Weights are initialized with isotropic gaussians with zero means: $\mathbf{w}_r^{(0)} \sim \mathcal{N}(0, \sigma_w^2 I)$, $a_r^{(0)} \sim \mathcal{N}(0, \sigma_a^2)$ $\forall r = 1 \dots d$. The evolution of weights is driven by the stochastic gradient descent (SGD):

$$\Delta \theta^{(k)} = \theta^{(k+1)} - \theta^{(k)} = -\eta_\theta \frac{\partial \ell(y_\theta^{(k)}, f(\mathbf{x}_\theta^{(k)}; \mathbf{a}, W))}{\partial \theta}, \quad (\mathbf{x}_\theta^{(k)}, y_\theta^{(k)}) \sim \mathcal{D}, \tag{2}$$

where $\theta$ is either $\mathbf{a}$ or $W$. We assume that gradients for $\mathbf{a}$ and $W$ are estimated using independent data samples $(\mathbf{x}_a^{(k)}, y_a^{(k)})$ and $(\mathbf{x}_w^{(k)}, y_w^{(k)})$. While this assumption is indeed non-standard, we note that corresponding stochastic gradients still give unbiased estimates for true gradients. Moreover, we have used either full-batch GD or standard mini-batch SGD in our experiments: see App. F for details. Define:

$$\hat{a}_r^{(k)} = \frac{a_r^{(k)}}{\sigma_a}, \quad \hat{\mathbf{w}}_r^{(k)} = \frac{\mathbf{w}_r^{(k)}}{\sigma_w}, \quad \hat{\eta}_a = \frac{\eta_a}{\sigma_a^2}, \quad \hat{\eta}_w = \frac{\eta_w}{\sigma_w^2}. \tag{3}$$

Then the dynamics transforms to:

$$\Delta \hat{\theta}_r^{(k)} = \hat{\eta}_\theta \frac{\partial \ell(y_\theta^{(k)}, f(\mathbf{x}_\theta^{(k)}; \sigma_a \hat{\mathbf{a}}, \sigma_w \hat{W}))}{\partial \hat{\theta}_r}, \quad (\mathbf{x}_\theta^{(k)}, y_\theta^{(k)}) \sim \mathcal{D}, \tag{4}$$

while scaled initial conditions become: $\hat{a}_r^{(0)} \sim \mathcal{N}(0, 1)$, $\hat{\mathbf{w}}_r^{(0)} \sim \mathcal{N}(0, I)$ $\forall r = 1 \dots d$.

By expanding gradients, we get the following:

$$\Delta \hat{a}_r^{(k)} = -\hat{\eta}_a \sigma_a \nabla_{f_d}^{(k)} \ell(\mathbf{x}_a^{(k)}, y_a^{(k)}) \, \phi(\sigma_w \hat{\mathbf{w}}_r^{(k),T} \mathbf{x}_a^{(k)}), \quad \hat{a}_r^{(0)} \sim \mathcal{N}(0, 1), \tag{5}$$

$$\Delta \hat{\mathbf{w}}_r^{(k)} = -\hat{\eta}_w \sigma_a \sigma_w \nabla_{f_d}^{(k)} \ell(\mathbf{x}_w^{(k)}, y_w^{(k)}) \, \hat{a}_r^{(k)} \phi'(\dots) \mathbf{x}_w^{(k)}, \quad \hat{\mathbf{w}}_r^{(0)} \sim \mathcal{N}(0, I), \tag{6}$$

$$\nabla_{f_d}^{(k)} \ell(\mathbf{x}, y) = \left. \frac{\partial \ell(y, z)}{\partial z} \right|_{z = f_d^{(k)}(\mathbf{x})} = \frac{-y}{1 + \exp(f_d^{(k)}(\mathbf{x})y)}, \quad f_d^{(k)}(\mathbf{x}) = \sigma_a \sum_{r=1}^d \hat{a}_r^{(k)} \phi(\sigma_w \hat{\mathbf{w}}_r^{(k),T} \mathbf{x}).$$

Without loss of generality assume $\sigma_w = 1$ (we can rescale inputs $\mathbf{x}$ otherwise). We shall omit a subscript of $\sigma_a$ from now on. Assume hyperparameters that drive the dynamics obey power-law dependence on $d$:

$$\sigma(d) = \sigma^* \times (d/d^*)^{q_\sigma}, \quad \hat{\eta}_a(d) = \hat{\eta}_a^* \times (d/d^*)^{\tilde{q}_a}, \quad \hat{\eta}_w(d) = \hat{\eta}_w^* \times (d/d^*)^{\tilde{q}_w}. \tag{7}$$

Given this, a network of width $d^*$ has hyperparameters $\sigma^*$ and $\hat{\eta}_{a \vee w}^*$. Here and then we write "$a \vee w$" meaning "$a$ or $w$".

This assumption is quite natural: for He initialization (He et al., 2015) commonly used in practice $\sigma \propto d^{-1/2}$, while we keep learning rates in the original parameterization constant while changing width by default: $\eta_{a \vee w} = \text{const}$, which implies $\hat{\eta}_a \propto d$ and $\hat{\eta}_w \propto d^0$. On the other hand, NTK scaling (Jacot et al., 2018; Lee et al., 2019) requires scaled learning rates to be constants: $\hat{\eta}_{a \vee w} \propto d^0$.

Scaling exponents $(q_\sigma, \tilde{q}_a, \tilde{q}_w)$ together with proportionality factors $(d^*, \sigma^*, \hat{\eta}_a^*, \hat{\eta}_w^*)$ define a limit model $f_\infty^{(k)}(\mathbf{x}) = \lim_{d \to \infty} f_d^{(k)}(\mathbf{x})$. We call a model "dynamically stable in the limit of large width" if it satisfies the following condition which we state formally in Appendix A:

**Condition 1** (informal version of Condition 4 in Appendix A). *Let* $\Delta f_d^{(k)}(\mathbf{x}) = f_d^{(k+1)}(\mathbf{x}) - f_d^{(k)}(\mathbf{x})$.

$$\exists k_{balance} \in \mathbb{N} : \forall k \geq k_{balance} \quad \frac{\Delta f_d^{(k)}}{f_d^{k_{balance}}} \textit{ stays finite for large } d.$$

Roughly speaking, this condition states that the change of logits after a single step is comparable to logits themselves. This means that the model learns.

Note that this condition is weaker than the one used in Golikov (2020), because it allows logits to vanish or diverge with width. Such situations are fine, because only logit signs matter for the binary classification.

For simplicity assume $\tilde{q}_a = \tilde{q}_w = \tilde{q}$. We prove the following in Appendix B.1:

**Proposition 1.** *Suppose $\tilde{q}_a = \tilde{q}_w = \tilde{q}$ and $\mathcal{D}$ is a continuous distribution. Then Condition 1 requires $q_\sigma + \tilde{q} \in [-1/2, 0]$ to hold.*

This statement gives a necessary condition for growth rates of $\sigma$ and $\hat{\eta}$ to lead to a well-defined limit model evolution. This condition corresponds to a band in $(q_\sigma, \tilde{q})$-plane: see Figure 1, left. We refer it as a "band of dynamical stability".

Each point of this band corresponds to a dynamically stable limit model evolution. We present several conditions that separate the dynamical stability band into regions. We then show that each region corresponds to a single limit model evolution.

We start with defining tangent kernels. Since $\phi$ is smooth, we have:

$$\Delta f_d^{(k)}(\mathbf{x}) = f_d^{(k+1)}(\mathbf{x}) - f_d^{(k)}(\mathbf{x}) = \sum_{r=1}^d \frac{\partial f_d(\mathbf{x})}{\partial \hat{\theta}_r}\bigg|_{\hat{\theta}_r = \hat{\theta}_r^{(k)}} \Delta\hat{\theta}_r^{(k)} + O_{\hat{\eta}_{a\vee w}^* \to 0}(\hat{\eta}_a^*\hat{\eta}_w^* + \hat{\eta}_w^{*,2}) =$$
$$= -\hat{\eta}_a^* \nabla_{f_d}^{(k)} \ell(\mathbf{x}_a^{(k)}, y_a^{(k)}) \, K_{a,d}^{(k)}(\mathbf{x}, \mathbf{x}_a^{(k)}) - \hat{\eta}_w^* \nabla_{f_d}^{(k)} \ell(\mathbf{x}_w^{(k)}, y_w^{(k)}) \, K_{w,d}^{(k)}(\mathbf{x}, \mathbf{x}_w^{(k)}) + O(\hat{\eta}_a^*\hat{\eta}_w^* + \hat{\eta}_w^{*,2}), \tag{8}$$

where we have defined kernels:

$$K_{a,d}^{(k)}(\mathbf{x}, \mathbf{x}') = (d/d^*)^{\tilde{q}_a} \sigma^2 \sum_{r=1}^d \phi(\hat{\mathbf{w}}_r^{(k),T}\mathbf{x})\phi(\hat{\mathbf{w}}_r^{(k),T}\mathbf{x}'), \tag{9}$$

$$K_{w,d}^{(k)}(\mathbf{x}, \mathbf{x}') = (d/d^*)^{\tilde{q}_w} \sigma^2 \sum_{r=1}^d |\hat{a}_r^{(k)}|^2 \phi'(\hat{\mathbf{w}}_r^{(k),T}\mathbf{x})\phi'(\hat{\mathbf{w}}_r^{(k),T}\mathbf{x}')\mathbf{x}^T\mathbf{x}'. \tag{10}$$

Here we deviate from the traditional definition of tangent kernels (e.g. from Jacot et al. (2018)) in embedding learning rate growth factors into kernels. This is done for avoiding $0 \times \infty$ ambiguity when $\hat{\eta}_{a\vee w}$ grows with $d$ while $\sigma$ vanishes so that "a learning rate times a kernel" stays finite. This is the case for the mean-field scaling: $\hat{\eta}_{a\vee w} \propto d$, while $\sigma \propto d^{-1}$.

While for the NTK scaling kernels stop evolving with $k$ in the limit of large $d$, this is not the case generally. Indeed, for the mean-field scaling mentioned above we have:

$$K_{a,d}^{(k)}(\mathbf{x}, \mathbf{x}') = \sigma^{*,2}(d/d^*)^{-1} \sum_{r=1}^d \phi(\hat{\mathbf{w}}_r^{(k),T}\mathbf{x})\phi(\hat{\mathbf{w}}_r^{(k),T}\mathbf{x}'). \tag{11}$$

Similarly to the NTK case, the kernel above converges due to the Law of Large Numbers, however in contrast to the NTK case the weights evolve in the limit: $\hat{\mathbf{w}}_r^{(k)} \not\to \hat{\mathbf{w}}_r^{(0)}$. This is due to the fact that weight increments are proportional to $\hat{\eta}_w \sigma$ which is $\propto d^0$ for the mean-field scaling but $\propto d^{-1/2}$ for the NTK one. For this reason, similarly to model increments $\Delta f_d^{(k)}$ we define kernel increments:

$$\Delta K_{a\vee w,d}^{(k)}(\mathbf{x}, \mathbf{x}') = K_{a\vee w,d}^{(k+1)}(\mathbf{x}, \mathbf{x}') - K_{a\vee w,d}^{(k)}(\mathbf{x}, \mathbf{x}'). \tag{12}$$

**Condition 2** (informal version of Condition 5 in Appendix A). *Following conditions separate the band of dynamical stability (Figure 1, left):*

1. *$f_d^{(0)}$ stays finite for large $d$.*

2. *$K_{a\vee w,d}^{(0)}$ stays finite for large $d$.*

3. *$K_{a\vee w,d}^{(0)}/f_d^{(0)}$ stays finite for large $d$.*

    *4. $\Delta K^{(0)}_{a\vee w,d}/K^{(0)}_{a\vee w,d}$ stays finite for large $d$.*

We prove the following in Appendix B.2:

**Proposition 2** (Separating conditions)**.** *Given Condition 1, Condition 2 reads as, point by point:*

    *1. A limit model at initialization is finite: $q_\sigma + 1/2 = 0$.*

    *2. Tangent kernels at initialization are finite: $2q_\sigma + \tilde{q} + 1 = 0$.*

    *3. Tangent kernels and a limit model are of the same order at initialization: $q_\sigma + \tilde{q} + 1/2 = 0$.*

    *4. Tangent kernels start to evolve: $q_\sigma + \tilde{q} = 0$.*

We have also checked this Proposition numerically for limit models discussed below: see Figure 1, right. Each condition corresponds to a straight line in the $(q_\sigma, \tilde{q})$-plane: see Figure 1, left. These four lines divide the well-definiteness band into 13 regions: three are two-dimensional, seven are one-dimensional, and three are zero-dimensional. In Appendix C we show that each region corresponds to a single distinct limit model evolution; we also list corresponding evolution equations. Note that a segment (a one-dimensional region) that corresponds to the Condition 2-2 exactly coincides with a family of "intermediate scalings" introduced in Golikov (2020).

## 3   Capturing the behavior of finite-width nets

A possible use-case for a limit model is being a proxy for a given finite-width net, useful for theoretical considerations. For example, a number of theoretical properties, including convergence to a global minimum and generalization, are already proven for nets near the NTK limit: see Arora et al. (2019b).

Note that a typical finite-width model satisfies all four statements of Condition 2 (if we exclude the word "limit" from them). Indeed, neural nets are typically initialized with He initialization (He et al., 2015) that guarantees finite $f^{(0)}_d$ even for large width $d$. Since learning rates of finite nets are finite, the tangent kernels are finite as well. Nevertheless, a neural tangent kernel of a typical finite-width network evolves significantly: Arora et al. (2019a) have shown that freezing NTK of practical convolutional nets sufficiently reduces their generalization ability; Woodworth et al. (2019) also noticed that evolution of NTK is sufficient for good performance.

Consequently, if we want a limit model to capture the dynamics of a finite-width net, we have to satisfy all four statements of Condition 2. However, as one can see from Figure 1, we cannot satisfy all of them simultaneously. We say that one limit model captures the behavior of a finite-width one better than the other, if all statements of Conditions 2 satisfied by the latter are satisfied by the former too. If we say in this case that "the former dominates the latter" then one can easily notice that there are only three "non-dominated" limit models which we discuss in the upcoming section. After that, we introduce a model modification that allows for a limit satisfying all four statements.

### 3.1   "Non-dominated" limit models: MF, NTK and "sym-default"

Obviously, the three "non-dominated" limit models are exactly three zero-dimensional regions (points) in Figure 1, left. First suppose statements 1, 2 and 3 hold, hence tangent kernels are constant throughout training (see Figure 1, right). A corresponding point $q_\sigma = -1/2$, $\tilde{q} = 0$ reads as $\sigma \propto d^{-1/2}$ and $\hat{\eta} = \text{const}$, which is the case considered in the seminal paper on NTK (Jacot et al., 2018). The limit dynamics is then given as (see App. C.1.1 and App. C for the general derivation, and eqs. (71-75) for a complete system of evolution equations):

$$f^{(k+1)}_{\text{ntk},\infty}(\mathbf{x}) = f^{(k)}_{\text{ntk},\infty}(\mathbf{x}) - \hat{\eta}^*_a \nabla^{(k)}_{f_{\text{ntk}}} \ell(\mathbf{x}^{(k)}_a, y^{(k)}_a) \, K^{(0)}_{a,\infty}(\mathbf{x}, \mathbf{x}^{(k)}_a) - \hat{\eta}^*_w \nabla^{(k)}_{f_{\text{ntk}}} \ell(\mathbf{x}^{(k)}_w, y^{(k)}_w) \, K^{(0)}_{w,\infty}(\mathbf{x}, \mathbf{x}^{(k)}_w),$$

$$f^{(0)}_{\text{ntk},\infty}(\mathbf{x}) \sim \mathcal{N}(0, \sigma^{*,2}\sigma^{(0),2}(\mathbf{x})), \tag{13}$$

where $(\mathbf{x}^{(k)}_{a\vee w}, y^{(k)}_{a\vee w}) \sim \mathcal{D}$ and limit tangent kernels $K^{(0)}_{a\vee w,\infty}$ and standard deviations at the initialization $\sigma^{(0)}(\mathbf{x})$ can be calculated along the same lines as in Lee et al. (2019).

Next, suppose statements 2 and 4 hold. In this case $K_\infty^{(k)}$ does not coincide with $K_\infty^{(0)}$ (see Figure 1, right), hence the dynamics analogous to (13) is not closed. However, the limit dynamics can be expressed as an evolution of a weight-space measure (see Rotskoff & Vanden-Eijnden (2019); Chizat & Bach (2018) for a similar dynamics for the gradient flow, App. C.2.1 and App. C for the general derivation, and eqs. (94-96) for a complete system of evolution equations):

$$\mu_\infty^{(k+1)} = \mu_\infty^{(k)} + \text{div}(\mu_\infty^{(k)} \Delta\theta_{\text{mf}}^{(k)}), \quad \mu_\infty^{(0)} = \mathcal{N}(0, I_{1+d_\mathbf{x}}), \tag{14}$$

$$f_{\text{mf},\infty}^{(k)}(\mathbf{x}) = \sigma^* \int \hat{a}\phi(\hat{\mathbf{w}}^T\mathbf{x}) \, \mu_\infty^{(k)}(d\hat{a}, d\hat{\mathbf{w}}), \tag{15}$$

where the vector field $\Delta\theta_{\text{mf}}^{(k)}$ is defined as follows:

$$\Delta\theta_{\text{mf}}^{(k)}(\hat{a}, \hat{\mathbf{w}}) = -[\nabla_{f_{\text{mf}}}^{(k)}\ell(\mathbf{x}_a^{(k)}, y_a^{(k)})\phi(\hat{\mathbf{w}}^T\mathbf{x}_a^{(k)}), \nabla_{f_{\text{mf}}}^{(k)}\ell(\mathbf{x}_w^{(k)}, y_w^{(k)})\hat{a}\phi'(\hat{\mathbf{w}}^T\mathbf{x}_w^{(k)})\mathbf{x}_w^{(k),T}]^T, \tag{16}$$

where we write "$[\mathbf{u}, \mathbf{v}]$" meaning a concatenation of two row vectors $\mathbf{u}$ and $\mathbf{v}$. Here we have $q_\sigma = -1$, $\tilde{q} = 1$, hence $\sigma \propto d^{-1}$ and $\hat{\eta} \propto d$; this hyperparameter scaling were used in Rotskoff & Vanden-Eijnden (2019); Chizat & Bach (2018). Note that since a measure at the initialization $\mu_\infty^{(0)}$ has a zero mean, a limit model vanishes at the initialization $f_{\text{mf},\infty}^{(0)} = 0$ (see Figure 1, right) thus violating statements 1 and 3 of Condition 2.

Finally, consider a point for which statements 1 and 4 hold: $q_\sigma = -1/2$, $\tilde{q} = 1/2$. This situation is very similar to what we call "default" scaling. Consider He initialization (He et al., 2015), typically used in practice: $\sigma_a \propto d^{-1/2}$ and $\sigma_w \propto d_\mathbf{x}^{-1/2}$. Assume learning rates (in original parameterization) are not modified with width: $\eta_a = \text{const}$ and $\eta_w = \text{const}$. This implies $\hat{\eta}_a \propto d$ and $\hat{\eta}_w \propto 1$, or $\tilde{q}_a = 1$ and $\tilde{q}_w = 0$. We refer the scaling $q_\sigma = -1/2$, $\tilde{q}_a = 1$ and $\tilde{q}_w = 0$ as "default", and the scaling $q_\sigma = -1/2$, $\tilde{q} = 1/2$ as "sym-default". A limit model evolution for the sym-default scaling looks as follows (see App. C.2.2 for an equivalent formulation and App. C for the general derivation, and eqs. (97-101) for a complete system of evolution equations):

$$\mu_\infty^{(k+1)} = \mu_\infty^{(k)} + \text{div}(\mu_\infty^{(k)} \Delta\theta_{\text{sym-def}}^{(k)}), \quad \mu_\infty^{(0)} = \mathcal{N}(0, I_{1+d_\mathbf{x}}), \tag{17}$$

$$f_{\text{sym-def},\infty}^{(0)}(\mathbf{x}) \sim \mathcal{N}(0, \sigma^{*,2}\sigma^{(0),2}(\mathbf{x})), \quad z_{\text{sym-def},\infty}^{(k)}(\mathbf{x}) = \left[\int \hat{a}\phi(\hat{\mathbf{w}}^T\mathbf{x}) \, \mu_\infty^{(k)}(d\hat{a}, d\hat{\mathbf{w}}) > 0\right], \tag{18}$$

where the vector field $\Delta\theta_{\text{sym-def}}^{(k)}$ is defined similarly to the MF case (16):

$$\Delta\theta_{\text{sym-def}}^{(k)}(\hat{a}, \hat{\mathbf{w}}) = -[\nabla_{f_{\text{sym-def}}}^{(k)}\ell(\mathbf{x}_a^{(k)}, y_a^{(k)})\phi(\hat{\mathbf{w}}^T\mathbf{x}_a^{(k)}), \nabla_{f_{\text{sym-def}}}^{(k)}\ell(\mathbf{x}_w^{(k)}, y_w^{(k)})\hat{a}\phi'(\hat{\mathbf{w}}^T\mathbf{x}_w^{(k)})\mathbf{x}_w^{(k),T}]^T,$$

$$\nabla_{f_{\text{sym-def}}}^{(k)}\ell(\mathbf{x}, y) = -y[yz_{\text{sym-def},\infty}^{(k)}(\mathbf{x}) < 0] \quad \text{for } k \geq 1. \tag{19}$$

As we show in Appendix D, the default scaling leads to an almost similar limit dynamics as the sym-default scaling: eqs. (114-117) for a complete system of the corresponding evolution equations. The quantity $z_{\text{sym-def},\infty}^{(k)}$ should be perceived as a sign of $f_{\text{sym-def},\infty}^{(k)} = \sigma^* \lim_{d\to\infty}\left(d^{q_\sigma+1}\int \hat{a}\phi(\hat{\mathbf{w}}^T\mathbf{x}) \, \mu_d^{(k)}(d\hat{a}, d\hat{\mathbf{w}})\right)$. The reason why we have to switch from logits to their signs is that the limit model diverges for $k \geq 1$: $\lim_{d\to\infty} f_d^{(k)}(\mathbf{x}) = \infty$. Nevertheless the gradient of the cross-entropy loss is well-defined even for infinite logits: it just degenerates into the gradient of a hinge-type loss: $\lim_{f\to+\infty\times z}\frac{\partial\ell(y,f)}{\partial f} = -y[yz < 0]$. For this reason, we redefine the loss gradient for $k \geq 1$ in terms of logit signs: eq. (19). Note that besides of the fact that logits diverge in the limit of large width, the measure in the parameter space $\mu_d^{(k)}$ stays well-defined.

## 3.2 INITIALIZATION-CORRECTED MEAN-FIELD (IC-MF) LIMIT

Here we propose a dynamics that satisfy all four statements of Condition 2. We then show how to modify the network training for the finite width in order to ensure that in the limit of the infinite width its training dynamics converge to the proposed limit one. Consider the following:

$$\mu_\infty^{(k+1)} = \mu_\infty^{(k)} + \text{div}(\mu_\infty^{(k)} \Delta\theta_{\text{icmf}}^{(k)}), \quad \mu_\infty^{(0)} = \mathcal{N}(0, I_{1+d_\mathbf{x}}), \tag{20}$$

$$f_{\text{icmf},\infty}^{(k)}(\mathbf{x}) = \sigma^* \int \hat{a}\phi(\hat{\mathbf{w}}^T\mathbf{x})\,\mu_\infty^{(k)}(d\hat{a}, d\hat{\mathbf{w}}) + f_{\text{ntk},\infty}^{(0)}(\mathbf{x}), \tag{21}$$

where $f_{\text{ntk},\infty}^{(0)}$ is defined similarly to above:

$$f_{\text{ntk},\infty}^{(0)}(\mathbf{x}) \sim \mathcal{N}(0, \sigma^{*,2}\sigma^{(0),2}(\mathbf{x})), \tag{22}$$

the vector field $\Delta\theta_{\text{icmf}}^{(k)}$ is defined analogously to the mean-field case:

$$\Delta\theta_{\text{icmf}}^{(k)}(\hat{a}, \hat{\mathbf{w}}) = -[\nabla_{f_{\text{icmf}}}^{(k)}\ell(\mathbf{x}_a^{(k)}, y_a^{(k)})\phi(\hat{\mathbf{w}}^T\mathbf{x}_a^{(k)}), \nabla_{f_{\text{icmf}}}^{(k)}\ell(\mathbf{x}_w^{(k)}, y_w^{(k)})\hat{a}\phi'(\hat{\mathbf{w}}^T\mathbf{x}_w^{(k)})\mathbf{x}_w^{(k),T}]^T, \tag{23}$$

See App. E and eqs. (131-134) for a complete system of evolution equations. The only difference between this dynamics and the mean-field dynamics is a bias term $f_{\text{ntk},\infty}^{(0)}$ in the definition of logits. This bias term does not depend on $k$ and stays finite for large $d$ in contrast to $f_{\text{mf},\infty}^{(0)}$ which vanishes for large $d$; it ensures Condition 2-1 to hold. As for Condition 2-4, tangent kernels evolve with $k$ simply because the measure $\mu_\infty^{(k)}$ evolves with $k$ similarly to the mean-field case (see Figure 1, right). Indeed,

$$K_{w,\infty}^{(k)}(\mathbf{x}', \mathbf{x}) = \sigma^{*,2}d^* \int |\hat{a}^{(k)}|^2 \phi'(\hat{\mathbf{w}}^{(k),T}\mathbf{x})\phi'(\hat{\mathbf{w}}^{(k),T}\mathbf{x}')\,\mu_\infty^{(k)}(d\hat{a}, d\hat{\mathbf{w}}), \tag{24}$$

and the limit of $K_{a,d}^{(k)}$ is written in a similar way. Kernels at initialization $K_{a\vee w,\infty}^{(0)}$ are finite due to the Law of Large Numbers (Condition 2-2); this, and the finiteness of $f_{\text{ntk}}^{(0)}$ ensures Condition 2-3.

As we show in Appendix E the dynamics (20) is a limit for the GD dynamics of the following model with learning rates $\hat{\eta}_{a\vee w} = \hat{\eta}_{a\vee w}^*(d/d^*)^1$:

$$f_{\text{icmf},d}(\mathbf{x}; \hat{\mathbf{a}}, \hat{W}) = \sigma^*(d/d^*)^{-1}\sum_{r=1}^d \hat{a}_r\phi(\hat{\mathbf{w}}_r^T\mathbf{x}) + \sigma^*((d/d^*)^{-1/2} - (d/d^*)^{-1})\sum_{r=1}^d \hat{a}_r^{(0)}\phi(\hat{\mathbf{w}}_r^{(0),T}\mathbf{x}). \tag{25}$$

Note that $f_{\text{icmf},d^*}(\mathbf{x}) = \sigma^* \sum_{r=1}^{d^*} \hat{a}_r\phi(\hat{\mathbf{w}}_r^T\mathbf{x})$: we have not altered the model definition at $d = d^*$.

## 3.3 EXPERIMENTS

Consider a network of width $d^*$ initialized with a standard deviation $\sigma^*$ and trained with learning rates $\hat{\eta}_{a\vee w}^*$. We call this model a "reference". Consider a family of models indexed by a width $d$ with hyperparameters specified by the power-law scaling (7). We train a reference network of width $d^* = 128$ for the binary classification with a cross-entropy loss on the CIFAR2 dataset (a subset of first two classes of CIFAR10). We track the divergence of a limit network from the reference one using the following quantity: $\mathbb{E}_{\mathbf{x}\sim\mathcal{D}_{test}}D_{logits}(f_\infty^{(k)}(\mathbf{x}) \,\|\, f_{d^*}^{(k)}(\mathbf{x}))$, where

$$D_{logits}(\xi \,\|\, \xi^*) = \text{KL}(\mathcal{N}(\mathbb{E}\,\xi, \mathbb{V}\text{ar}\,\xi) \,\|\, \mathcal{N}(\mathbb{E}\,\xi^*, \mathbb{V}\text{ar}\,\xi^*)). \tag{26}$$

We have also tried other divergence measures; see Appendix I.

Results are shown in Figure 2. The NTK limit tracks the reference network well only for the first 20 training steps; a similar observation has been already made by Lee et al. (2019). At the same time, the mean-field limit starts with a high divergence (since the initial limit model is zero in this case), however, after the 80-th step, it becomes smaller than that of the NTK limit. This can be the implication of non-stationary kernels. As for the default case, divergence of logits results in a blow-up of the KL-divergence.

The best overall case is the proposed IC-MF limit, which retains the small KL-divergence related to the reference model throughout the training process. Capturing the behavior of finite-width nets is also possible by introducing finite-width corrections for the NTK (Dyer & Gur-Ari, 2019; Huang & Yau, 2019). However, this gives us an infinite sequence of equations, which is intractable. We have to cut this sequence; this gives us an approximate dynamics, which is still complicated. In contrast, our IC-MF limit is a simple modification of the MF limit, and at the same time, a good proxy for finite-width networks.

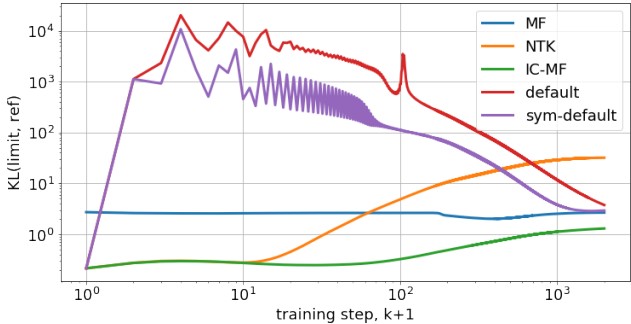

Figure 2: **Initialization-corrected mean-field (IC-MF) limit captures the behavior of a given finite-width network best among other limit models.** We plot a KL-divergence of logits of different infinite-width limits of a fixed finite-width reference model relative to logits of this reference model. *Setup:* we train a one hidden layer network with SGD on CIFAR2 dataset; see Appendix F for details. KL-divergences are estimated using gaussian fits with 10 samples.

## 4 RELATED WORK

A pioneering work of Jacot et al. (2018) have shown that a gradient descent training of a neural net can be viewed as a kernel gradient descent in the space of predictors. The corresponding kernel is called a neural tangent kernel (NTK). Generally, NTK is random and non-stationary, however Jacot et al. (2018) have shown that in the limit of infinite width it becomes constant given a network is parameterized appropriately. In this case the evolution of the model is determined by this constant kernel; see eq. (13). The training regime when NTK is hardly varying is coined as "lazy training", as opposed to the "rich" training regime, when NTK evolves significantly (Woodworth et al., 2019). Chizat et al. (2019) noted that the training becomes lazy for a finite width if one scales the output of the network appropriately. While being theoretically appealing, "laziness" assumption turns out to have a number of limitations in explaining the success of deep learning (Arora et al., 2019a; Ghorbani et al., 2019).

Another line of works considers the evolution of weights as an evolution of a weight-space measure, similar to eq. (14) (Mei et al., 2018; 2019; Sirignano & Spiliopoulos, 2020; Chizat & Bach, 2018; Rotskoff & Vanden-Eijnden, 2019; Yarotsky, 2018). This weight-space measure becomes deterministic in the limit of infinite width, given the network is parameterized appropriately; the corresponding limit dynamics is called "mean-field". Note that the parameterization required here for the convergence to a limit dynamics differs from the one used in the NTK literature.

Our framework for reasoning about scaling of hyperparameters is similar in spirit to the one used in Golikov (2020). However, there are several crucial differences. First, we do not consider weight increments, as well as a model decomposition, and do not try to estimate exponents of the former and for terms of the latter, which arguebly complicates the work of Golikov (2020). Instead, we present derivations in terms of the limit behavior of logits and kernels which appears to be simpler and clearer. Second, our criterion of "dynamical stability" of scaling is weaker compared to the one of Golikov (2020) and more suitable for classification problems, since it allows for diverging or vanishing logits, as long as they give meaningful classification responses. In particular, our dynamical stability condition covers practically important "default" limit for which learning rates are kept constant while width grow up to infinity. Note that "intermediate limits" investigated in Golikov (2020) exactly correspond to limit models which satisfy Condition 2-2. Moreover, both "sym-default" and IC-MF limit models we propose in the present work have not been discussed previously; we present limit evolution equations for both of them (see Appendix C). Finally, our analysis suggests that there are only 13 distinct limit models that can be induced by power-law scaling of hyperparameters.

## 5 CONCLUSIONS

The current work follows a direction started in Golikov (2020): we study how one should scale hyperparameters of a neural network with a single hidden layer in order to converge to a "dynamically

stable" limit training dynamics. A weaker dynamical stability condition leads us to a richer class of possible limit models as compared to Golikov (2020). In particular, the class of limit models we consider includes a "default" limit model that corresponds to a network with infinitely large number of nodes and finite learning rates in the original parameterization. This "default" limit model does not satisfy a "well-definiteness" condition of Golikov (2020).

Moreover, we show that the class of limit models that can be achieved by scaling hyperparameters of finite-width nets is finite. The space of hyperparameter scalings is divided by regions with certain conditions on the training dynamics, and each region corresponds to a single limit model. All of these conditions are satisfied by finite-width networks, but cannot be satisfied by limit models all simultaneously. We propose a modification of a finite-width model; the limit of this modification corresponds to a limit model that satisfy all of the conditions mentioned above and tracks the dynamics of a "reference" finite-width net better than other limit models.

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

## A  FORMAL CONDITIONS FOR SECTION 2

Here we present formal definitions for notions that appear in Section 2; they are required for mathematical rigor. First, recall the definition of tangent kernels:

$$K_{a,d}^{(k)}(\mathbf{x}, \mathbf{x}') = (d/d^*)^{\tilde{q}_a} \sigma^2 \sum_{r=1}^{d} \phi(\hat{\mathbf{w}}_r^{(k),T}\mathbf{x})\phi(\hat{\mathbf{w}}_r^{(k),T}\mathbf{x}'), \tag{27}$$

$$K_{w,d}^{(k)}(\mathbf{x}, \mathbf{x}') = (d/d^*)^{\tilde{q}_w} \sigma^2 \sum_{r=1}^{d} |\hat{a}_r^{(k)}|^2 \phi'(\hat{\mathbf{w}}_r^{(k),T}\mathbf{x})\phi'(\hat{\mathbf{w}}_r^{(k),T}\mathbf{x}')\mathbf{x}^T\mathbf{x}'. \tag{28}$$

The kernels are used to express a model increment:

$$\Delta f_d^{(k)}(\mathbf{x}) = f_d^{(k+1)}(\mathbf{x}) - f_d^{(k)}(\mathbf{x}) = \sum_{r=1}^{d} \left.\frac{\partial f_d(\mathbf{x})}{\partial \hat{\theta}_r}\right|_{\hat{\theta}_r = \hat{\theta}_r^{(k)}} \Delta\hat{\theta}_r^{(k)} + O_{\hat{\eta}_{a\vee w}^* \to 0}(\hat{\eta}_a^*\hat{\eta}_w^* + \hat{\eta}_w^{*,2}) =$$

$$= -\hat{\eta}_a^*\nabla_{f_d}^{(k)}\ell(\mathbf{x}_a^{(k)}, y_a^{(k)})\, K_{a,d}^{(k)}(\mathbf{x}, \mathbf{x}_a^{(k)}) - \hat{\eta}_w^*\nabla_{f_d}^{(k)}\ell(\mathbf{x}_w^{(k)}, y_w^{(k)})\, K_{w,d}^{(k)}(\mathbf{x}, \mathbf{x}_w^{(k)}) + O(\hat{\eta}_a^*\hat{\eta}_w^* + \hat{\eta}_w^{*,2}), \tag{29}$$

Define the linear part of the model increment with respect to learning rate proportionality factors:

$$\Delta f_{d,a\vee w}^{(k),\prime}(\mathbf{x}) = \left.\frac{\partial \Delta f_d^{(k)}(\mathbf{x})}{\partial \hat{\eta}_{a\vee w}^*}\right|_{\substack{\hat{\eta}_a^*=0 \\ \hat{\eta}_w^*=0}} = -\nabla_{f_d}^{(k)}\ell(\mathbf{x}_{a\vee w}^{(k)}, y_{a\vee w}^{(k)})\, K_{a\vee w,d}^{(k)}(\mathbf{x}, \mathbf{x}_{a\vee w}^{(k)}). \tag{30}$$

We use this quantity to rewrite the model increment:

$$\Delta f_d^{(k)}(\mathbf{x}) = \hat{\eta}_a^*\Delta f_{d,a}^{(k),\prime}(\mathbf{x}) + \hat{\eta}_w^*\Delta f_{d,w}^{(k),\prime}(\mathbf{x}) + O(\hat{\eta}_a^*\hat{\eta}_w^* + \hat{\eta}_w^{*,2}). \tag{31}$$

Let us consider kernel definitions (27) and (28) again. Their increments are given by:

$$\Delta K_{a,d}^{(k)}(\mathbf{x}, \mathbf{x}') = -\hat{\eta}_w^*(d/d^*)^{2\tilde{q}}\sigma^3 \sum_{r=1}^{d} \left(\phi(\hat{\mathbf{w}}_r^{(k),T}\mathbf{x})\phi'(\hat{\mathbf{w}}_r^{(k),T}\mathbf{x}') + \phi'(\hat{\mathbf{w}}_r^{(k),T}\mathbf{x})\phi(\hat{\mathbf{w}}_r^{(k),T}\mathbf{x}')\right) \times$$

$$\times \nabla_{f_d}^{(k)}\ell(\mathbf{x}_w^{(k)}, y_w^{(k)})\hat{a}_r^{(k)}\phi'(\hat{\mathbf{w}}_r^{(k),T}\mathbf{x}_w^{(k)})(\mathbf{x} + \mathbf{x}')^T\mathbf{x}_w^{(k)} + O_{\substack{\hat{\eta}_w^* \to 0 \\ d\to\infty}}(\hat{\eta}_w^{*,2}d^{3\tilde{q}+4q_\sigma+1}), \tag{32}$$

$$\Delta K_{w,d}^{(k)}(\mathbf{x}, \mathbf{x}') = -\hat{\eta}_w^*(d/d^*)^{2\tilde{q}}\sigma^3 \sum_{r=1}^{d} |\hat{a}_r^{(k)}|^2 \Big(\phi'(\hat{\mathbf{w}}_r^{(k),T}\mathbf{x})\phi''(\hat{\mathbf{w}}_r^{(k),T}\mathbf{x}')+$$

$$+ \phi''(\hat{\mathbf{w}}_r^{(k),T}\mathbf{x})\phi'(\hat{\mathbf{w}}_r^{(k),T}\mathbf{x}')\Big)\mathbf{x}^T\mathbf{x}'\times$$

$$\times \nabla_{f_d}^{(k)}\ell(\mathbf{x}_w^{(k)}, y_w^{(k)})\hat{a}_r^{(k)}\phi'(\hat{\mathbf{w}}_r^{(k),T}\mathbf{x}_w^{(k)})(\mathbf{x}+\mathbf{x}')^T\mathbf{x}_w^{(k)} + O_{\substack{\hat{\eta}_w^*\to 0\\ d\to\infty}}(\hat{\eta}_w^{*,2}d^{3\tilde{q}+4q_\sigma+1})-$$

$$- \hat{\eta}_a^*(d/d^*)^{2\tilde{q}}\sigma^3 \sum_{r=1}^{d} 2\hat{a}_r^{(k)}\phi'(\hat{\mathbf{w}}_r^{(k),T}\mathbf{x})\phi'(\hat{\mathbf{w}}_r^{(k),T}\mathbf{x}')\times$$

$$\times \nabla_f^{(k)}\ell(\mathbf{x}_a^{(k)}, y_a^{(k)})\phi(\hat{\mathbf{w}}_r^{(k),T}\mathbf{x}_a^{(k)}) + O_{\substack{\hat{\eta}_a^*\to 0\\ d\to\infty}}(\hat{\eta}_a^{*,2}d^{3\tilde{q}+4q_\sigma+1}). \quad (33)$$

Similarly to what was done for model increments, we define linear parts of the kernel increments with respect to learning rate proportionality factors:

$$\Delta K_{aw,d}^{(k),'}(\mathbf{x}, \mathbf{x}') = \left.\frac{\partial \Delta K_{a,d}^{(k)}(\mathbf{x}, \mathbf{x}')}{\partial \hat{\eta}_w^*}\right|_{\hat{\eta}_w^*=0} =$$

$$= -(d/d^*)^{2\tilde{q}}\sigma^3 \sum_{r=1}^{d} \Big(\phi(\hat{\mathbf{w}}_r^{(k),T}\mathbf{x})\phi'(\hat{\mathbf{w}}_r^{(k),T}\mathbf{x}') + \phi'(\hat{\mathbf{w}}_r^{(k),T}\mathbf{x})\phi(\hat{\mathbf{w}}_r^{(k),T}\mathbf{x}')\Big) \times$$

$$\times \nabla_{f_d}^{(k)}\ell(\mathbf{x}_w^{(k)}, y_w^{(k)})\hat{a}_r^{(k)}\phi'(\hat{\mathbf{w}}_r^{(k),T}\mathbf{x}_w^{(k)})(\mathbf{x}+\mathbf{x}')^T\mathbf{x}_w^{(k)}, \quad (34)$$

$$\Delta K_{ww,d}^{(k),'}(\mathbf{x}, \mathbf{x}') = \left.\frac{\partial \Delta K_{w,d}^{(k)}(\mathbf{x}, \mathbf{x}')}{\partial \hat{\eta}_w^*}\right|_{\hat{\eta}_w^*=0} =$$

$$= -(d/d^*)^{2\tilde{q}}\sigma^3 \sum_{r=1}^{d} |\hat{a}_r^{(k)}|^2 \Big(\phi'(\hat{\mathbf{w}}_r^{(k),T}\mathbf{x})\phi''(\hat{\mathbf{w}}_r^{(k),T}\mathbf{x}') + \phi''(\hat{\mathbf{w}}_r^{(k),T}\mathbf{x})\phi'(\hat{\mathbf{w}}_r^{(k),T}\mathbf{x}')\Big)\mathbf{x}^T\mathbf{x}'\times$$

$$\times \nabla_{f_d}^{(k)}\ell(\mathbf{x}_w^{(k)}, y_w^{(k)})\hat{a}_r^{(k)}\phi'(\hat{\mathbf{w}}_r^{(k),T}\mathbf{x}_w^{(k)})(\mathbf{x}+\mathbf{x}')^T\mathbf{x}_w^{(k)}, \quad (35)$$

$$\Delta K_{wa,d}^{(k),'}(\mathbf{x}, \mathbf{x}') = \left.\frac{\partial \Delta K_{w,d}^{(k)}(\mathbf{x}, \mathbf{x}')}{\partial \hat{\eta}_a^*}\right|_{\hat{\eta}_a^*=0} =$$

$$= -(d/d^*)^{2\tilde{q}}\sigma^3 \sum_{r=1}^{d} 2\hat{a}_r^{(k)}\phi'(\hat{\mathbf{w}}_r^{(k),T}\mathbf{x})\phi'(\hat{\mathbf{w}}_r^{(k),T}\mathbf{x}')\nabla_{f_d}^{(k)}\ell(\mathbf{x}_a^{(k)}, y_a^{(k)})\phi(\hat{\mathbf{w}}_r^{(k),T}\mathbf{x}_a^{(k)}). \quad (36)$$

Note that $\Delta K_{aa,d}^{(k),'}(\mathbf{x}, \mathbf{x}') = 0$ since $\hat{a}_r$-terms are absent in the definition of $K_{a,d}$, eq. (27).

Define $p_{err,d}^{(k)} = \mathcal{P}_{(y,\mathbf{x},y_a^{(:k-1)},\mathbf{x}_a^{(:k-1)},y_w^{(:k-1)},\mathbf{x}_w^{(:k-1)})\sim\mathcal{D}^{2k-1}}\{yf_d^{(k)}(\mathbf{x}) < 0\}$ — the probability of giving a wrong answer on the step $k$. Let $k_{term,d} \in \mathbb{N} \cup \{+\infty\}$ be a maximal $k$ such that $\forall k' < k$ $p_{err,d}^{(k')} > 0$. Generally, $k_{term,d}$ depends on hyperparameters, as well as on the data distribution $\mathcal{D}$.

Scaling exponents $(q_\sigma, \tilde{q}_a, \tilde{q}_w)$ together with proportionality factors $(d^*, \sigma^*, \hat{\eta}_a^*, \hat{\eta}_w^*)$ define a limit model $f_\infty^{(k)}(\mathbf{x}) = \lim_{d\to\infty} f_d^{(k)}(\mathbf{x})$. We call a model "dynamically stable in the limit of large width" if it satisfies the following condition:

**Condition 3.** $\exists k_{balance} \in \mathbb{N} : \forall k \in [k_{balance}, k_{term,\infty}) \cap \mathbb{N}$ $y_a^{(k)}f_\infty^{(k)}(\mathbf{x}_a^{(k)}) < 0$ and $y_w^{(k)}f_\infty^{(k)}(\mathbf{x}_w^{(k)}) < 0$ imply $\Delta f_{d,a\vee w}^{(k),'}(\mathbf{x}) = \Theta_{d\to\infty}(f_d^{(k_{balance})}(\mathbf{x}))$ $\mathbf{x}$-a.e. $(y_{a\vee w}^{(:k)}, \mathbf{x}_{a\vee w}^{(:k)})$-a.s.

This condition puts a constraint on exponents $(q_\sigma, \tilde{q}_a, \tilde{q}_w)$; this constraint generally depends on the train data distribution $\mathcal{D}$ and on proportionality factors $d^*$, $\sigma^*$, and $\hat{\eta}_{a\vee w}^*$. In order to obtain a data-independent hyperparameter-independent constraint, we need the condition above to hold for any value of $k_{term,\infty}$ and any values of $d^*$, $\sigma^*$, and $\hat{\eta}_{a\vee w}^*$. Without loss of generality we can assume $k_{term,\infty}$ to be infinite, which gives the following condition:

**Condition 4** (a formal version of Condition 1). *Given $k_{term,\infty} = +\infty$, $\exists k_{balance} \in \mathbb{N} : \forall \sigma^* > 0$ $\forall \hat{\eta}^*_{a \vee w} > 0 \; \forall k \geq k_{balance} \; y_a^{(k)} f_\infty^{(k)}(\mathbf{x}_a^{(k)}) < 0$ and $y_w^{(k)} f_\infty^{(k)}(\mathbf{x}_w^{(k)}) < 0$ imply $\Delta f_{d,a \vee w}^{(k),'}(\mathbf{x}) = \Theta_{d \to \infty}(f_d^{(k_{balance})}(\mathbf{x}))$ $\mathbf{x}$-a.e. $(y_{a \vee w}^{(:k)}, \mathbf{x}_{a \vee w}^{(:k)})$-a.s.*

**Condition 5** (a formal version of Condition 2). *Following conditions separate the band of dynamical stability (Figure 1, left):*

1. *A limit model at initialization is finite: $f_d^{(0)}(\mathbf{x}) = \Theta_{d \to \infty}(1)$ $\mathbf{x}$-a.e.*

2. *Tangent kernels at initialization are finite: $K_{d,a \vee w}^{(0)}(\mathbf{x}, \mathbf{x}') = \Theta_{d \to \infty}(1)$ $(\mathbf{x}, \mathbf{x}')$-a.e.*

3. *Tangent kernels and a limit model are of the same order at initialization: $K_{d,a \vee w}^{(0)}(\mathbf{x}, \mathbf{x}') = \Theta_{d \to \infty}(f_d^{(0)}(\mathbf{x}))$ $(\mathbf{x}, \mathbf{x}')$-a.e.*

4. *Tangent kernels start to evolve: $\Delta K_{d,wa \vee w}^{(0),'}(\mathbf{x}, \mathbf{x}') = \Theta_{d \to \infty}(K_{d,w}^{(0)}(\mathbf{x}, \mathbf{x}'))$ $(\mathbf{x}, \mathbf{x}')$-a.e. and $\Delta K_{d,aw}^{(0),'}(\mathbf{x}, \mathbf{x}') = \Theta_{d \to \infty}(K_{d,a}^{(0)}(\mathbf{x}, \mathbf{x}'))$ $(\mathbf{x}, \mathbf{x}')$-a.e.*

# B    PROOFS OF PROPOSITIONS

We restate all necessary definitions here. We assume the non-linearity $\phi$ to be real analytic and asymptotically linear: $\phi(z) = \Theta_{z \to \infty}(z)$. We assume the loss function $\ell(y, z)$ to be the standard binary cross-entropy loss: $\ell(y, z) = \ln(1 + e^{-yz})$, where labels $y \in \{-1, 1\}$.

The training dynamics is given as:

$$\Delta \hat{a}_r^{(k)} = -\hat{\eta}_a \sigma \nabla_{f_d}^{(k)} \ell(\mathbf{x}_a^{(k)}, y_a^{(k)}) \; \phi(\hat{\mathbf{w}}_r^{(k),T} \mathbf{x}_a^{(k)}), \quad \hat{a}_r^{(0)} \sim \mathcal{N}(0, 1), \tag{37}$$

$$\Delta \hat{\mathbf{w}}_r^{(k)} = -\hat{\eta}_w \sigma \nabla_{f_d}^{(k)} \ell(\mathbf{x}_w^{(k)}, y_w^{(k)}) \; \hat{a}_r^{(k)} \phi'(\hat{\mathbf{w}}_r^{(k),T} \mathbf{x}_w^{(k)}) \mathbf{x}_w^{(k)}, \quad \hat{\mathbf{w}}_r^{(0)} \sim \mathcal{N}(0, I) \quad \forall r \in [d], \tag{38}$$

$$\nabla_{f_d}^{(k)} \ell(\mathbf{x}, y) = \left. \frac{\partial \ell(y, z)}{\partial z} \right|_{z = f_d^{(k)}(\mathbf{x})} = \frac{-y}{1 + \exp(f_d^{(k)}(\mathbf{x})y)}, \quad f_d^{(k)}(\mathbf{x}) = \sigma \sum_{r=1}^d \hat{a}_r^{(k)} \phi(\hat{\mathbf{w}}_r^{(k),T} \mathbf{x}),$$

where $(\mathbf{x}_{a \vee w}^{(k)}, y_{a \vee w}^{(k)}) \sim \mathcal{D}$ for $\mathcal{D}$ being the data distribution.

We assume hyperparameters to be scaled with width as power-laws:

$$\sigma(d) = \sigma^* \times (d/d^*)^{q_\sigma}, \quad \hat{\eta}_a(d) = \hat{\eta}_a^* \times (d/d^*)^{\tilde{q}_a}, \quad \hat{\eta}_w(d) = \hat{\eta}_w^* \times (d/d^*)^{\tilde{q}_w}.$$

## B.1    PROOF OF PROPOSITION 1

Define:

$$q_\theta^{(k)} = \inf\{q : \; \theta^{(k)} = O_{d \to \infty}(d^q)\}, \quad q_{\Delta \theta}^{(k)} = \inf\{q : \; \Delta \theta^{(k)} = O_{d \to \infty}(d^q)\}, \tag{39}$$

where $\theta$ should be substituted with $a$ or $\mathbf{w}$. We define $\inf(\emptyset) = +\infty$. We introduce similar definitions for other quantities:

$$q_f^{(k)}(\mathbf{x}) = \inf\{q : \; f_d^{(k)}(\mathbf{x}) = O_{d \to \infty}(d^q)\}, \quad q_{\nabla \ell}^{(k)}(\mathbf{x}, y) = \inf\{q : \; \nabla_{f_d}^{(k)} \ell(\mathbf{x}, y) = O_{d \to \infty}(d^q)\}, \tag{40}$$

$$q_{\Delta f}^{(k)}(\mathbf{x}) = \inf\{q : \; \Delta f_d^{(k)}(\mathbf{x}) = O_{d \to \infty}(d^q)\}, \quad q_{\Delta f'_{a \vee w}}^{(k)}(\mathbf{x}) = \inf\{q : \; \Delta f_{d,a \vee w}^{(k),'}(\mathbf{x}) = O_{d \to \infty}(d^q)\}. \tag{41}$$

**Lemma 1.** *Assume $\mathcal{D}$ is a continuous distribution. Then following hold:*

1. *$\forall k \geq 0 \; \forall \mathbf{x}, y \; q_{\nabla \ell}^{(k)}(\mathbf{x}, y) \leq 0$, while $[y f_\infty^{(k)}(\mathbf{x}) < 0]$ implies $q_{\nabla \ell}^{(k)}(\mathbf{x}, y) = 0$.*

2. *$q_{a \vee w}^{(0)} = 0$, $q_f^{(0)}(\mathbf{x}) = q_\sigma + \frac{1}{2}$ $\mathbf{x}$-a.e.*

3. $\forall k \geq 0$ $q_{\Delta a/\Delta w}^{(k)} = \tilde{q}_{a\vee w} + q_{\sigma} + q_{w/a}^{(k)} + q_{\nabla\ell}^{(k)}(\mathbf{x}^{(k)}, y^{(k)})$ $(\mathbf{x}_{a\vee w}^{(k)}, y_{a\vee w}^{(k)})$-a.s.

4. $\forall k \geq 0$ $q_{\Delta f'_{a\vee w}}^{(k)}(\mathbf{x}) = 2q_{\sigma} + 1 + \tilde{q}_{a\vee w} + 2q_{w/a}^{(k)} + q_{\nabla\ell}^{(k)}(\mathbf{x}_{a\vee w}^{(k)}, y_{a\vee w}^{(k)})$ $\mathbf{x}$-a.e. $(\mathbf{x}_{a\vee w}^{(k)}, y_{a\vee w}^{(k)})$-a.s.

5. $\forall k \geq 0$ $q_{\sigma} + \tilde{q}_w + q_a^{(k)} \leq 0$ *implies that for sufficiently small* $\hat{\eta}_a^*$ *and* $\hat{\eta}_w^*$ $q_{\Delta f}^{(k)}(\mathbf{x}) = \max(q_{\Delta f'_a}^{(k)}(\mathbf{x}), q_{\Delta f'_w}^{(k)}(\mathbf{x}))$ $\mathbf{x}$-a.e. $(\mathbf{x}_a^{(k)}, y_a^{(k)}, \mathbf{x}_w^{(k)}, y_w^{(k)})$-a.s.

6. $\forall k \geq 0$ $q_{a\vee w}^{(k+1)} = \max(q_{a\vee w}^{(k)}, q_{\Delta a/\Delta w}^{(k)})$ $(\mathbf{x}^{(k)}, y^{(k)})$-a.s., $q_f^{(k+1)}(\mathbf{x}) = \max(q_f^{(k)}(\mathbf{x}), q_{\Delta f}^{(k)}(\mathbf{x}))$ $\mathbf{x}$-a.e. $(\mathbf{x}_a^{(k)}, y_a^{(k)}, \mathbf{x}_w^{(k)}, y_w^{(k)})$-a.s.

*Proof.* (1) follows from the fact that $\partial\ell(y, z)/\partial z$ is bounded $\forall y$, while $|\partial\ell(y, z)/\partial z| \in [1/2, 1]$ when $yz < 0$.

$\hat{a}_r^{(0)} \sim \mathcal{N}(0, 1)$ which is not zero and does not depend on $d$, hence $q_a^{(0)} = 0$; similar holds for $\mathbf{w}$. For $\mathbf{x} \neq 0$ we have $f_d^{(0)}(\mathbf{x}) = \sigma \sum_{r=1}^d \hat{a}_r^{(0)} \phi(\hat{\mathbf{w}}_r^{(0),T}\mathbf{x}) = \Theta_{d\to\infty}(d^{1/2+q_{\sigma}})$ due to the Central Limit Theorem. Hence (2) holds.

Since $\mathcal{D}$ is a.c. wrt Lebesgue measure on $\mathbb{R}^{1+d_{\mathbf{x}}}$, and $\phi$ is real analytic and non-zero, $\phi(\hat{\mathbf{w}}_r^{(k),T}\mathbf{x}_{a\vee w}^{(k)}) \neq 0$ and $\phi'(\hat{\mathbf{w}}_r^{(k),T}\mathbf{x}_{a\vee w}^{(k)})$ is well-defined $(\mathbf{x}_{a\vee w}^{(k)}, y_{a\vee w}^{(k)})$-a.s. This implies that $q_{\Delta a/\Delta w}^{(k)} = \tilde{q}_{a\vee w} + q_{\sigma} + q_{w/a}^{(k)} + q_{\nabla\ell}^{(k)}(\mathbf{x}_{a\vee w}^{(k)}, y_{a\vee w}^{(k)})$ $(\mathbf{x}_{a\vee w}^{(k)}, y_{a\vee w}^{(k)})$-a.s., which is exactly (3).

Consider $\Delta f_{d,a}^{(k),\prime}$:

$$\Delta f_{d,a}^{(k),\prime}(\mathbf{x}) = -\nabla_f^{(k)}\ell(\mathbf{x}_a^{(k)}, y_a^{(k)}) \, K_{a,d}^{(k)}(\mathbf{x}, \mathbf{x}_a^{(k)}) =$$

$$= -\nabla_f^{(k)}\ell(\mathbf{x}_a^{(k)}, y_a^{(k)}) \, (d/d^*)^{\tilde{q}_a}\sigma^2 \sum_{r=1}^d \phi(\hat{\mathbf{w}}_r^{(k),T}\mathbf{x})\phi(\hat{\mathbf{w}}_r^{(k),T}\mathbf{x}_a^{(k)}). \quad (42)$$

For the same reason as discussed above $\phi(\hat{\mathbf{w}}_r^{(k),T}\mathbf{x}_a^{(k)}) \neq 0$ $(\mathbf{x}_a^{(k)}, y_a^{(k)})$-a.s., and $\phi(\hat{\mathbf{w}}_r^{(k),T}\mathbf{x}) \neq 0$ $\mathbf{x}$-a.e. Since the summands are distributed identically and are generally non-zero, the sum introduces a factor of $d$. Indeed, an expectation of the sum is a sum of expectations by a linearity property. Moreover, the absolute value of the $k$-th moment of the sum of $d$ indentically distributed terms does not exceed $d^k$ times the $k$-th moment of each summand. Hence the sum itself scales as $d$. Since $\phi$ is asymptotically linear, each $\phi$-term scales as $d^{q_w^{(k)}}$. Collecting all terms together, we obtain $q_{\Delta f'_a}^{(k)}(\mathbf{x}) = 2q_{\sigma} + 1 + \tilde{q}_a + 2q_w^{(k)} + q_{\nabla\ell}^{(k)}(\mathbf{x}_a^{(k)}, y_a^{(k)})$ $\mathbf{x}$-a.e. $(\mathbf{x}_a^{(k)}, y_a^{(k)})$-a.s. Following the same steps for $\Delta f_w^{(k),\prime}$, we get (4).

Let us overview $\Delta f_d^{(k)}(\mathbf{x})$ in detail:

$$\Delta f_d^{(k)}(\mathbf{x}) = \sum_{r=1}^d \Bigg( \sum_{j=1}^\infty \frac{1}{j!} \frac{\partial^j f_d(\mathbf{x})}{\partial \hat{w}_r^{i_1} \dots \partial \hat{w}_r^{i_j}} \Bigg|_{\substack{\hat{\mathbf{w}}_r = \hat{\mathbf{w}}_r^{(k)} \\ \hat{a}_r = \hat{a}_r^{(k)}}} \Delta\hat{w}_r^{(k),i_1} \dots \Delta\hat{w}_r^{(k),i_j} +$$

$$+ \sum_{j=1}^\infty \frac{1}{j!} \frac{\partial^j f_d(\mathbf{x})}{\partial \hat{a}_r \partial \hat{w}_r^{i_2} \dots \partial \hat{w}_r^{i_j}} \Bigg|_{\substack{\hat{\mathbf{w}}_r = \hat{\mathbf{w}}_r^{(k)} \\ \hat{a}_r = \hat{a}_r^{(k)}}} \Delta\hat{a}_r \Delta\hat{w}_r^{(k),i_2} \dots \Delta\hat{w}_r^{(k),i_j} \Bigg) =$$

$$= \sum_{r=1}^d \Bigg( \sum_{j=1}^\infty \frac{1}{j!}(-1)^j \hat{\eta}_w^j \sigma^{j+1}(\nabla_{f_d}^{(k)}\ell(\mathbf{x}_w^{(k)}, y_w^{(k)}))^j \times$$

$$\times (\hat{a}_r^{(k)})^{j+1}(\phi'(\hat{\mathbf{w}}_r^{(k),T}\mathbf{x}_w^{(k)}))^j \phi^{(j)}(\hat{\mathbf{w}}_r^{(k),T}\mathbf{x})(\mathbf{x}_w^{(k),T}\mathbf{x})^j +$$

$$+ \sum_{j=1}^\infty \frac{1}{j!}(-1)^j \hat{\eta}_a \hat{\eta}_w^{j-1} \sigma^{j+1} \nabla_{f_d}^{(k)}\ell(\mathbf{x}_a^{(k)}, y_a^{(k)})(\nabla_{f_d}^{(k)}\ell(\mathbf{x}_w^{(k)}, y_w^{(k)}))^{j-1} \times$$

$$\times (\hat{a}_r^{(k)})^{j-1}\phi(\hat{\mathbf{w}}_r^{(k),T}\mathbf{x}_a^{(k)})(\phi'(\hat{\mathbf{w}}_r^{(k),T}\mathbf{x}_w^{(k)}))^{j-1}\phi^{(j-1)}(\hat{\mathbf{w}}_r^{(k),T}\mathbf{x})(\mathbf{x}_w^{(k),T}\mathbf{x})^{j-1} \Bigg). \quad (43)$$

Assumption $q_\sigma + \tilde{q}_w + q_a^{(k)} \leq 0$ implies $\hat{\eta}_w^j \sigma^{j+1} (\hat{a}_r^{(k)})^{j+1} = O_{d\to\infty}(\hat{\eta}_w \sigma^2 (\hat{a}_r^{(k)})^2)$ and $\hat{\eta}_a \hat{\eta}_w^{j-1} \sigma^{j+1} (\hat{a}_r^{(k)})^{j-1} = O_{d\to\infty}(\hat{\eta}_a \sigma^2)$.

Since $q_{\nabla\ell}^{(k)}(\mathbf{x}, y) \leq 0 \; \forall \mathbf{x}, y$ due to (1), $(\nabla_{f_d}^{(k)} \ell(\mathbf{x}_w^{(k)}, y_w^{(k)}))^j = O_{d\to\infty}(\nabla_{f_d}^{(k)} \ell(\mathbf{x}_w^{(k)}, y_w^{(k)}))$ and $\nabla_{f_d}^{(k)} \ell(\mathbf{x}_a^{(k)}, y_a^{(k)})(\nabla_{f_d}^{(k)} \ell(\mathbf{x}_w^{(k)}, y_w^{(k)}))^{j-1} = O_{d\to\infty}(\nabla_{f_d}^{(k)} \ell(\mathbf{x}_a^{(k)}, y_a^{(k)}))$.

Since $\phi(z) = \Theta_{z\to\infty}(z)$, $\phi'(\hat{\mathbf{w}}_r^{(k),T} \mathbf{x}_w^{(k)}) = O_{d\to\infty}(1)$ and $(\phi'(\hat{\mathbf{w}}_r^{(k),T} \mathbf{x}_w^{(k)}))^j = O_{d\to\infty}(\phi'(\hat{\mathbf{w}}_r^{(k),T} \mathbf{x}_w^{(k)}))$ for $j \geq 1$.

Hence for small enough $\hat{\eta}_a^*$ and $\hat{\eta}_w^*$ the first term of each sum which corresponds to $j = 1$ dominates all others, even in the limit of infinite $d$:

$$\Delta f_d^{(k)}(\mathbf{x}) = -\sum_{r=1}^d \left( \hat{\eta}_w \sigma^2 \nabla_{f_d}^{(k)} \ell(\mathbf{x}_w^{(k)}, y_w^{(k)})(\hat{a}_r^{(k)})^2 \phi'(\hat{\mathbf{w}}_r^{(k),T} \mathbf{x}_w^{(k)}) \phi'(\hat{\mathbf{w}}_r^{(k),T} \mathbf{x}) \mathbf{x}_w^{(k),T} \mathbf{x} + \right.$$
$$+ \hat{\eta}_a \sigma^2 \nabla_{f_d}^{(k)} \ell(\mathbf{x}_a^{(k)}, y_a^{(k)}) \phi(\hat{\mathbf{w}}_r^{(k),T} \mathbf{x}_a^{(k)}) \phi(\hat{\mathbf{w}}_r^{(k),T} \mathbf{x}) +$$
$$\left. + o_{\hat{\eta}_{a\vee w}^* \to 0} \left( O_{d\to\infty} \left( \left( \hat{\eta}_a \nabla_{f_d}^{(k)} \ell(\mathbf{x}_a^{(k)}, y_a^{(k)}) + \hat{\eta}_w \nabla_{f_d}^{(k)} \ell(\mathbf{x}_w^{(k)}, y_w^{(k)})(\hat{a}_r^{(k)})^2 \right) \sigma^2 \right) \right) \right) =$$
$$= \hat{\eta}_w^* \Delta f_{d,w}^{(k),\prime}(\mathbf{x}) + \hat{\eta}_a^* \Delta f_{d,a}^{(k),\prime}(\mathbf{x}) +$$
$$+ o_{\hat{\eta}_{a\vee w}^* \to 0} \left( O_{d\to\infty} \left( \left( \hat{\eta}_a \nabla_{f_d}^{(k)} \ell(\mathbf{x}_a^{(k)}, y_a^{(k)}) + \hat{\eta}_w \nabla_{f_d}^{(k)} \ell(\mathbf{x}_w^{(k)}, y_w^{(k)})(\hat{a}_r^{(k)})^2 \right) \sigma^2 d \right) \right). \quad (44)$$

Note that two summands depend on $(\mathbf{x}_w^{(k)}, y_w^{(k)})$ and $(\mathbf{x}_a^{(k)}, y_a^{(k)})$ respectively, which do not depend on each other. Hence $q_{\Delta f}^{(k)}(\mathbf{x}) = \max(q_{\Delta f_a'}^{(k)}(\mathbf{x}), q_{\Delta f_w'}^{(k)}(\mathbf{x}))$ $\mathbf{x}$-a.e. $(\mathbf{x}_{a\vee w}^{(k)}, y_{a\vee w}^{(k)})$-a.s., which is (5). Note that the o-term does not alter the exponent. Indeed,

$$\left( \hat{\eta}_a \nabla_{f_d}^{(k)} \ell(\mathbf{x}_a^{(k)}, y_a^{(k)}) + \hat{\eta}_w \nabla_{f_d}^{(k)} \ell(\mathbf{x}_w^{(k)}, y_w^{(k)})(\hat{a}_r^{(k)})^2 \right) \sigma^2 d =$$
$$= O_{d\to\infty} \left( d^{1+2q_\sigma + \max(\tilde{q}_a + q_{\nabla\ell}^{(k)}(\mathbf{x}_a^{(k)}, y_a^{(k)}), \tilde{q}_w + q_{\nabla\ell}^{(k)}(\mathbf{x}_w^{(k)}, y_w^{(k)}) + 2q_a^{(k)})} \right) =$$
$$= O_{d\to\infty} \left( d^{1+2q_\sigma + \max(\tilde{q}_a + q_{\nabla\ell}^{(k)}(\mathbf{x}_a^{(k)}, y_a^{(k)}) + 2q_w^{(k)}, \tilde{q}_w + q_{\nabla\ell}^{(k)}(\mathbf{x}_w^{(k)}, y_w^{(k)}) + 2q_a^{(k)})} \right) =$$
$$= O_{d\to\infty} \left( d^{\max(q_{\Delta f_a'}^{(k)}(\mathbf{x}), q_{\Delta f_w'}^{(k)}(\mathbf{x}))} \right). \quad (45)$$

One before the last equality holds, because $q_w^{(k)} \geq 0$ due to (2) and (6), while the last equality holds due to (4).

By definition we have $\hat{a}_r^{(k+1)} = \hat{a}_r^{(k)} + \Delta \hat{a}_r^{(k)}$. Since the second term depends on $(\mathbf{x}_a^{(k)}, y_a^{(k)})$, while the first term does not, we get $q_a^{(k+1)} = \max(q_a^{(k)}, q_{\Delta a}^{(k)})$. Similar holds for $\hat{\mathbf{w}}_r$ and $f_d^{(k)}(\mathbf{x})$ $\mathbf{x}$-a.e., which gives (6). $\qquad \square$

**Lemma 2.** *Assume $\mathcal{D}$ is a continuous distribution, $k_{term} = +\infty$ and $\tilde{q}_a = \tilde{q}_w = \tilde{q}$. Then*

1. *If $q_\sigma + \tilde{q} \leq 0$ then $\forall k \geq 0 \; q_{a\vee w}^{(k)} = 0 \; (\mathbf{x}_a^{(:k-1)}, y_a^{(:k-1)}, \mathbf{x}_w^{(:k-1)}, y_w^{(:k-1)})$-a.s.*

2. *If $q_\sigma + \tilde{q} > 0$ then $\forall k \geq 0 \; q_{a\vee w}^{(k)} = k(q_\sigma + \tilde{q})$ with positive probability wrt $(\mathbf{x}_a^{(:k-1)}, y_a^{(:k-1)}, \mathbf{x}_w^{(:k-1)}, y_w^{(:k-1)})$.*

*Proof.* Here and in subsequent proofs we will write "almost surely" meaning "almost surely wrt $(\mathbf{x}_a^{(:k)}, y_a^{(:k)}, \mathbf{x}_w^{(:k)}, y_w^{(:k)})$" for appropriate $k$; we apply a similar shortening for "with positive probability wrt $(\mathbf{x}_a^{(:k)}, y_a^{(:k)}, \mathbf{x}_w^{(:k)}, y_w^{(:k)})$".

If $q_\sigma + \tilde{q} \leq 0$ then statements 1, 2, 3 and 6 of Lemma 1 imply $\forall k \geq 0 \; q_{a\vee w}^{(k)} = 0$ a.s.

Assume $q_\sigma + \tilde{q} > 0$. We will prove that $\forall k \geq 0 \; q_{a\vee w}^{(k)} = \max(0, k(q_\sigma + \tilde{q}))$ with positive probability by induction. Induction base is given by Lemma 1-2.

Combining the induction assumption and Lemma 1-3 we get $q^{(k)}_{\Delta a/\Delta w} = (k+1)(q_\sigma + \tilde{q}) + q^{(k)}_{\nabla \ell}(\mathbf{x}^{(k)}, y^{(k)})$ with positive probability wrt $(\mathbf{x}^{(:k-1)}_a, y^{(:k-1)}_a, \mathbf{x}^{(:k-1)}_w, y^{(:k-1)}_w)$ $(\mathbf{x}^{(k)}_{a\vee w}, y^{(k)}_{a\vee w})$-a.s.

Since $k_{term} = +\infty > k$, $y^{(k)}_{a\vee w} f^{(k)}_d(\mathbf{x}^{(k)}_{a\vee w}) < 0$ with positive probability wrt $(\mathbf{x}^{(k)}_a, y^{(k)}_a, \mathbf{x}^{(k)}_w, y^{(k)}_w)$, and Lemma 1-1 implies that $q^{(k)}_{\Delta a/\Delta w} = (k+1)(q_\sigma + \tilde{q})$ with positive probability wrt $(\mathbf{x}^{(:k)}_a, y^{(:k)}_a, \mathbf{x}^{(:k)}_w, y^{(:k)}_w)$.

Finally, Lemma 1-6 concludes the proof of the induction step. $\qquad\square$

**Lemma 3.** *Assume $\mathcal{D}$ is a continuous distribution, $k_{term,\infty} = +\infty$, $\tilde{q}_a = \tilde{q}_w = \tilde{q}$ and $q_\sigma + \tilde{q} \le 0$. Then $\forall k \ge 0$*

1. $y^{(k)}_{a\vee w} f^{(k)}_\infty(\mathbf{x}^{(k)}_{a\vee w}) \quad < \quad 0 \quad implies \quad q^{(k)}_{\Delta f'_{a\vee w}}(\mathbf{x}) \quad = \quad 2q_\sigma + 1 + \tilde{q}$ $\mathbf{x}$-a.e. $(\mathbf{x}^{(:k-1)}_a, y^{(:k-1)}_a, \mathbf{x}^{(:k-1)}_w, y^{(:k-1)}_w)$-a.s.

2. $y^{(k)}_a f^{(k)}_\infty(\mathbf{x}^{(k)}_a) < 0$ *and* $y^{(k)}_w f^{(k)}_\infty(\mathbf{x}^{(k)}_w) < 0$ *imply* $q^{(k)}_{\Delta f}(\mathbf{x}) = 2q_\sigma + 1 + \tilde{q}$ $\mathbf{x}$-a.e. $(\mathbf{x}^{(:k-1)}_a, y^{(:k-1)}_a, \mathbf{x}^{(:k-1)}_w, y^{(:k-1)}_w)$-a.s. *for sufficiently small $\hat{\eta}^*_a$ and $\hat{\eta}^*_w$.*

*Proof.* By Lemma 2 $\forall k \ge 0$ $q^{(k)}_{a\vee w} = 0$ a.s. Since $y^{(k)}_{a\vee w} f^{(k)}_\infty(\mathbf{x}^{(k)}_{a\vee w}) < 0$, $q^{(k)}_{\nabla \ell} = 0$ due to Lemma 1-1. Given this, Lemma 1-4 implies $\forall k \ge 0$ $q^{(k)}_{\Delta f'_{a\vee w}}(\mathbf{x}) = 2q_\sigma + 1 + \tilde{q}$ $\mathbf{x}$-a.e. a.s. Hence by virtue of Lemma 1-5 $\forall k \ge 0$ $q^{(k)}_{\Delta f}(\mathbf{x}) = 2q_\sigma + 1 + \tilde{q}$ $\mathbf{x}$-a.e. a.s. for sufficiently small $\hat{\eta}^*_a$ and $\hat{\eta}^*_w$. $\qquad\square$

**Proposition 3.** *Suppose $\tilde{q}_a = \tilde{q}_w = \tilde{q}$ and $\mathcal{D}$ is a continuous distribution. Then Condition 4 requires $q_\sigma + \tilde{q} \in [-1/2, 0]$ to hold.*

*Proof.* By Lemma 2 if $q_\sigma + \tilde{q} > 0$ then $q^{(k)}_{a\vee w} = k(q_\sigma + \tilde{q})$ with positive probability. At the same time by virtue of Lemma 1-1 $k_{term,\infty} = +\infty$ implies $q^{(k)}_{\nabla \ell} = 0$ with positive probability. Given this, Lemma 1-4 implies $q^{(k)}_{\Delta f'_{a\vee w}}(\mathbf{x}) = q_\sigma + 1 + (2k+1)(q_\sigma + \tilde{q})$ $\mathbf{x}$-a.e. with positive probability. This means that the last quantity cannot be almost surely equal to $q^{(k_{balance})}_f(\mathbf{x})$ for any $k_{balance}$ independent on $k$. Since $\Delta f^{(k),'}_{d,a\vee w}(\mathbf{x}) = \Theta_{d\to\infty}(f^{(k_{balance})}_d(\mathbf{x}))$ requires $q^{(k)}_{\Delta f'_{a\vee w}}(\mathbf{x}) = q^{(k_{balance})}_f(\mathbf{x})$, we conclude that Condition 4 cannot be satisfied if $q_\sigma + \tilde{q} > 0$.

Hence $q_\sigma + \tilde{q} \le 0$. Then by Lemma 3 $\forall k \ge 0$ $y^{(k)}_a f^{(k)}_\infty(\mathbf{x}^{(k)}_a) < 0$ and $y^{(k)}_w f^{(k)}_\infty(\mathbf{x}^{(k)}_w) < 0$ imply $q^{(k)}_{\Delta f}(\mathbf{x}) = 2q_\sigma + 1 + \tilde{q}$ $\mathbf{x}$-a.e. $(\mathbf{x}^{(:k-1)}_a, y^{(:k-1)}_a, \mathbf{x}^{(:k-1)}_w, y^{(:k-1)}_w)$-a.s. for sufficiently small $\hat{\eta}^*_a$ and $\hat{\eta}^*_w$. We will show that Condition 4 requires $q_\sigma + \tilde{q} \in [-1/2, 0]$ to hold already for these sufficiently small $\hat{\eta}^*_a$ and $\hat{\eta}^*_w$.

Suppose $y^{(k)}_a f^{(k)}_\infty(\mathbf{x}^{(k)}_a) < 0$ and $y^{(k)}_w f^{(k)}_\infty(\mathbf{x}^{(k)}_w) < 0$. Given this, points 1 and 6 of Lemma 1 imply $\forall k_{balance} \ge 1$ $q^{k_{balance}}_f(\mathbf{x}) = \max(q^{(0)}_f(\mathbf{x}), 2q_\sigma + 1 + \tilde{q}) = \max(q_\sigma + \frac{1}{2}, 2q_\sigma + 1 + \tilde{q})$ $\mathbf{x}$-a.e. a.s. Hence $q^{(k)}_{\Delta f'_{a\vee w}}(\mathbf{x}) = q^{(k_{balance})}_f(\mathbf{x})$ $\mathbf{x}$-a.e. a.s. if and only if $q_\sigma + \frac{1}{2} \le 2q_\sigma + 1 + \tilde{q}$, which is $q_\sigma + \tilde{q} \ge -1/2$; we can take $k_{balance} = 1$ without loss of generality. Having $q^{(k)}_{\Delta f'_{a\vee w}}(\mathbf{x}) = q^{(k_{balance})}_f(\mathbf{x})$ is necessary to have $\Delta f^{(k),'}_{d,a\vee w}(\mathbf{x}) = \Theta_{d\to\infty}(f^{(k_{balance})}_d(\mathbf{x}))$.

Summing all together, Condition 4 requires $q_\sigma + \tilde{q} \in [-1/2, 0]$ to hold. $\qquad\square$

### B.2 Proof of Proposition 2

**Proposition 4.** *Let Condition 4 holds; then*

1. $f^{(0)}_d(\mathbf{x}) = \Theta_{d\to\infty}(1)$ *$\mathbf{x}$-a.e. is equivalent to $q_\sigma + 1/2 = 0$.*

2. $K^{(0)}_{d,a\vee w}(\mathbf{x}, \mathbf{x}') = \Theta_{d\to\infty}(1)$ *$(\mathbf{x}, \mathbf{x}')$-a.e. is equivalent to $2q_\sigma + \tilde{q} + 1 = 0$.*

3. $K_{d,a\vee w}^{(0)}(\mathbf{x},\mathbf{x}') = \Theta_{d\to\infty}(f_d^{(0)}(\mathbf{x}))$ $(\mathbf{x},\mathbf{x}')$-*a.e. is equivalent to* $q_\sigma + \tilde{q} + 1/2 = 0$.

4. $\Delta K_{d,wa\vee w}^{(0),'}(\mathbf{x},\mathbf{x}') = \Theta_{d\to\infty}(K_{d,w}^{(0)}(\mathbf{x},\mathbf{x}'))$ $(\mathbf{x},\mathbf{x}')$-*a.e.* *and* $\Delta K_{d,aw}^{(0),'}(\mathbf{x},\mathbf{x}') = \Theta_{d\to\infty}(K_{d,a}^{(0)}(\mathbf{x},\mathbf{x}'))$ $(\mathbf{x},\mathbf{x}')$-*a.e. is equivalent to* $q_\sigma + \tilde{q} = 0$.

*Proof.* Statement (1) directly follows from Lemma 1-2:

$$f_d^{(0)}(\mathbf{x}) = \sigma \sum_{r=1}^{d} \hat{a}_r^{(0)} \phi(\hat{\mathbf{w}}_r^{(0),T}\mathbf{x}) = \Theta_{d\to\infty}(d^{q_\sigma+1/2}) \tag{46}$$

$(\mathbf{x})$-a.e. due to the Central Limit Theorem.

Statement (2) follows from the definition of kernels and the Law of Large Numbers:

$$K_{a,d}^{(0)}(\mathbf{x},\mathbf{x}') = (d/d^*)^{\tilde{q}}\sigma^2 \sum_{r=1}^{d} \phi(\hat{\mathbf{w}}_r^{(0),T}\mathbf{x})\phi(\hat{\mathbf{w}}_r^{(0),T}\mathbf{x}') = \Theta_{d\to\infty}(d^{\tilde{q}+2q_\sigma+1}) \tag{47}$$

$(\mathbf{x},\mathbf{x}')$-a.e.; the same logic holds for the other kernel: $K_{w,d}^{(0)}(\mathbf{x},\mathbf{x}') = \Theta_{d\to\infty}(d^{\tilde{q}+2q_\sigma+1})$ $(\mathbf{x},\mathbf{x}')$-a.e.

Combining derivations of the two previous statements, we get the statement (3). Now we proceed to the last statement. Consider again the kernel $K_{a,d}^{(0)}$; a linear part of this increment with respect to proportionality factors of learning rates is given by, see eq. (34):

$$\Delta K_{aw,d}^{(0),'}(\mathbf{x},\mathbf{x}') = \left.\frac{\partial \Delta K_{a,d}^{(0)}(\mathbf{x},\mathbf{x}')}{\partial \hat{\eta}_w^*}\right|_{\hat{\eta}_w^*=0} =$$

$$= -(d/d^*)^{2\tilde{q}}\sigma^3 \sum_{r=1}^{d} \left( \phi(\hat{\mathbf{w}}_r^{(0),T}\mathbf{x})\phi'(\hat{\mathbf{w}}_r^{(0),T}\mathbf{x}') + \phi'(\hat{\mathbf{w}}_r^{(0),T}\mathbf{x})\phi(\hat{\mathbf{w}}_r^{(0),T}\mathbf{x}') \right) \times$$

$$\times \nabla_{f_d}^{(0)}\ell(\mathbf{x}_w^{(0)},y_w^{(0)})\hat{a}_r^{(0)}\phi'(\hat{\mathbf{w}}_r^{(0),T}\mathbf{x}_w^{(0)})(\mathbf{x}+\mathbf{x}')^T\mathbf{x}_w^{(0)}, \tag{48}$$

Hence $\Delta K_{aw,d}^{(0),'} = \Theta_{d\to\infty}(K_{a,d}^{(0)})$ is equivalent to $q_\sigma + \tilde{q} = 0$. Considering the second kernel $K_{w,d}^{(0)}$ and its increment is equivalent to the same condition. $\square$

## C   THE NUMBER OF DISTINCT LIMIT MODELS IS FINITE

It is easy to see that due to the Proposition 4 Condition 5 divides the well-definiteness band into 13 regions. We now show that when proportionality factors $\sigma^*$ and $\hat{\eta}_{a\vee w}^*$ are fixed, choosing a limit model evolution is equivalent to picking a single region from these 13.

Indeed, for any width $d$ a model evolution can be written as follows:

$$\Delta f_d^{(k)}(\mathbf{x}) = -\hat{\eta}_w^* \nabla_{f_d}^{(k)}\ell(\mathbf{x}_w^{(k)},y_w^{(k)})\Big( K_{w,d}^{(k)}(\mathbf{x},\mathbf{x}_w^{(k)}) +$$

$$+ O_{\substack{\hat{\eta}_{a\vee w}^*\to 0 \\ d\to\infty}}(\hat{\eta}_w^* \Delta K_{ww,d}^{(k),'}(\mathbf{x},\mathbf{x}_w^{(k)}) + \hat{\eta}_a^* \Delta K_{wa,d}^{(k),'}(\mathbf{x},\mathbf{x}_w^{(k)})) \Big) -$$

$$- \hat{\eta}_a^* \nabla_{f_d}^{(k)}\ell(\mathbf{x}_a^{(k)},y_a^{(k)})\Big( K_{a,d}^{(k)}(\mathbf{x},\mathbf{x}_a^{(k)}) + O_{\substack{\hat{\eta}_w^*\to 0 \\ d\to\infty}}(\hat{\eta}_w^* \Delta K_{aw,d}^{(k),'}(\mathbf{x},\mathbf{x}_a^{(k)})) \Big). \tag{49}$$

$$f_d^{(k+1)}(\mathbf{x}) = f_d^{(k)}(\mathbf{x}) + \Delta f_d^{(k)}(\mathbf{x}), \quad \nabla_{f_d}^{(k)}\ell(\mathbf{x},y) = \frac{-y}{1+\exp(f_d^{(k)}(\mathbf{x})y)}, \tag{50}$$

$$f_d^{(0)}(\mathbf{x}) = \sigma^* d^{q_\sigma} \sum_{r=1}^{d} \hat{a}_r^{(0)}\phi(\hat{\mathbf{w}}_r^{(0),T}\mathbf{x}), \quad (\hat{a}_r^{(0)}, \hat{\mathbf{w}}_r^{(0)}) \sim \mathcal{N}(0, I_{1+d_\mathbf{x}}). \tag{51}$$

Now we introduce normalized kernels:

$$\tilde{K}_{a,d}^{(k)}(\mathbf{x},\mathbf{x}') = (d/d^*)^{-1-\tilde{q}-2q_\sigma} K_{a,d}^{(k)}(\mathbf{x},\mathbf{x}') = \sigma^{*,2}\frac{d^*}{d}\sum_{r=1}^{d}\phi(\hat{\mathbf{w}}_r^{(k),T}\mathbf{x})\phi(\hat{\mathbf{w}}_r^{(k),T}\mathbf{x}'), \tag{52}$$

$$\tilde{K}_{w,d}^{(k)}(\mathbf{x}, \mathbf{x}') = (d/d^*)^{-1-\tilde{q}-2q_\sigma} K_{w,d}^{(k)}(\mathbf{x}, \mathbf{x}') = \sigma^{*,2} \frac{d^*}{d} \sum_{r=1}^{d} |\hat{a}_r^{(k)}|^2 \phi'(\hat{\mathbf{w}}_r^{(k),T}\mathbf{x}) \phi'(\hat{\mathbf{w}}_r^{(k),T}\mathbf{x}') \mathbf{x}^T \mathbf{x}'. \tag{53}$$

Note that after normalization kernels stay finite in the limit of large width due to the Law of Large Numbers. Similarly, we normalize logits, as well as kernel and logit increments:

$$\Delta \tilde{K}_{**,d}^{(k),'} = (d/d^*)^{-1-\tilde{q}-2q_\sigma} \Delta K_{**,d}^{(k),'}, \tag{54}$$

$$\Delta \tilde{f}_d^{(k)} = (d/d^*)^{-1-\tilde{q}-2q_\sigma} \Delta f_d^{(k)}, \quad \tilde{f}_d^{(k)} = (d/d^*)^{-1-\tilde{q}-2q_\sigma} f_d^{(k)}. \tag{55}$$

We then rewrite the model evolution as:

$$\Delta \tilde{f}_d^{(k)}(\mathbf{x}) = -\hat{\eta}_w^* \nabla_{f_d}^{(k)} \ell(\mathbf{x}_w^{(k)}, y_w^{(k)}) \Big( \tilde{K}_{w,d}^{(k)}(\mathbf{x}, \mathbf{x}_w^{(k)}) +$$
$$+ O_{\substack{\hat{\eta}_{a\vee w}^* \to 0 \\ d \to \infty}}(\hat{\eta}_w^* \Delta \tilde{K}_{ww,d}^{(k),'}(\mathbf{x}, \mathbf{x}_w^{(k)}) + \hat{\eta}_a^* \Delta \tilde{K}_{wa,d}^{(k),'}(\mathbf{x}, \mathbf{x}_w^{(k)}))\Big) -$$
$$- \hat{\eta}_a^* \nabla_{f_d}^{(k)} \ell(\mathbf{x}_a^{(k)}, y_a^{(k)}) \Big( \tilde{K}_{a,d}^{(k)}(\mathbf{x}, \mathbf{x}_a^{(k)}) + O_{\substack{\hat{\eta}_w^* \to 0 \\ d \to \infty}}(\hat{\eta}_w^* \Delta \tilde{K}_{aw,d}^{(k),'}(\mathbf{x}, \mathbf{x}_a^{(k)}))\Big). \tag{56}$$

$$\tilde{f}_d^{(k+1)}(\mathbf{x}) = \tilde{f}_d^{(k)}(\mathbf{x}) + \Delta \tilde{f}_d^{(k)}(\mathbf{x}) \quad \forall k \geq 0, \tag{57}$$

$$\tilde{f}_d^{(0)}(\mathbf{x}) = \sigma^*(d/d^*)^{-1-\tilde{q}-q_\sigma} \sum_{r=1}^{d} \hat{a}_r^{(0)} \phi(\hat{\mathbf{w}}_r^{(0),T}\mathbf{x}), \quad (\hat{a}_r^{(0)}, \hat{\mathbf{w}}_r^{(0)}) \sim \mathcal{N}(0, I_{1+d_\mathbf{x}}), \tag{58}$$

$$f_d^{(k)}(\mathbf{x}) = (d/d^*)^{1+\tilde{q}+2q_\sigma} \tilde{f}_d^{(k)}(\mathbf{x}), \quad \nabla_{f_d}^{(k)} \ell(\mathbf{x}, y) = \frac{-y}{1 + \exp(f_d^{(k)}(\mathbf{x})y)} \quad \forall k \geq 0. \tag{59}$$

## C.1 Constant normalized kernels case

Kernels $\tilde{K}_{a\vee w,d}^{(k)}$ are either constants (hence $\Delta \tilde{K}_{**,d}^{(k),'} \to 0$ as $d \to \infty$) or evolve with $k$ in the limit of large $d$. First assume they are constants; in this case $q_\sigma + \tilde{q} < 0$ due to Proposition 4-4, and

$$\Delta \tilde{f}_d^{(k)}(\mathbf{x}) = -\hat{\eta}_w^* \nabla_{f_d}^{(k)} \ell(\mathbf{x}_w^{(k)}, y_w^{(k)}) \Big( \tilde{K}_{w,d}^{(0)}(\mathbf{x}, \mathbf{x}_w^{(k)}) + o_{\substack{\hat{\eta}_{a\vee w}^* \to 0 \\ d \to \infty}}(1) \Big) -$$
$$- \hat{\eta}_a^* \nabla_{f_d}^{(k)} \ell(\mathbf{x}_a^{(k)}, y_a^{(k)}) \Big( \tilde{K}_{a,d}^{(0)}(\mathbf{x}, \mathbf{x}_a^{(k)}) + o_{\substack{\hat{\eta}_w^* \to 0 \\ d \to \infty}}(1) \Big). \tag{60}$$

Since normalized kernels $\tilde{K}_{a\vee w,d}^{(0)}$ converge to non-zero limit kernels $\tilde{K}_{a\vee w,\infty}^{(0)}$, we can rewrite the formula above as:

$$\Delta \tilde{f}_d^{(k)}(\mathbf{x}) = -\hat{\eta}_w^* \nabla_{f_d}^{(k)} \ell(\mathbf{x}_w^{(k)}, y_w^{(k)}) \Big( \tilde{K}_{w,\infty}^{(0)}(\mathbf{x}, \mathbf{x}_w^{(k)}) + o_{d\to\infty}(1) \Big) -$$
$$- \hat{\eta}_a^* \nabla_{f_d}^{(k)} \ell(\mathbf{x}_a^{(k)}, y_a^{(k)}) \Big( \tilde{K}_{a,\infty}^{(0)}(\mathbf{x}, \mathbf{x}_a^{(k)}) + o_{d\to\infty}(1) \Big). \tag{61}$$

$$\tilde{f}_d^{(0)}(\mathbf{x}) = \sigma^*(d/d^*)^{-1/2-\tilde{q}-q_\sigma} (\mathcal{N}(0, \sigma^{(0),2}(\mathbf{x})) + o_{d\to\infty}(1)), \tag{62}$$

where $\sigma^{(0)}(\mathbf{x})$ can be calculated in the same manner as in Lee et al. (2019). As required by Proposition 3 $1/2 + \tilde{q} + q_\sigma \geq 0$, hence $\tilde{f}_d^{(0)}(\mathbf{x}) = O_{d\to\infty}(1)$. This implies the following:

$$\nabla_{f_\infty}^{(0)} \ell(\mathbf{x}, y) = \lim_{d\to\infty} \nabla_{f_d}^{(0)} \ell(\mathbf{x}, y) =$$

$$= \lim_{d\to\infty} \frac{-y}{1 + \exp((d/d^*)^{1+\tilde{q}+2q_\sigma} \tilde{f}_d^{(0)}(\mathbf{x})y)} = \begin{cases} -y[\mathcal{N}(0, \sigma^{(0),2}(\mathbf{x}))y < 0] & \text{for } 1/2 + q_\sigma > 0; \\ \frac{-y}{1+\exp(\sigma^* d^{*,1/2} \mathcal{N}(0,\sigma^{(0),2}(\mathbf{x}))y)} & \text{for } 1/2 + q_\sigma = 0; \\ -y/2 & \text{for } 1/2 + q_\sigma < 0. \end{cases} \tag{63}$$

On the other hand, $\Delta \tilde{f}_d^{(0)}(\mathbf{x}) = \Theta_{d\to\infty}(1)$ with positive probability over $(\mathbf{x}_{a\lor w}^{(0)}, y_{a\lor w}^{(0)})$. Hence $\tilde{f}_d^{(0)} = O_{d\to\infty}(\Delta \tilde{f}_d^{(0)})$ and $\tilde{f}_d^{(1)} = \tilde{f}_d^{(0)} + \Delta \tilde{f}_d^{(0)} = \Theta_{d\to\infty}(1)$. For the same reason, $\tilde{f}_d^{(k+1)} = \tilde{f}_d^{(k)} + \Delta \tilde{f}_d^{(k)} = \Theta_{d\to\infty}(1) \; \forall k \geq 0$.

This implies the following:

$$\forall k \geq 0 \quad \nabla_{f_\infty}^{(k+1)}\ell(\mathbf{x},y) = \lim_{d\to\infty} \nabla_{f_d}^{(k+1)}\ell(\mathbf{x},y) = \lim_{d\to\infty} \frac{-y}{1+\exp((d/d^*)^{1+\tilde{q}+2q_\sigma}\tilde{f}_d^{(k+1)}(\mathbf{x})y)} =$$

$$= \begin{cases} -y[\lim_{d\to\infty}\tilde{f}_d^{(k+1)}(\mathbf{x})y < 0] & \text{for } 1+\tilde{q}+2q_\sigma > 0; \\ \frac{-y}{1+\exp(\lim_{d\to\infty}\tilde{f}_d^{(k+1)}(\mathbf{x})y)} & \text{for } 1+\tilde{q}+2q_\sigma = 0; \quad (64) \\ -y/2 & \text{for } 1+\tilde{q}+2q_\sigma < 0. \end{cases}$$

If we define $f_\infty^{(k)}(\mathbf{x}) = \lim_{d\to\infty} f_d^{(k)}(\mathbf{x})$, we get the following limit dynamics:

$$\Delta \tilde{f}_\infty^{(k)}(\mathbf{x}) = -\hat{\eta}_w^* \nabla_{f_\infty}^{(k)}\ell(\mathbf{x}_w^{(k)}, y_w^{(k)})\tilde{K}_{w,\infty}^{(0)}(\mathbf{x}, \mathbf{x}_w^{(k)}) - \hat{\eta}_a^* \nabla_{f_\infty}^{(k)}\ell(\mathbf{x}_a^{(k)}, y_a^{(k)})\tilde{K}_{a,\infty}^{(0)}(\mathbf{x}, \mathbf{x}_a^{(k)}), \quad (65)$$

$$\tilde{K}_{a,\infty}^{(0)}(\mathbf{x}, \mathbf{x}') = \sigma^{*,2}d^* \mathbb{E}_{\hat{\mathbf{w}}\sim\mathcal{N}(0,I_{d_\mathbf{x}})}\phi(\hat{\mathbf{w}}^T\mathbf{x})\phi(\hat{\mathbf{w}}^T\mathbf{x}'), \quad (66)$$

$$\tilde{K}_{w,\infty}^{(0)}(\mathbf{x}, \mathbf{x}') = \sigma^{*,2}d^* \mathbb{E}_{(\hat{a},\hat{\mathbf{w}})\sim\mathcal{N}(0,I_{1+d_\mathbf{x}})}|\hat{a}|^2\phi'(\hat{\mathbf{w}}^T\mathbf{x})\phi'(\hat{\mathbf{w}}^T\mathbf{x}')\mathbf{x}^T\mathbf{x}', \quad (67)$$

$$\tilde{f}_\infty^{(k+1)}(\mathbf{x}) = \tilde{f}_\infty^{(k)}(\mathbf{x}) + \Delta \tilde{f}_\infty^{(k)}(\mathbf{x}), \quad \tilde{f}_\infty^{(0)}(\mathbf{x}) = \begin{cases} \sigma^* d^{*,1/2}\mathcal{N}(0,\sigma^{(0),2}(\mathbf{x})) & \text{for } 1/2+\tilde{q}+q_\sigma = 0; \\ 0 & \text{for } 1/2+\tilde{q}+q_\sigma > 0; \end{cases}$$
$$(68)$$

$$\nabla_{f_\infty}^{(0)}\ell(\mathbf{x},y) = \begin{cases} -y[\mathcal{N}(0,\sigma^{(0),2}(\mathbf{x}))y < 0] & \text{for } 1/2+q_\sigma > 0; \\ \frac{-y}{1+\exp(\sigma^* d^{*,1/2}\mathcal{N}(0,\sigma^{(0),2}(\mathbf{x}))y)} & \text{for } 1/2+q_\sigma = 0; \quad (69) \\ -y/2 & \text{for } 1/2+q_\sigma < 0; \end{cases}$$

$$\nabla_{f_\infty}^{(k+1)}\ell(\mathbf{x},y) = \begin{cases} -y[\tilde{f}_\infty^{(k+1)}(\mathbf{x})y < 0] & \text{for } 1+\tilde{q}+2q_\sigma > 0; \\ \frac{-y}{1+\exp(\tilde{f}_\infty^{(k+1)}(\mathbf{x})y)} & \text{for } 1+\tilde{q}+2q_\sigma = 0; \quad \forall k \geq 0. \quad (70) \\ -y/2 & \text{for } 1+\tilde{q}+2q_\sigma < 0; \end{cases}$$

This dynamics is defined by proportionality factors $\sigma^*$, $d^*$, $\hat{\eta}_{a\lor w}^*$, and signs of three exponents: $1/2+q_\sigma$, $1+\tilde{q}+2q_\sigma$ and $1/2+\tilde{q}+q_\sigma$. Since we assume proportionality factors to be fixed, choosing signs of exponents is equivalent to choosing a limit model. Note that these exponents exactly correspond to those mentioned in Proposition 4, points 1, 2 and 3. One can easily notice from Figure 1 (left) that given $q_\sigma + \tilde{q} < 0$, there are 8 distinct sign configurations.

Note also that since we are interested in binary classification problems, only the sign of logits matters. Since $f_d^{(k)} = (d/d^*)^{1+\tilde{q}+2q_\sigma}\tilde{f}_d^{(k)}$, signs of $f_d^{(k)}$ and of $\tilde{f}_d^{(k)}$ are the same for all $d$. Hence $\forall \mathbf{x}, y$ $\lim_{d\to\infty}\text{sign}(f_d^{(k)}(\mathbf{x})) = \lim_{d\to\infty}\text{sign}(\tilde{f}_d^{(k)}(\mathbf{x})) = \text{sign}(\tilde{f}_\infty^{(k)}(\mathbf{x}))$.

### C.1.1 NTK LIMIT MODEL

We state here a special case of the NTK scaling ($q_\sigma = -1/2$, $\tilde{q} = 0$, see Jacot et al. (2018)) explicitly. Since in this case $1+\tilde{q}+2q_\sigma = 0$, we can omit tildas everywhere. This results in the following limit dynamics:

$$\Delta f_\infty^{(k)}(\mathbf{x}) = -\hat{\eta}_w^* \nabla_{f_\infty}^{(k)}\ell(\mathbf{x}_w^{(k)}, y_w^{(k)})K_{w,\infty}^{(0)}(\mathbf{x}, \mathbf{x}_w^{(k)}) - \hat{\eta}_a^* \nabla_{f_\infty}^{(k)}\ell(\mathbf{x}_a^{(k)}, y_a^{(k)})K_{a,\infty}^{(0)}(\mathbf{x}, \mathbf{x}_a^{(k)}), \quad (71)$$

$$K_{a,\infty}^{(0)}(\mathbf{x}, \mathbf{x}') = \sigma^{*,2}d^* \mathbb{E}_{\hat{\mathbf{w}}\sim\mathcal{N}(0,I_{d_\mathbf{x}})}\phi(\hat{\mathbf{w}}^T\mathbf{x})\phi(\hat{\mathbf{w}}^T\mathbf{x}'), \quad (72)$$

$$K_{w,\infty}^{(0)}(\mathbf{x}, \mathbf{x}') = \sigma^{*,2}d^* \mathbb{E}_{(\hat{a},\hat{\mathbf{w}})\sim\mathcal{N}(0,I_{1+d_\mathbf{x}})}|\hat{a}|^2\phi'(\hat{\mathbf{w}}^T\mathbf{x})\phi'(\hat{\mathbf{w}}^T\mathbf{x}')\mathbf{x}^T\mathbf{x}', \quad (73)$$

$$f_\infty^{(k+1)}(\mathbf{x}) = f_\infty^{(k)}(\mathbf{x}) + \Delta f_\infty^{(k)}(\mathbf{x}), \quad f_\infty^{(0)}(\mathbf{x}) = \sigma^* d^{*,1/2} \mathcal{N}(0, \sigma^{(0),2}(\mathbf{x})), \tag{74}$$

$$\nabla_{f_\infty}^{(k)} \ell(\mathbf{x}, y) = \frac{-y}{1 + \exp(f_\infty^{(k)}(\mathbf{x})y)} \quad \forall k \geq 0. \tag{75}$$

## C.2 Non-stationary normalized kernels case

Suppose now $q_\sigma + \tilde{q} = 0$. In this case $\Delta K_{d,wa\vee w}^{(0),'}(\mathbf{x}, \mathbf{x}') = \Theta_{d\to\infty}(K_{d,w}^{(0)}(\mathbf{x}, \mathbf{x}'))$ $(\mathbf{x}, \mathbf{x}')$-a.e. and $\Delta K_{d,aw}^{(0),'}(\mathbf{x}, \mathbf{x}') = \Theta_{d\to\infty}(K_{d,a}^{(0)}(\mathbf{x}, \mathbf{x}'))$ $(\mathbf{x}, \mathbf{x}')$-a.e. by virtue of the Proposition 4-4. Hence kernels evolve in the limit of large width (at least, for sufficiently small $\eta_{a\vee w}^*$).

If we follow the lines of the previous section, we will get a limit dynamics which is not closed:

$$\Delta \tilde{f}_\infty^{(k)}(\mathbf{x}) = -\hat{\eta}_w^* \nabla_{f_\infty}^{(k)} \ell(\mathbf{x}_w^{(k)}, y_w^{(k)}) \Big( \tilde{K}_{w,\infty}^{(k)}(\mathbf{x}, \mathbf{x}_w^{(k)}) +$$
$$+ O_{\hat{\eta}_{a\vee w}^* \to 0}(\hat{\eta}_w^* \Delta \tilde{K}_{ww,\infty}^{(k),'}(\mathbf{x}, \mathbf{x}_w^{(k)}) + \hat{\eta}_a^* \Delta \tilde{K}_{wa,\infty}^{(k),'}(\mathbf{x}, \mathbf{x}_w^{(k)})) \Big) -$$
$$- \hat{\eta}_a^* \nabla_{f_\infty}^{(k)} \ell(\mathbf{x}_a^{(k)}, y_a^{(k)}) \Big( \tilde{K}_{a,\infty}^{(k)}(\mathbf{x}, \mathbf{x}_a^{(k)}) + O_{\hat{\eta}_w^* \to 0}(\hat{\eta}_w^* \Delta \tilde{K}_{aw,\infty}^{(k),'}(\mathbf{x}, \mathbf{x}_a^{(k)})) \Big), \tag{76}$$

$$\tilde{f}_\infty^{(k+1)}(\mathbf{x}) = \tilde{f}_\infty^{(k)}(\mathbf{x}) + \Delta \tilde{f}_\infty^{(k)}(\mathbf{x}), \quad \tilde{f}_\infty^{(0)}(\mathbf{x}) = 0, \tag{77}$$

$$\nabla_{f_\infty}^{(0)} \ell(\mathbf{x}, y) = \begin{cases} -y[\mathcal{N}(0, \sigma^{(0),2}(\mathbf{x}))y < 0] & \text{for } 1/2 + q_\sigma > 0; \\ \frac{-y}{1 + \exp(\sigma^* d^{*,1/2} \mathcal{N}(0, \sigma^{(0),2}(\mathbf{x}))y)} & \text{for } 1/2 + q_\sigma = 0; \\ -y/2 & \text{for } 1/2 + q_\sigma < 0; \end{cases} \tag{78}$$

$$\nabla_{f_\infty}^{(k+1)} \ell(\mathbf{x}, y) = \begin{cases} -y[\tilde{f}_\infty^{(k+1)}(\mathbf{x})y < 0] & \text{for } 1 + q_\sigma > 0; \\ \frac{-y}{1 + \exp(\tilde{f}_\infty^{(k+1)}(\mathbf{x})y)} & \text{for } 1 + q_\sigma = 0; \quad \forall k \geq 0. \\ -y/2 & \text{for } 1 + q_\sigma < 0; \end{cases} \tag{79}$$

The reason for this is non-stationarity of kernels. As a workaround we consider a measure in the weight space:

$$\mu_d^{(k)} = \frac{1}{d} \sum_{r=1}^d \delta_{\hat{a}_r^{(k)}} \otimes \delta_{\hat{\mathbf{w}}_r^{(k)}}. \tag{80}$$

Recall the stochastic gradient descent dynamics:

$$\Delta \hat{a}_r^{(k)} = -\hat{\eta}_a^* \sigma^* \nabla_{f_d}^{(k)} \ell(\mathbf{x}_a^{(k)}, y_a^{(k)}) \, \phi(\hat{\mathbf{w}}_r^{(k),T} \mathbf{x}_a^{(k)}), \quad \hat{a}_r^{(0)} \sim \mathcal{N}(0, 1), \tag{81}$$

$$\Delta \hat{\mathbf{w}}_r^{(k)} = -\hat{\eta}_w^* \sigma^* \nabla_{f_d}^{(k)} \ell(\mathbf{x}_w^{(k)}, y_w^{(k)}) \, \hat{a}_r^{(k)} \phi'(\hat{\mathbf{w}}_r^{(k),T} \mathbf{x}_w^{(k)}) \mathbf{x}_w^{(k)}, \quad \hat{\mathbf{w}}_r^{(0)} \sim \mathcal{N}(0, I_{d_\mathbf{x}}). \tag{82}$$

Here we have replaced $\hat{\eta}_{a\vee w} \sigma$ with $\hat{\eta}_{a\vee w}^* \sigma^*$, because $q_\sigma + \tilde{q} = 0$. Similar to Rotskoff & Vanden-Eijnden (2019); Chizat & Bach (2018), this dynamics can be expressed in terms of the measure defined above:

$$\mu_d^{(k+1)} = \mu_d^{(k)} + \text{div}(\mu_d^{(k)} \Delta \theta_d^{(k)}), \quad \mu_d^{(0)} = \frac{1}{d} \sum_{r=1}^d \delta_{\hat{\theta}_r^{(0)}}, \quad \hat{\theta}_r^{(0)} \sim \mathcal{N}(0, I_{1+d_\mathbf{x}}) \quad \forall r \in [d], \tag{83}$$

$$\Delta \theta_d^{(k)}(\hat{a}, \hat{\mathbf{w}}) =$$
$$= -[\hat{\eta}_a^* \sigma^* \nabla_{f_d}^{(k)} \ell(\mathbf{x}_a^{(k)}, y_a^{(k)}) \phi(\hat{\mathbf{w}}^T \mathbf{x}_a^{(k)}), \; \hat{\eta}_w^* \sigma^* \nabla_{f_d}^{(k)} \ell(\mathbf{x}_w^{(k)}, y_w^{(k)}) \hat{a} \phi'(\hat{\mathbf{w}}^T \mathbf{x}_w^{(k)}) \mathbf{x}_w^{(k),T}]^T, \tag{84}$$

$$f_d^{(k)}(\mathbf{x}) = \sigma^* (d/d^*)^{1+q_\sigma} \int \hat{a} \phi(\hat{\mathbf{w}}^T \mathbf{x}) \, \mu_d^{(k)}(d\hat{a}, d\hat{\mathbf{w}}), \tag{85}$$

$$\nabla_{f_d}^{(k)}\ell(\mathbf{x}, y) = \frac{-y}{1 + \exp(f_d^{(k)}(\mathbf{x})y)} \quad \forall k \geq 0. \tag{86}$$

We rewrite the last equation in terms of $\tilde{f}_d^{(k)}(\mathbf{x}) = (d/d^*)^{-1-q_\sigma} f_d^{(k)}(\mathbf{x})$:

$$\tilde{f}_d^{(k)}(\mathbf{x}) = \sigma^* d^* \int \hat{a}\phi(\hat{\mathbf{w}}^T\mathbf{x})\, \mu_d^{(k)}(d\hat{a}, d\hat{\mathbf{w}}), \tag{87}$$

$$\nabla_{f_d}^{(k)}\ell(\mathbf{x}, y) = \frac{-y}{1 + \exp((d/d^*)^{1+q_\sigma}\tilde{f}_d^{(k)}(\mathbf{x})y)} \quad \forall k \geq 0. \tag{88}$$

This dynamics is closed. Taking the limit $d \to \infty$ yields:

$$\mu_\infty^{(k+1)} = \mu_\infty^{(k)} + \mathrm{div}(\mu_\infty^{(k)}\Delta\theta_\infty^{(k)}), \quad \mu_\infty^{(0)} = \mathcal{N}(0, I_{1+d_\mathbf{x}}), \tag{89}$$

$$\Delta\theta_\infty^{(k)}(\hat{a}, \hat{\mathbf{w}}) =$$
$$= -[\hat{\eta}_a^*\sigma^*\nabla_{f_\infty}^{(k)}\ell(\mathbf{x}_a^{(k)}, y_a^{(k)})\phi(\hat{\mathbf{w}}^T\mathbf{x}_a^{(k)}),\ \hat{\eta}_w^*\sigma^*\nabla_{f_\infty}^{(k)}\ell(\mathbf{x}_w^{(k)}, y_w^{(k)})\hat{a}\phi'(\hat{\mathbf{w}}^T\mathbf{x}_w^{(k)})\mathbf{x}_w^{(k),T}]^T, \tag{90}$$

$$\tilde{f}_\infty^{(k)}(\mathbf{x}) = \sigma^* d^* \int \hat{a}\phi(\hat{\mathbf{w}}^T\mathbf{x})\, \mu_\infty^{(k)}(d\hat{a}, d\hat{\mathbf{w}}), \tag{91}$$

$$\nabla_{f_\infty}^{(0)}\ell(\mathbf{x}, y) = \begin{cases} -y[\mathcal{N}(0, \sigma^{(0),2}(\mathbf{x}))y < 0] & \text{for } 1/2 + q_\sigma > 0; \\ \frac{-y}{1+\exp(\sigma^* d^{*,1/2}\mathcal{N}(0,\sigma^{(0),2}(\mathbf{x}))y)} & \text{for } 1/2 + q_\sigma = 0; \\ -y/2 & \text{for } 1/2 + q_\sigma < 0; \end{cases} \tag{92}$$

$$\nabla_{f_\infty}^{(k+1)}\ell(\mathbf{x}, y) = \begin{cases} -y[\tilde{f}_\infty^{(k+1)}(\mathbf{x})y < 0] & \text{for } 1 + q_\sigma > 0; \\ \frac{-y}{1+\exp(\tilde{f}_\infty^{(k+1)}(\mathbf{x})y)} & \text{for } 1 + q_\sigma = 0; \quad \forall k \geq 0. \\ -y/2 & \text{for } 1 + q_\sigma < 0; \end{cases} \tag{93}$$

Since proportionality factors $\sigma^*$, $d^*$, and $\hat{\eta}_{a\vee w}^*$ are assumed to be fixed, choosing $q_\sigma$ is sufficient to define the dynamics. Signs of exponents $1/2 + q_\sigma$ and $1 + q_\sigma$ give 5 distinct limit dynamics. Together with 8 limit dynamics for constant normalized kernels case, this gives 13 distinct limit dynamics, each corresponding to a region in the band of a dynamical stability (Figure 1, left).

As was noted earlier, only the sign of logits matters, and our $\tilde{f}_d^{(k)}$ preserve the sign for any $d$: $\forall\mathbf{x}$ $\lim_{d\to\infty}\mathrm{sign}(f_d^{(k)}(\mathbf{x})) = \lim_{d\to\infty}\mathrm{sign}(\tilde{f}_d^{(k)}(\mathbf{x})) = \mathrm{sign}(\tilde{f}_\infty^{(k)}(\mathbf{x}))$.

### C.2.1 MF LIMIT MODEL

We state here a special case of the mean-field scaling ($q_\sigma = -1$, $\tilde{q} = 1$, see Rotskoff & Vanden-Eijnden (2019) or Chizat & Bach (2018)) explicitly. Similar to NTK case, since $1 + q_\sigma = 0$ we can omit tildas. This results in the following limit dynamics:

$$\mu_\infty^{(k+1)} = \mu_\infty^{(k)} + \mathrm{div}(\mu_\infty^{(k)}\Delta\theta_\infty^{(k)}), \quad \mu_\infty^{(0)} = \mathcal{N}(0, I_{1+d_\mathbf{x}}), \tag{94}$$

$$\Delta\theta_\infty^{(k)}(\hat{a}, \hat{\mathbf{w}}) =$$
$$= -[\hat{\eta}_a^*\sigma^*\nabla_{f_\infty}^{(k)}\ell(\mathbf{x}_a^{(k)}, y_a^{(k)})\phi(\hat{\mathbf{w}}^T\mathbf{x}_a^{(k)}),\ \hat{\eta}_w^*\sigma^*\nabla_{f_\infty}^{(k)}\ell(\mathbf{x}_w^{(k)}, y_w^{(k)})\hat{a}\phi'(\hat{\mathbf{w}}^T\mathbf{x}_w^{(k)})\mathbf{x}_w^{(k),T}]^T, \tag{95}$$

$$f_\infty^{(k)}(\mathbf{x}) = \sigma^* d^* \int \hat{a}\phi(\hat{\mathbf{w}}^T\mathbf{x})\, \mu_\infty^{(k)}(d\hat{a}, d\hat{\mathbf{w}}), \quad \nabla_{f_\infty}^{(k)}\ell(\mathbf{x}, y) = \frac{-y}{1 + \exp(f_\infty^{(k)}(\mathbf{x})y)} \quad \forall k \geq 0. \tag{96}$$

### C.2.2 SYM-DEFAULT LIMIT MODEL

Another special case which deserves explicit formulation is what we have called a "sym-default" limit model. The corresponding scaling is: $q_\sigma = -1/2$, $\tilde{q} = 1/2$. The resulting limit dynamics is the following:

$$\mu_\infty^{(k+1)} = \mu_\infty^{(k)} + \operatorname{div}(\mu_\infty^{(k)}\Delta\theta_\infty^{(k)}), \quad \mu_\infty^{(0)} = \mathcal{N}(0, I_{1+d_\mathbf{x}}), \tag{97}$$

$$\Delta\theta_\infty^{(k)}(\hat{a}, \hat{\mathbf{w}}) =$$
$$= -[\hat{\eta}_a^* \sigma^* \nabla_{f_\infty}^{(k)}\ell(\mathbf{x}_a^{(k)}, y_a^{(k)})\phi(\hat{\mathbf{w}}^T\mathbf{x}_a^{(k)}), \ \hat{\eta}_w^* \sigma^* \nabla_{f_\infty}^{(k)}\ell(\mathbf{x}_w^{(k)}, y_w^{(k)})\hat{a}\phi'(\hat{\mathbf{w}}^T\mathbf{x}_w^{(k)})\mathbf{x}_w^{(k),T}]^T, \tag{98}$$

$$\tilde{f}_\infty^{(k)}(\mathbf{x}) = \sigma^* d^* \int \hat{a}\phi(\hat{\mathbf{w}}^T\mathbf{x})\,\mu_\infty^{(k)}(d\hat{a}, d\hat{\mathbf{w}}), \tag{99}$$

$$\nabla_{f_\infty}^{(0)}\ell(\mathbf{x}, y) = \frac{-y}{1 + \exp(\sigma^* d^{*,1/2}\mathcal{N}(0, \sigma^{(0),2}(\mathbf{x}))y)}, \tag{100}$$

$$\nabla_{f_\infty}^{(k+1)}\ell(\mathbf{x}, y) = -y[\tilde{f}_\infty^{(k+1)}(\mathbf{x})y < 0] \quad \forall k \geq 0. \tag{101}$$

## D  DEFAULT SCALING

Consider the special case of the default scaling: $q_\sigma = -1/2$, $\tilde{q}_a = 1$, $\tilde{q}_w = 0$. Then corresponding dynamics can be written as follows:

$$\Delta\hat{a}_r^{(k)} = -\hat{\eta}_a^* \sigma^* (d/d^*)^{1/2}\nabla_{f_d}^{(k)}\ell(\mathbf{x}_a^{(k)}, y_a^{(k)})\,\phi(\hat{\mathbf{w}}_r^{(k),T}\mathbf{x}_a^{(k)}), \quad \hat{a}_r^{(0)} \sim \mathcal{N}(0, 1), \tag{102}$$

$$\Delta\hat{\mathbf{w}}_r^{(k)} = -\hat{\eta}_w^* \sigma^* (d/d^*)^{-1/2}\nabla_{f_d}^{(k)}\ell(\mathbf{x}_w^{(k)}, y_w^{(k)})\,\hat{a}_r^{(k)}\phi'(\hat{\mathbf{w}}_r^{(k),T}\mathbf{x}_w^{(k)})\mathbf{x}_w^{(k)}, \quad \hat{\mathbf{w}}_r^{(0)} \sim \mathcal{N}(0, I_{d_\mathbf{x}}), \tag{103}$$

$$f_d^{(k)}(\mathbf{x}) = \sigma^* (d/d^*)^{-1/2}\sum_{r=1}^d \hat{a}_r^{(k)}\phi(\hat{\mathbf{w}}_r^{(k),T}\mathbf{x}), \quad \nabla_{f_d}^{(k)}\ell(\mathbf{x}, y) = \frac{-y}{1 + \exp(f_d^{(k)}(\mathbf{x})y)} \quad \forall k \geq 0. \tag{104}$$

As one can see, increments of output layer weights $\Delta\hat{a}_r^{(k)}$ diverge with $d$. We introduce their normalized versions: $\Delta\tilde{a}_r^{(k)} = (d/d^*)^{-1/2}\Delta\hat{a}_r^{(k)}$. Similarly, we normalize output layer weights themselves: $\tilde{a}_r^{(k)} = (d/d^*)^{-1/2}\hat{a}_r^{(k)}$. Then the dynamics transforms to:

$$\Delta\tilde{a}_r^{(k)} = -\hat{\eta}_a^* \sigma^* \nabla_{f_d}^{(k)}\ell(\mathbf{x}_a^{(k)}, y_a^{(k)})\,\phi(\hat{\mathbf{w}}_r^{(k),T}\mathbf{x}_a^{(k)}), \quad \tilde{a}_r^{(0)} \sim \mathcal{N}(0, (d/d^*)^{-1}), \tag{105}$$

$$\Delta\hat{\mathbf{w}}_r^{(k)} = -\hat{\eta}_w^* \sigma^* \nabla_{f_d}^{(k)}\ell(\mathbf{x}_w^{(k)}, y_w^{(k)})\,\tilde{a}_r^{(k)}\phi'(\hat{\mathbf{w}}_r^{(k),T}\mathbf{x}_w^{(k)})\mathbf{x}_w^{(k)}, \quad \hat{\mathbf{w}}_r^{(0)} \sim \mathcal{N}(0, I_{d_\mathbf{x}}), \tag{106}$$

$$f_d^{(k)}(\mathbf{x}) = \sigma^* \sum_{r=1}^d \tilde{a}_r^{(k)}\phi(\hat{\mathbf{w}}_r^{(k),T}\mathbf{x}), \quad \nabla_{f_d}^{(k)}\ell(\mathbf{x}, y) = \frac{-y}{1 + \exp(f_d^{(k)}(\mathbf{x})y)} \quad \forall k \geq 0. \tag{107}$$

Similar to Appendix C.2, we have to introduce a weight-space measure in order to take a limit of $d \to \infty$:

$$\mu_d^{(k)} = \frac{1}{d}\sum_{r=1}^d \delta_{\tilde{a}_r^{(k)}} \otimes \delta_{\hat{\mathbf{w}}_r^{(k)}}. \tag{108}$$

In terms of the measure the dynamics is expressed then as follows:

$$\mu_d^{(k+1)} = \mu_d^{(k)} + \operatorname{div}(\mu_d^{(k)}\Delta\theta_d^{(k)}), \tag{109}$$

$$\mu_d^{(0)} = \frac{1}{d} \sum_{r=1}^{d} \delta_{\tilde{a}_r^{(0)}} \otimes \delta_{\hat{\mathbf{w}}_r^{(0)}}, \quad \tilde{a}_r^{(0)} \sim \mathcal{N}(0, (d/d^*)^{-1}), \quad \hat{\mathbf{w}}_r^{(0)} \sim \mathcal{N}(0, I_{d_{\mathbf{x}}}) \quad \forall r \in [d], \quad (110)$$

$$\Delta\theta_d^{(k)}(\tilde{a}, \hat{\mathbf{w}}) =$$
$$= -[\hat{\eta}_a^* \sigma^* \nabla_{f_d}^{(k)} \ell(\mathbf{x}_a^{(k)}, y_a^{(k)}) \phi(\hat{\mathbf{w}}^T \mathbf{x}_a^{(k)}), \ \hat{\eta}_w^* \sigma^* \nabla_{f_d}^{(k)} \ell(\mathbf{x}_w^{(k)}, y_w^{(k)}) \tilde{a} \phi'(\hat{\mathbf{w}}^T \mathbf{x}_w^{(k)}) \mathbf{x}_w^{(k),T}]^T, \quad (111)$$

$$f_d^{(k)}(\mathbf{x}) = \sigma^* d \int \tilde{a} \phi(\hat{\mathbf{w}}^T \mathbf{x}) \, \mu_d^{(k)}(d\tilde{a}, d\hat{\mathbf{w}}), \quad \nabla_{f_d}^{(k)} \ell(\mathbf{x}, y) = \frac{-y}{1 + \exp(f_d^{(k)}(\mathbf{x})y)} \quad \forall k \geq 0. \quad (112)$$

We rewrite the last equation in terms of $\tilde{f}_d^{(k)}(\mathbf{x}) = d^{-1} f_d^{(k)}(\mathbf{x})$:

$$\tilde{f}_d^{(k)}(\mathbf{x}) = \sigma^* \int \hat{a} \phi(\hat{\mathbf{w}}^T \mathbf{x}) \, \mu_d^{(k)}(d\hat{a}, d\hat{\mathbf{w}}), \quad \nabla_{\tilde{f}_d}^{(k)} \ell(\mathbf{x}, y) = \frac{-y}{1 + \exp(d\tilde{f}_d^{(k)}(\mathbf{x})y)} \quad \forall k \geq 0. \quad (113)$$

A limit dynamics then takes the following form:

$$\mu_\infty^{(k+1)} = \mu_\infty^{(k)} + \mathrm{div}(\mu_\infty^{(k)} \Delta\theta_d^{(k)}), \quad \mu_\infty^{(0)} = \delta \otimes \mathcal{N}(0, I_{d_{\mathbf{x}}}) \quad (114)$$

$$\Delta\theta_\infty^{(k)}(\tilde{a}, \hat{\mathbf{w}}) =$$
$$= -[\hat{\eta}_a^* \sigma^* \nabla_{f_\infty}^{(k)} \ell(\mathbf{x}_a^{(k)}, y_a^{(k)}) \phi(\hat{\mathbf{w}}^T \mathbf{x}_a^{(k)}), \ \hat{\eta}_w^* \sigma^* \nabla_{f_\infty}^{(k)} \ell(\mathbf{x}_w^{(k)}, y_w^{(k)}) \tilde{a} \phi'(\hat{\mathbf{w}}^T \mathbf{x}_w^{(k)}) \mathbf{x}_w^{(k),T}]^T, \quad (115)$$

$$\nabla_{f_\infty}^{(0)} \ell(\mathbf{x}, y) = \frac{-y}{1 + \exp(\sigma^* d^{*,1/2} \mathcal{N}(0, \sigma^{(0),2}(\mathbf{x}))y)}, \quad (116)$$

$$\tilde{f}_\infty^{(k)}(\mathbf{x}) = \sigma^* \int \tilde{a} \phi(\hat{\mathbf{w}}^T \mathbf{x}) \, \mu_\infty^{(k)}(d\tilde{a}, d\hat{\mathbf{w}}), \quad \nabla_{f_\infty}^{(k+1)} \ell(\mathbf{x}, y) = -y[\tilde{f}_\infty^{(k+1)}(\mathbf{x})y < 0] \quad \forall k \geq 0. \quad (117)$$

As one can notice, the only difference between this limit dynamics and the limit dynamics of sym-default scaling (Appendix C.2.2) is the initial measure.

We now check the Condition 5. First of all, by the Central Limit Theorem, $f_d^{(0)}(\mathbf{x}) = \Theta_{d \to \infty}(1)$, hence the first point of Condition 5 holds. As for kernels, we have:

$$K_{a,d}^{(k)}(\mathbf{x}, \mathbf{x}') = \sigma^{*,2} \sum_{r=1}^{d} \phi(\hat{\mathbf{w}}_r^{(k),T} \mathbf{x}) \phi(\hat{\mathbf{w}}_r^{(k),T} \mathbf{x}'), \quad (118)$$

$$K_{w,d}^{(k)}(\mathbf{x}, \mathbf{x}') = \sigma^{*,2} (d/d^*)^{-1} \sum_{r=1}^{d} |\hat{a}_r^{(k)}|^2 \phi'(\hat{\mathbf{w}}_r^{(k),T} \mathbf{x}) \phi'(\hat{\mathbf{w}}_r^{(k),T} \mathbf{x}') \mathbf{x}^T \mathbf{x}'. \quad (119)$$

We see that while $K_{w,d}^{(0)}$ converges to a constant due to the Law of Large Numbers, $K_{a,d}^{(0)}$ diverges as $d \to \infty$. This violates the second statement of Condition 5, and the third as well, since $f_\infty^{(0)}$ is finite. Consider now kernel increments:

$$\Delta K_{aw,d}^{(k),'}(\mathbf{x}, \mathbf{x}') = -\sigma^{*,3} (d/d^*)^{-1/2} \sum_{r=1}^{d} \Big( \phi(\hat{\mathbf{w}}_r^{(k),T} \mathbf{x}) \phi'(\hat{\mathbf{w}}_r^{(k),T} \mathbf{x}') +$$
$$+ \phi'(\hat{\mathbf{w}}_r^{(k),T} \mathbf{x}) \phi(\hat{\mathbf{w}}_r^{(k),T} \mathbf{x}') \Big) \times$$
$$\times \nabla_{f_d}^{(k)} \ell(\mathbf{x}_w^{(k)}, y_w^{(k)}) \hat{a}_r^{(k)} \phi'(\hat{\mathbf{w}}_r^{(k),T} \mathbf{x}_w^{(k)}) (\mathbf{x} + \mathbf{x}')^T \mathbf{x}_w^{(k)}, \quad (120)$$

$$\Delta K_{ww,d}^{(k),'}(\mathbf{x},\mathbf{x}') = -\sigma^{*,3}(d/d^*)^{-3/2} \sum_{r=1}^{d} |\hat{a}_r^{(k)}|^2 \Big( \phi'(\hat{\mathbf{w}}_r^{(k),T}\mathbf{x})\phi''(\hat{\mathbf{w}}_r^{(k),T}\mathbf{x}') +$$
$$+ \phi''(\hat{\mathbf{w}}_r^{(k),T}\mathbf{x})\phi'(\hat{\mathbf{w}}_r^{(k),T}\mathbf{x}') \Big) \mathbf{x}^T\mathbf{x}' \times$$
$$\times \nabla_{f_d}^{(k)}\ell(\mathbf{x}_w^{(k)}, y_w^{(k)})\hat{a}_r^{(k)}\phi'(\hat{\mathbf{w}}_r^{(k),T}\mathbf{x}_w^{(k)})(\mathbf{x}+\mathbf{x}')^T\mathbf{x}_w^{(k)}, \quad (121)$$

$$\Delta K_{wa,d}^{(k),'}(\mathbf{x},\mathbf{x}') = -\sigma^{*,3}(d/d^*)^{-1/2} \sum_{r=1}^{d} 2\hat{a}_r^{(k)}\phi'(\hat{\mathbf{w}}_r^{(k),T}\mathbf{x})\phi'(\hat{\mathbf{w}}_r^{(k),T}\mathbf{x}') \times$$
$$\times \nabla_{f_d}^{(k)}\ell(\mathbf{x}_a^{(k)}, y_a^{(k)})\phi(\hat{\mathbf{w}}_r^{(k),T}\mathbf{x}_a^{(k)}). \quad (122)$$

For $k = 0$ terms inside sums of each increment have zero expectations. Hence the Central Limit Theorem can be used here. We get: $\Delta K_{aw,d}^{(0),'} = \Theta_{d\to\infty}(1)$, $\Delta K_{ww,d}^{(0),'} = \Theta_{d\to\infty}(d^{-1})$, $\Delta K_{wa,d}^{(0),'} = \Theta_{d\to\infty}(1)$. Since $K_{a,d}^{(0)} = \Theta_{d\to\infty}(d)$, $K_{w,d}^{(0)} = \Theta_{d\to\infty}(1)$, the last statement of Condition 5 is violated as well.

## E INITIALIZATION-CORRECTED MEAN-FIELD (IC-MF) LIMIT

Here we consider the same training dynamics as for the mean-field scaling (see Appendix C.2), but with a modified model definition:

$$\Delta\hat{a}_r^{(k)} = -\hat{\eta}_a^*\sigma^*\nabla_{f_d}^{(k)}\ell(\mathbf{x}_a^{(k)}, y_a^{(k)})\,\phi(\hat{\mathbf{w}}_r^{(k),T}\mathbf{x}_a^{(k)}), \quad \hat{a}_r^{(0)} \sim \mathcal{N}(0,1), \quad (123)$$

$$\Delta\hat{\mathbf{w}}_r^{(k)} = -\hat{\eta}_w^*\sigma^*\nabla_{f_d}^{(k)}\ell(\mathbf{x}_w^{(k)}, y_w^{(k)})\,\hat{a}_r^{(k)}\phi'(\hat{\mathbf{w}}_r^{(k),T}\mathbf{x}_w^{(k)})\mathbf{x}_w^{(k)}, \quad \hat{\mathbf{w}}_r^{(0)} \sim \mathcal{N}(0, I_{d_\mathbf{x}}). \quad (124)$$

$$f_d^{(k)}(\mathbf{x}) = \sigma^*(d/d^*)^{-1}\sum_{r=1}^{d}\hat{a}_r^{(k)}\phi(\hat{\mathbf{w}}_r^{(k),T}\mathbf{x}) + \sigma^*(d/d^*)^{-1/2}\sum_{r=1}^{d}\hat{a}_r^{(0)}\phi(\hat{\mathbf{w}}_r^{(0),T}\mathbf{x}), \quad (125)$$

$$\nabla_{f_d}^{(k)}\ell(\mathbf{x}, y) = \frac{-y}{1+\exp(f_d^{(k)}(\mathbf{x})y)} \quad \forall k \geq 0. \quad (126)$$

Similar to the mean-field case (Appendix C.2), we rewrite the dynamics above in terms of the weight-space measure:

$$\mu_d^{(k+1)} = \mu_d^{(k)} + \mathrm{div}(\mu_d^{(k)}\Delta\theta_d^{(k)}), \quad \mu_d^{(0)} = \frac{1}{d}\sum_{r=1}^{d}\delta_{\hat{\theta}_r^{(0)}}, \quad \hat{\theta}_r^{(0)} \sim \mathcal{N}(0, I_{1+d_\mathbf{x}}) \quad \forall r \in [d], \quad (127)$$

$$\Delta\theta_d^{(k)}(\hat{a}, \hat{\mathbf{w}}) =$$
$$= -[\hat{\eta}_a^*\sigma^*\nabla_{f_d}^{(k)}\ell(\mathbf{x}_a^{(k)}, y_a^{(k)})\phi(\hat{\mathbf{w}}^T\mathbf{x}_a^{(k)}), \ \hat{\eta}_w^*\sigma^*\nabla_{f_d}^{(k)}\ell(\mathbf{x}_w^{(k)}, y_w^{(k)})\hat{a}\phi'(\hat{\mathbf{w}}^T\mathbf{x}_w^{(k)})\mathbf{x}_w^{(k),T}]^T, \quad (128)$$

$$f_d^{(k)}(\mathbf{x}) = \sigma^*d^*\int \hat{a}\phi(\hat{\mathbf{w}}^T\mathbf{x})\,\mu_d^{(k)}(d\hat{a}, d\hat{\mathbf{w}}) + \sigma^*(dd^*)^{1/2}\int \hat{a}\phi(\hat{\mathbf{w}}^T\mathbf{x})\,\mu_d^{(0)}(d\hat{a}, d\hat{\mathbf{w}}), \quad (129)$$

$$\nabla_{f_d}^{(k)}\ell(\mathbf{x}, y) = \frac{-y}{1+\exp(f_d^{(k)}(\mathbf{x})y)} \quad \forall k \geq 0. \quad (130)$$

Note that here $f_d^{(k)}$ stays finite in the limit of $d \to \infty$ for any $k \geq 0$. Hence taking the limit $d \to \infty$ yields:

$$\mu_\infty^{(k+1)} = \mu_\infty^{(k)} + \mathrm{div}(\mu_\infty^{(k)}\Delta\theta_\infty^{(k)}), \quad \mu_\infty^{(0)} = \mathcal{N}(0, I_{1+d_\mathbf{x}}), \quad (131)$$

$$\Delta\theta_\infty^{(k)}(\hat{a}, \hat{\mathbf{w}}) =$$
$$= -[\hat{\eta}_a^*\sigma^*\nabla_{f_\infty}^{(k)}\ell(\mathbf{x}_a^{(k)}, y_a^{(k)})\phi(\hat{\mathbf{w}}^T\mathbf{x}_a^{(k)}), \ \hat{\eta}_w^*\sigma^*\nabla_{f_\infty}^{(k)}\ell(\mathbf{x}_w^{(k)}, y_w^{(k)})\hat{a}\phi'(\hat{\mathbf{w}}^T\mathbf{x}_w^{(k)})\mathbf{x}_w^{(k),T}]^T, \quad (132)$$

$$f_\infty^{(k)}(\mathbf{x}) = \sigma^* d^* \int \hat{a}\phi(\hat{\mathbf{w}}^T\mathbf{x})\, \mu_\infty^{(k)}(d\hat{a}, d\hat{\mathbf{w}}) + \sigma^* d^{*,1/2}\mathcal{N}(0, \sigma^{(0),2}(\mathbf{x})), \qquad (133)$$

$$\nabla_{f_\infty^{(k)}}\ell(\mathbf{x}, y) = \frac{-y}{1 + \exp(f_\infty^{(k)}(\mathbf{x})y)} \quad \forall k \geq 0. \qquad (134)$$

## F  EXPERIMENTAL DETAILS

We perform our experiments on a feed-forward fully-connected network with a single hidden layer with no biases. We learn our network as a binary classifier on a subset of the CIFAR2 dataset (which is a dataset of first two classes of CIFAR10[1]) of size 1024. We report results using a test set from the same dataset of size 2000. We do not do a hyperparameter search, for this reason, we do not use a validation set.

We train our network for 2000 training steps to minimize the binary cross-entropy loss. We use a full-batch GD as an optimization algorithm. We repeat our experiments for 10 random seeds and report mean and deviations in plots for logits and kernels (e.g. Figure 1, left). For plots of the KL-divergence, we use logits from these 10 random seeds to fit a single gaussian. Where necessary, we estimate data expectations (e.g. $\mathbb{E}_{\mathbf{x}\sim\mathcal{D}}|f(\mathbf{x})|$) using 10 samples from the test dataset.

We experiment with other setups (i.e. using a mini-batch gradient estimation instead of exact one, a larger train dataset, a multi-class classification) in Appendix G. All experiments were conducted on a single NVIDIA GeForce GTX 1080 Ti GPU using the PyTorch framework (Paszke et al., 2017). Our code is available online: ⟨*suppressed for anonymity*⟩.

Although our analysis assumes initializing variables with samples from a gaussian, nothing changes if we sample $\sigma\xi$ instead, where $\xi$ can be any symmetric random variable with a distribution independent on hyperparameters.

In our experiments, we took a network of width $d^* = 2^7 = 128$ with leaky ReLU activation and apply the Kaiming He uniform initialization (He et al., 2015) to its layers; we call this network a reference network. According to the Kaiming He initialization strategy, initial weights have a zero mean and a standard deviation $\sigma^* \propto (d^*)^{-1/2}$ for the output layer, while the standard deviation of the input layer does not depend on the reference width $d^*$. For this network we take learning rates in the original parameterization $\eta_a^* = \eta_w^* = 0.02$. After that, we scale its initial weights and learning rates with width $d$ according to a scaling at hand:

$$\sigma = \sigma^*\left(\frac{d}{d^*}\right)^{q_\sigma}, \quad \hat{\eta}_{a\vee w} = \hat{\eta}_{a\vee w}^*\left(\frac{d}{d^*}\right)^{\tilde{q}_{a\vee w}}.$$

Note that we have assumed $\sigma_w = 1$. By definition, $\hat{\eta}_{a\vee w} = \eta_{a\vee w}/\sigma_{a\vee w}^2$; this implies:

$$\eta_a = \eta_a^*\left(\frac{\sigma}{\sigma^*}\right)^2\left(\frac{d}{d^*}\right)^{\tilde{q}_a} = \eta_a^*\left(\frac{d}{d^*}\right)^{\tilde{q}_a+2q_\sigma}, \quad \eta_w = \eta_w^*\left(\frac{d}{d^*}\right)^{\tilde{q}_w}.$$

## G  EXPERIMENTS FOR OTHER SETUPS

Although plots provided in the main body represent the full-batch GD on a subset of CIFAR2, we have experimented with other setups as well. For instance, we have varied the batch size and the size of the train dataset. Results are shown in Figures 3-7. Here differences are marginal and not qualitative.

We have also experimented with multi-class classification: see Figure 8. Here we have trained our network on the full CIFAR10 dataset with SGD with batches of size 100. As we see on left plot, the IC-MF limit model has the lowest KL-divergence relative to the reference model, however, in terms of the test accuracy, all the limit models are similar.

---

[1]CIFAR10 can be downloaded at `https://www.cs.toronto.edu/~kriz/cifar.html`

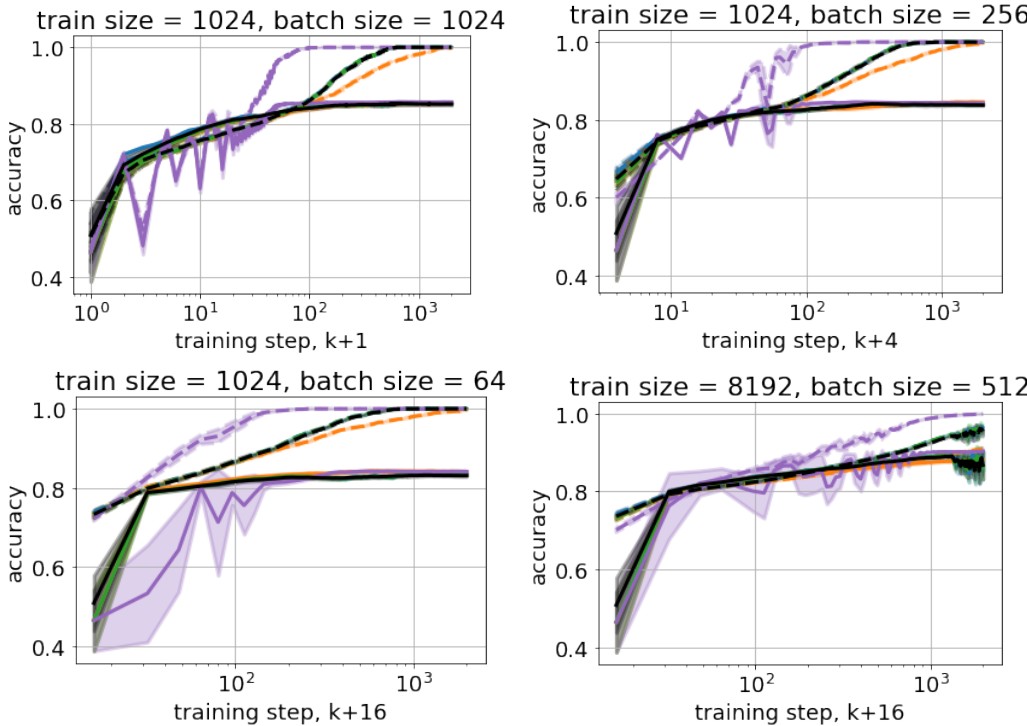

Figure 3: Test accuracy of different limit models, as well as of the reference model. *Setup:* We train a one hidden layer network on subsets of the CIFAR2 dataset of different sizes with SGD with varying batch sizes.

## H  GENERALIZATION TO DEEP NETS PROPOSAL

While our present analysis is devoted to networks with a single hidden layer, we discuss possible generalizations to deep nets here.

Consider a network with $H$ hidden layers. For simplicity, assume that widths of all hidden layers are equal to $d$. We thus have to consider $H+1$ learning rates $\tilde{q}_{0:H}$, one for each layer, and similarly $H+1$ initialization variances $\sigma_{0:H}$. Without loss of generality, we may assume the input layer variance to be equal to $1$ (we can rescale inputs otherwise). This gives $2H+1$ hyperparameters in total.

Similarly to what we did for $H=1$, we assume that each hyperparameter obeys a power-law with respect to width. Let us refer the set of the power-law exponents as a "scaling". Again, we want to reason about what the scaling should be in order to converge to a dynamically stable limit model: see Condition 1. Moreover, we want to derive conditions that separate the domain of "dynamically stable" scalings, such that each region corresponds to a distinct unique dynamically stable limit model: see Condition 2.

Having that much hyperparameters seems burdening, and this prohibits us to draw a nice two-dimensional scaling plane as we did for $H=1$: see Figure 1. For this reason, one have to reduce the dimensionality of a scaling.

First, it is tempting to consider a homogeneous activation function: a leaky ReLU. This introduces a symmetry in the weight space that guarantees that dynamics depends only on the product of initialization variances: $\sigma_H \times \ldots \times \sigma_0$; let us refer this product as $\sigma$. This approach was previously used by Golikov (2020), however we have to note that non-smoothness of the activation function introduces certain mathematical obstacles. Nevertheless, one may consider sacrificing mathematical rigor in favor of reducing the number of hyperparameters from $2H+1$ to $H+1$.

The next simplification should affect learning rate scaling exponents. Similar to what we have done for a shallow net, we may assume all learning rate exponents to be equal: $\tilde{q}_0 = \ldots = \tilde{q}_H = \tilde{q}$.

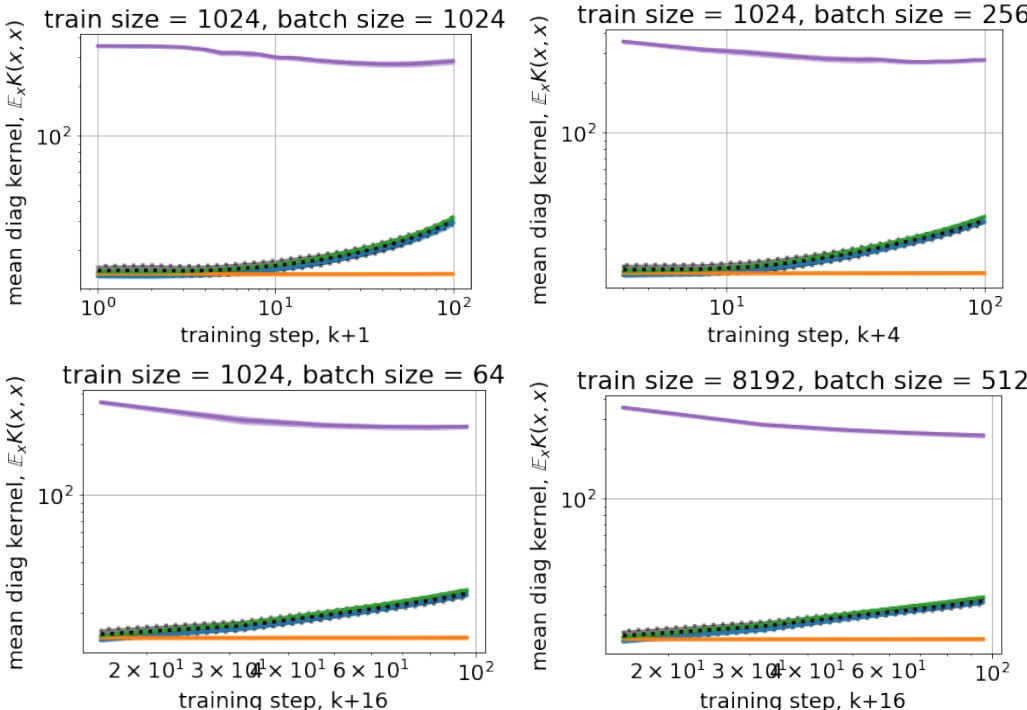

Figure 4: Mean kernel diagonals $\mathbb{E}_{\mathbf{x} \sim \mathcal{D}}(\hat{\eta}_a^* K_{a,d}(\mathbf{x}, \mathbf{x}) + \hat{\eta}_w^* K_{w,d}(\mathbf{x}, \mathbf{x}))$ of different limit models, as well as of the reference model. *Setup:* We train a one hidden layer network on subsets of the CIFAR2 dataset of different sizes with SGD with varying batch sizes. Data expectations are estimated with 10 test data samples.

The NTK limit, which generalizes naturally to deep nets, requires $\tilde{q}_0 = \ldots = \tilde{q}_H = 0$, and hence conforms the assumption above. However, a possible generalization of the mean-field limit requires $\tilde{q}_0 = \tilde{q}_H = 1$, while $\tilde{q}_1 = \ldots = \tilde{q}_{H-1} = 2$; see Sirignano & Spiliopoulos (2019); Araújo et al. (2019); Golikov (2020). This aspect suggests the following alternatives:

1. Consider $\tilde{q}_0 = \tilde{q}_H = \tilde{q}$, while $\tilde{q}_1 = \ldots = \tilde{q}_{H-1} = \tilde{q}_{hid}$; this results in a three-dimensional space of scalings: $(q_\sigma, \tilde{q}, \tilde{q}_{hid})$.

2. Consider $\tilde{q}_0 = \tilde{q}_H = \tilde{q}$, while $\tilde{q}_1 = \ldots = \tilde{q}_{H-1} = 2\tilde{q}$; this results in a two-dimensional space of scalings that covers both of the NTK and the mean-field scalings.

The former class of scalings is richer, but if it does not contain any interesting limit models that are present in the second class, it can be more expository to tighten the class to the latter. By "interesting" we mean limit models that are "non-dominated" in a similar sense as we have specified in Section 3.

In order to define which limit models are better than others in approximating finite-width nets ("non-dominated"), we have to derive conditions that separate the domain of dynamically stable scalings into regions of distinct unique corresponding limit models, similar to Condition 2. We hypothesize that these conditions are similar to the shallow case: (1) a limit model at initialization is finite, (2) kernels at initialization are finite, (3) a limit model and kernels are of the same order, (4) kernels evolve at initialization. Since we have decided to consider separate learning rate scalings for hidden layers and for input and output layers, we expect that the above-proposed conditions should consider two distinct families of kernels respectively: hidden kernels and input plus output kernels.

It will be very interesting to check if all of the dynamically stable limit models are specified either by an evolution in a model space driven by constant kernel, or by an evolution of a weight-space measure, as was the case for $H = 1$; see Appendix C. Investigating a non-dominated limit model, different from both the NTK and the mean-field models, should be a valuable outcome of the proposed research

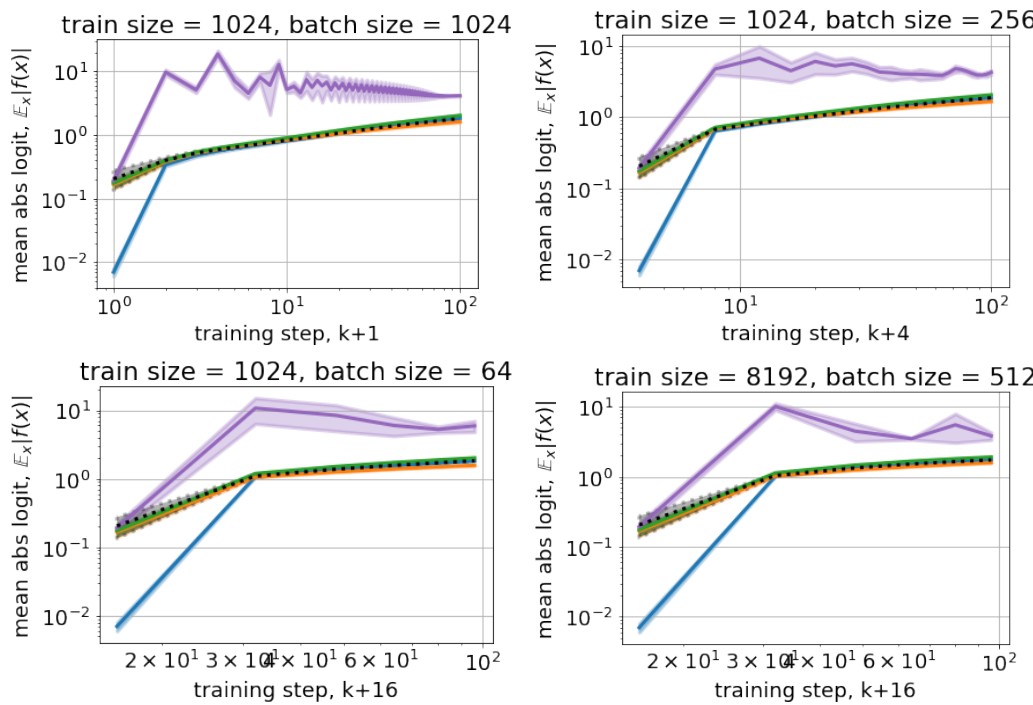

Figure 5: Mean absolute logits $\mathbb{E}_{\mathbf{x} \sim \mathcal{D}} |f(\mathbf{x})|$ of different limit models, as well as of the reference model. *Setup:* We train a one hidden layer network on subsets of the CIFAR2 dataset of different sizes with SGD with varying batch sizes. Data expectations are estimated with 10 test data samples.

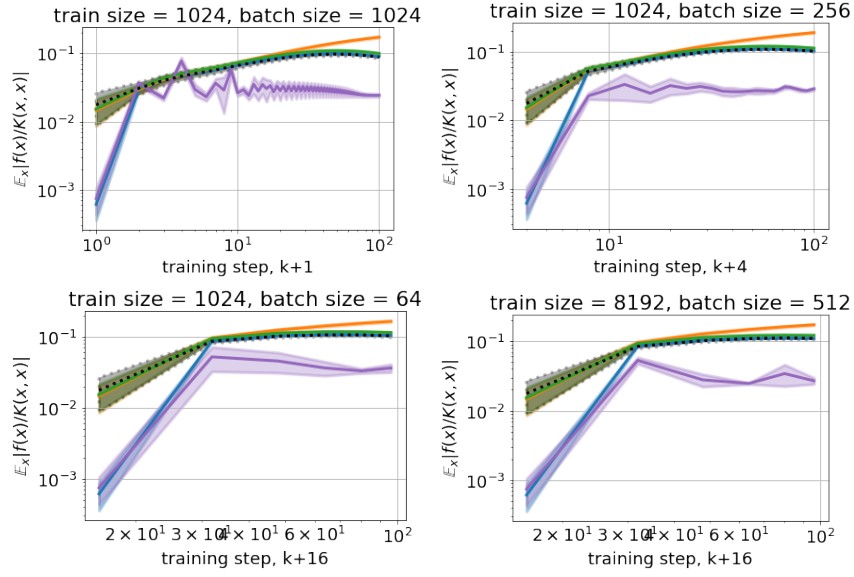

Figure 6: Mean absolute logits relative to kernel diagonals $\mathbb{E}_{\mathbf{x} \sim \mathcal{D}} |f_d(\mathbf{x})/(\hat{\eta}_a^* K_{a,d}(\mathbf{x}, \mathbf{x}) + \hat{\eta}_w^* K_{w,d}(\mathbf{x}, \mathbf{x}))|$ of different limit models, as well as of the reference model. *Setup:* We train a one hidden layer network on subsets of the CIFAR2 dataset of different sizes with SGD with varying batch sizes. Data expectations are estimated with 10 test data samples.

program; it will be even more valuable if this limit model will not be covered by both mean-field and constant kernel formalisms.

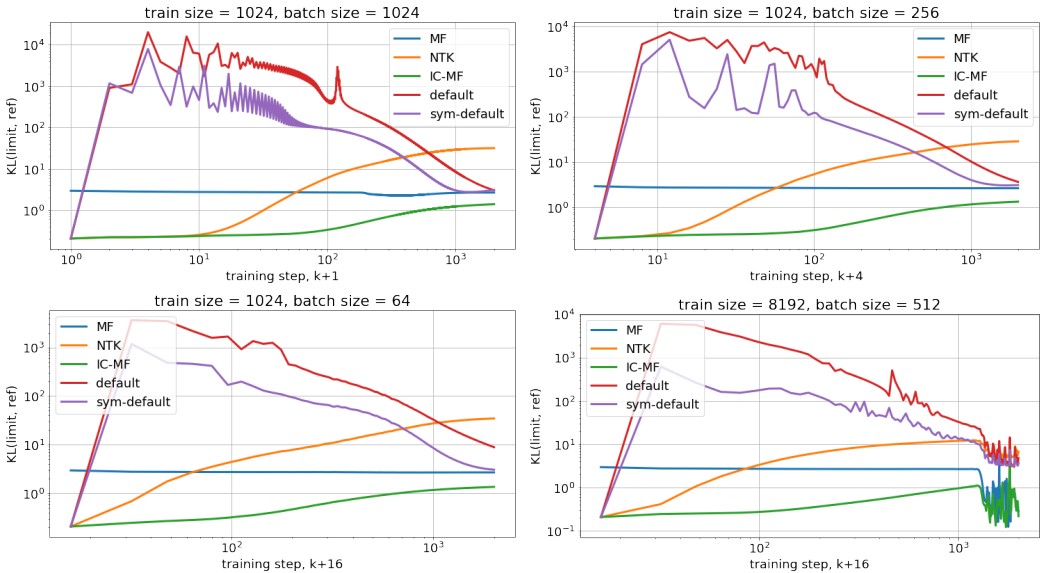

Figure 7: KL-divergence of different limit models relative to a reference model. *Setup:* We train a one hidden layer network on subsets of the CIFAR2 dataset of different sizes with SGD with varying batch sizes.

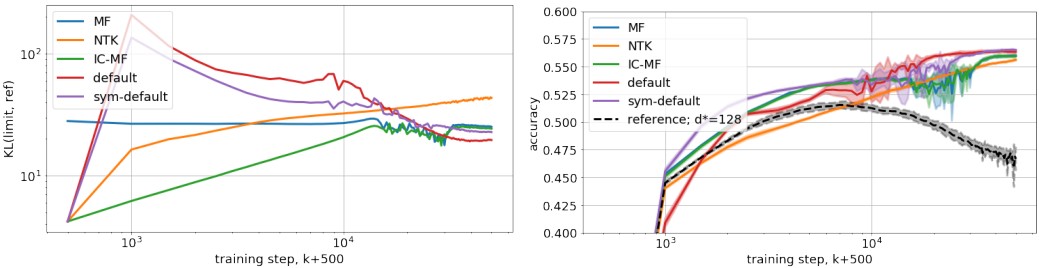

Figure 8: *Left:* KL-divergence of different limit models relative to a reference model. *Right:* Accuracies on the test set of different limit models as well as of the reference model. *Setup:* We train a one hidden layer network on the full CIFAR10 dataset with SGD with batches of size 100.

We also have to note that according to Golikov (2020), the mean-field limit vanishes for $H > 2$. This fact suggests that the analysis for deep nets should be held for $H = 2$ and for $H > 2$ separately.

## I   MEASURING DIVERGENCE BETWEEN A LIMIT MODEL AND A REFERENCE ONE

We track the divergence of a limiting network from a reference one, which can be done in two ways. The first one is tracking divergence directly between logits: $\mathbb{E}_{\mathbf{x} \sim \mathcal{D}_{test}} D_{logits}(f^{(k)}_{\infty}(\mathbf{x}) \,||\, f^{(k)}_{d^*}(\mathbf{x}))$ for some divergence measure $D(\cdot \,||\, \cdot)$. The second one is tracking divergence between probabilities: $\mathbb{E}_{\mathbf{x} \sim \mathcal{D}_{test}} D_{prob}(\sigma(f^{(k)}_{\infty}(\mathbf{x})) \,||\, \sigma(f^{(k)}_{d^*}(\mathbf{x})))$, where we have overloaded the notation by denoting the standard sigmoid as $\sigma$: $\sigma(x) = (1 + \exp(-x))^{-1}$.

We choose a KL-divergence for the first case. However, measuring a KL-divergence between logits is hardly possible, since we do not have an access to the distribution of $f^{(k)}(\mathbf{x})$ as a random variable depending on initialization. For this reason, we fit a gaussian to its samples:

$$D_{logits}(\xi \,||\, \xi^*) = \mathrm{KL}(\mathcal{N}(\mathbb{E}\,\xi, \mathbb{V}\mathrm{ar}\,\xi) \,||\, \mathcal{N}(\mathbb{E}\,\xi^*, \mathbb{V}\mathrm{ar}\,\xi^*)). \tag{135}$$

This case is depicted in Figure 9, left.

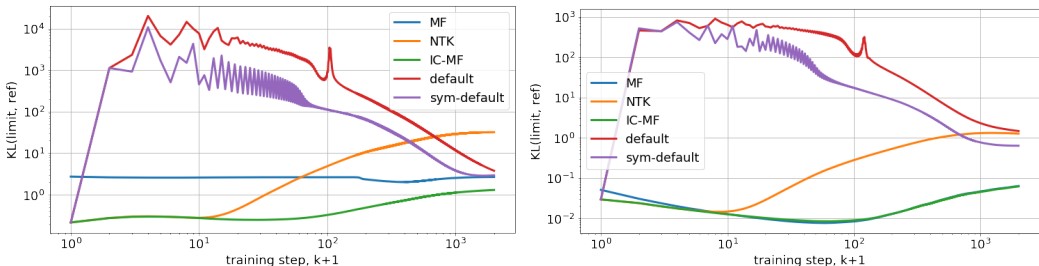

Figure 9: *Left:* we plot a KL-divergence of logits of different infinite-width limits of a fixed finite-width reference model relative to logits of this reference model. KL-divergences are estimated using gaussian fits with 10 samples. *Right:* same, for probabilities instead of logits. KL-divergences are estimated using beta-distribution fits with 10 samples. *Setup:* we train a one hidden layer network with SGD on CIFAR2 dataset; see Appendix F for details.

As for the second case, we may want to measure a KL-divergence between distributions on probabilities. Again, this is not possible, because the true distribution of $\sigma(f_\infty^{(k)}(\mathbf{x}))$ is not known; for this reason, we decide to first fit a beta distribution, and then measure the divergence:

$$D_{prob}(\xi \, || \, \xi^*) = \text{KL}(\text{Beta}(\alpha_{mle}(\xi), \beta_{mle}(\xi)) \, || \, \text{Beta}(\alpha_{mle}(\xi^*), \beta_{mle}(\xi^*))), \qquad (136)$$

where $\alpha_{mle}(\xi)$ and $\beta_{mle}(\xi)$ are maximum-likelihood estimations of hyperparameters of a beta:

$$\alpha_{mle}(\xi) = \mathbb{E}\, \xi \left( \frac{\mathbb{E}\, \xi(1 - \mathbb{E}\, \xi)}{\mathbb{V}\text{ar}\, \xi} - 1 \right), \qquad \beta_{mle}(\xi) = (1 - \mathbb{E}\, \xi) \left( \frac{\mathbb{E}\, \xi(1 - \mathbb{E}\, \xi)}{\mathbb{V}\text{ar}\, \xi} - 1 \right). \qquad (137)$$

This case is plotted in Figure 9, right.

Both cases can be generalized to work with multi-class classification. In the first case, we can simply fit a gaussian with a diagonal covariance matrix; this was done in Figure 8, left. In the second case, we can fit a Dirichlet random variable using a maximum-likelihood estimation as before.

