# OpenReview forum: "Dynamically Stable Infinite-Width Limits of Neural Classifiers"
_ICLR.cc/2021/Conference — Reject_

### Official Review · AnonReviewer2 · 2020-10-27
**Interesting proposals, but the paper requires substantial revisions and additional analyses.**

**Rating:** 3
**Confidence:** 5

**Review:**

The paper analyzes joint scalings of the parameter initialization and the learning rate, with respect to the limit of infinite width, in the context of two-layer neural networks with stochastic gradient descent and binary logistic loss. It proposes some “dynamically stable” conditions and identifies a range of scalings that satisfy these conditions. This covers the neural tangent kernel (NTK) and mean field (MF) scalings, as well as others. The paper then proposes to add an extra correction function to the initialization of the MF limit and argues experimentally that this correction can give a proxy for standard neural networks.

Overall there are interesting points in the paper, but I think the paper needs a lot more works to make the study rigorous, complete and easier to read. More analyses into why any point in the “dynamically stable model evolution” band leads to stable dynamics at all training time are needed.


### Positive points:
The recent trend in analysis of large-width neural networks has drawn a lot of attention, so the paper is timely.

The idea of joint scalings between the initialization and the learning rate, w.r.t. the width d, has been explored before; but here the paper proposes to do so in relation with a new set of criteria. The results make an interesting view on the connection between two distinct regimes NTK and MF, and beyond.


### Negative points:

**Presentation:**
There are many unnecessary notations, for example, \tilde{q}_a and \tilde{q}_w which are equal to each other in the main analysis. It is better to keep all discussions around just the two key scaling exponents.

The main figure, Figure 1, is very hard to read, and I cannot see where the curve for MF is in this plot.

The statements also seem to miss several assumptions e.g. assumptions on the data (will things hold if |x| ~ exp(d)??).

Some notations are not explained, for example, \sigma^{(0)}(x) that appears in Eq (13).

**Incomplete proofs:**
Aside from many derivations which are heuristic (which are fine as long as they are not stated as propositions/theorems), the proofs for the stated lemmas / propositions are unrigorous and may entirely omit the more difficult technical points. For example, in the proof of Lemma 1.4 in Appendix B.1, why is it that the collection of the weights \hat{w}_r^{(k), r=1…,d, at any time k>1 allows one to apply the law of large numbers? The variables in this collection are not independent; if they are mildly dependent, in what sense are they so that one can apply the law of large numbers?

In fact, there are simple mathematical mistakes throughout the proofs. For example, to prove Lemma 1.2 in Appendix B.1, the paper only proves the statement for a single fixed x; why does it hold for almost every x?

**Abstract missing key information:**
The abstract should be more precise: it should mention logistic loss for binary classification and that the initializations are zero-mean. These are important; without them, a number of key results will fail.

**Dynamical stability criteria are insufficiently justified; most are limited to just initializations:**
The paper does not give sufficient justification of why the proposed criteria (Conditions 1 and 2) should be considered. In particular, Condition 2 only concerns with what happens at initialization; does it guarantee stable dynamics well beyond initialization and in what sense? The paper does not discuss this for most points in the “dynamically stable model evolution” band. In fact, compared to the well-studied NTK and MF scalings, the newly identified scaling sym-default is said to have some divergence behavior (in which its prediction function is always infinite, for infinite d, at any positive training time). If anything, the paper should justify why this divergence behavior does not lead to instability; by looking at Figure 1, I think it suffers from numerical instability.

In fact, scalings other than NTK and MF have been studied well beyond initialization in a rigorous fashion. In particular, [1] shows that with a wide range of different scalings, it’s entirely possible to have the same stable behavior: NTK behavior where weights do not move and the model tracks some random-feature models. Given that such rigorous and thorough study is possible, it is unclear how the proposal of Conditions 1 and 2 leads to novel and significant insights.

**Condition 2.3 is unusual in terms of physical units and not motivated:**
In particular, why should we compare the magnitude of the kernel K with the magnitude of the prediction function f, while they should have very different physical interpretation? Why is this condition interesting? The second paragraph in Section 3 seems to give some arguments for Condition 2.1 and 2.2, but none for Condition 2.3.

Given that the role of Condition 2.3 is unclear, if one removes this condition from the diagram of Figure 1, the band picture disappears and we are effectively left with a half plane, separated by the “evolving kernels” boundary. To the left of this boundary, it is the NTK behavior as expected in [1]. So unless Condition 2.3 signifies some significant and meaningful change in network behaviors, the diagram would not show new significant insights compared to the literature.

**The derivations for the dynamics in Section 3 are incomplete:**
In the regime of small learning rate (\hat{\eta}* goes to 0), what we should obtain is a continuous time evolution with an expectation over data, not a discrete-time one with stochastically drawn data. Evidently Eq (14) as given in the paper completely contradicts all previous published reports on the MF limit.

It does not make sense to claim, after Equation (19), that the prediction function diverges from iteration k=1 onwards. Firstly, as said, the analysis has to be done with respect to small learning rate and hence the iteration k has to be determined in relation with the learning rate. Secondly, that claim contradicts Figure 1: the prediction function does not seem to diverge at iteration 1.

**The initialization-corrected mean field (IC-MF) limit lacks justification:**
Its aim is to correct the MF limit w.r.t. Condition 2.1, but there are very simple alternatives to do this correction. The first way is to add any fixed function, whose magnitude are independent of d in suitable sense, to the MF limit. The second way is to initialize the MF limit with non-zero mean distributions; in fact, it is known that non-zero mean initializations are more typical in MF limit.

The paper argues that IC-MF limit can be good proxy for standard networks, giving only one simple experiment shown in Figure 1. There is no proof or mathematical heuristics provided. I think this IC-MF idea does not have anything to do with the theory in the previous sections, nor the binary classification problem with logistic loss. As such, it should be at least further tested on more complex experimental tasks, e.g. CIFAR-10. From a mathematical standpoint, I do not foresee a simple argument to show why it can be a good proxy for standard networks.


[1] A comparative analysis of the optimization and generalization property of two-layer neural network and random feature models under gradient descent dynamics, E, Ma and Wu, 2019.

---------------------

Post rebuttal:

I thank the authors for their rebuttal. Let me focus my reply on a few important points. I first thank the authors for clarifying the meaning of Condition 2. In this sense, Condition 1 is the key main contribution; however the current proof does not look correct to me, and the revised argument is far from being sufficient. In particular:

- Point 5 of the rebuttal: I think the revised argument here is incomplete. The given argument concerns trivial facts and does not imply the claim. For example, what if the distribution of the terms is symmetric, the expected sum is zero, and hence the quantity might be of order smaller than d? Note that this is an example problem; there are multiple problems with the proof of Lemma 1.4. For example, the paper claims this for all k; but if k is something like d^100, would things hold? What would stop the magnitude of the weights to grow with time?

- Point 6 of the rebuttal: The CLT, when applied w.r.t. the randomness of the weights, says that for a fixed $x$, $\sum_{r=1}^d \hat{a}_r \phi (\hat{w}_r x) / \sqrt{d} \sim N(0,v_x)$ approximately. That is, there is a non-zero probability (w.r.t. the randomness of weights) that the claim in the paper for a fixed $x$ fails. As such, to reason the claim for many $x$, one requires doing probabilistic arguments very carefully.

The paper should execute the proof very carefully. It is not just a matter of technicality; I suspect some of the claims are actually wrong.

More importantly, Condition 1 alone is insufficient to claim dynamical stability at any time k. What should qualify for dynamical stability is rather the existence of a well-defined limiting dynamics exists (which is argued heuristically in Appendix C), and its proof. In the current writing, it’s unclear how Condition 2 is crucial; while it studies interesting properties, it is very restrictive.

As said in my last review, one thing that has been missing is really whether the insight here differs qualitatively from the known NTK and MF limits. Further looking at the limiting dynamics in Appendix C, one sees that they are qualitatively either NTK or MF. There are possible degeneracies due to scalings and the use of logistic loss, but these do not lead to much deviation from NTK or MF behaviors. If one is to use a squared loss for instance, what one would obtain in Figure 1 is just the line connecting NTK and MF; all other points in the band outside this line are degeneracies due to logistic loss. The behavior on this line, again, is qualitatively either NTK or MF, and this is shown (somewhat implicitly) already by a number of past works.

I would imagine a rigorous derivation of the limiting dynamics for each point in the band revolves around the renormalized dynamics in Appendix C. When translating from the renormalized dynamics to the original one, the extra scaling factors will complicate the proof (for instance, they can blow up Lipschitz constants). Again this has to be done very carefully.

---

> ### Author Response · Authors · 2020-11-13
> **A response to Reviewer 2 - Part 1**
>
> We thank the anonymous reviewer for thoroughly reading the manuscript and for sharing a list of insightful comments.
>
> Since there is a long list of issues raised, we divide our answer into several parts. Here is the first part:
>
> 1. *There are many unnecessary notations, for example, $\tilde{q}_a$ and $\tilde{q}_w$ which are equal to each other in the main analysis.*
> We feel that taking $\tilde q_a = \tilde q_w$ from the very beginning may confuse the reader since it requires an additional explanation for this choice. In fact, the “default” scaling, which is important and which is covered in Appendix D, has $\tilde q_a \neq \tilde q_w$.
>
> 2. *The main figure, Figure 1, is very hard to read, and I cannot see where the curve for MF is in this plot.*
> We agree; if merging curves are the main issue, we can fix it in the revised version by adding some vertical bias to each curve.
>
> 3. *The statements also seem to miss several assumptions e.g. assumptions on the data (will things hold if |x| ~ exp(d)??).*
> We have implicitly assumed that the data distribution does not depend on $d$. We can mention this assumption explicitly in the upcoming revision.
>
> 4. *Some notations are not explained, for example, \sigma^{(0)}(x) that appears in Eq (13).*
> As for $\sigma^{(0)}(x)$, it is the std of an NNGP that corresponds to the network at initialization. Exact formulae can be obtained e.g. from [Lee et al., 2019] or from [Jacot et al., 2018] - there is a reference in the text.
>
> 5. *In the proof of Lemma 1.4 in Appendix B.1, why is it that the collection of the weights \hat{w}_r^{(k), r=1…,d, at any time k>1 allows one to apply the law of large numbers?*
> Referring to the law of large numbers here is indeed a mistake - we thank the reviewer for noticing it. Here we have a sum of identically distributed (possibly, dependent) random variables. The expected sum equals the sum of expectations by the linearity of the expectation. Moreover, the absolute value of the $k$-th moment of the sum does not exceed the $k$-th moment of each term times $d^k$. Hence the sum itself scales as $d$ times each term of the sum.
>
> 6. *To prove Lemma 1.2 in Appendix B.1, the paper only proves the statement for a single fixed x; why does it hold for almost every x?*
> The statement of Lemma 1.2 should be read as follows: $\mathcal{P}_{x \sim \mathcal{D}} \left(q_f^{(0)}(x) = q_\sigma + 1/2\right) = 1$. Hence due to absolute continuity of $\mathcal{D}$ it is enough to prove the statement for any $x$ excluding a set of measure zero. We have proven it for $x\neq 0$.
>
> 7. *The abstract should be more precise: it should mention logistic loss for binary classification and that the initializations are zero-mean.*
> We agree that both conditions are important. We shall mention them explicitly in the next revision.
>
> 8. *The paper does not give sufficient justification of why the proposed criteria (Conditions 1 and 2) should be considered. In particular, Condition 2 only concerns with what happens at initialization; does it guarantee stable dynamics well beyond initialization and in what sense? The paper does not discuss this for most points in the “dynamically stable model evolution” band.*
> Following Proposition 1, all points outside the “dynamical stability band” are not stable according to Condition 1, whereas we cannot guarantee that all points of this band are stable. Condition 2 has nothing to do with the stability property. Its purpose is to divide the band into regions, each corresponding to a separate limit model evolution dynamics.
>
> 9. *In fact, compared to the well-studied NTK and MF scalings, the newly identified scaling sym-default is said to have some divergence behavior (in which its prediction function is always infinite, for infinite d, at any positive training time). If anything, the paper should justify why this divergence behavior does not lead to instability; by looking at Figure 1, I think it suffers from numerical instability.*
> As can be observed from Figure 1, the sym-default limit has diverging logits at any $k > 0$; this does not contradict Condition 1. In order to check whether it indeed conforms Condition 1, we have to measure $\Delta f^{(k)} / f^{(1)}$ since we see that $f^{(k)}$ starts to diverge at $k=1$, and see whether it diverges with $d$ or not. In fact, even for diverging logits, $\nabla_f \ell$ is still finite for $\ell$ being a cross-entropy loss; that is why diverging logits cannot be a source of numerical instability.
>
> References:
> * [Lee et al., 2019] Wide Neural Networks of Any Depth Evolve as Linear Models Under Gradient Descent.
> * [Jacot et al., 2018] Neural Tangent Kernel: Convergence and Generalization in Neural Networks.

---

> ### Author Response · Authors · 2020-11-13
> **A response to Reviewer 2 - Part 2**
>
> We continue addressing a list of questions raised by the reviewer:
>
> 10. *In fact, scalings other than NTK and MF have been studied well beyond initialization in a rigorous fashion. In particular, [E et al., 2019] shows that with a wide range of different scalings, it’s entirely possible to have the same stable behavior: NTK behavior where weights do not move and the model tracks some random-feature models. Given that such a rigorous and thorough study is possible, it is unclear how the proposal of Conditions 1 and 2 leads to novel and significant insights.*
> [E et al., 2019] considered essentially the same model as we did; however, the only hyper-parameter they varied was $\beta$, which corresponds to our $\sigma$. They used the standard parameterization, not the NTK one, and keep learning rates in this parameterization constant (independent on width $m$, which is $d$ in our notation). Keeping learning rates in standard parameterization constant corresponds to taking $\hat\eta_a \propto \sigma^{-2}$ and $\hat\eta_w \propto 1$, or $\tilde q_a = -2q_\sigma$ and $\tilde q_w = 0$. We see that $\tilde q_a \neq \tilde q_w$ unless $q_\sigma = 0$. This means that our work and [E et al., 2019] has only one regime in common: $\tilde q_a = \tilde q_w = q_\sigma = 0$. The class of scalings considered by [E et al., 2019] covers neither the mean-field scaling nor the NTK scaling. The last statement seems strange, however, note that the factor $\beta$ is absent in the definition of the model (one can find it in the initialization), while both [Jacot et al., 2018] and [Du et al., 2018] put the factor $n^{-1/2}$ explicitly in the definition of the model. This is equivalent to a different learning rate scaling. The feature of our Conditions 1 and 2 is that they cover both the NTK and the mean-field scalings, as well as classify other possible scaings.
>
> 11. *Why should we compare the magnitude of the kernel K with the magnitude of the prediction function f, while they should have very different physical interpretation? If one removes this condition from the diagram of Figure 1, the band picture disappears and we are effectively left with a half plane, separated by the “evolving kernels” boundary. To the left of this boundary, it is the NTK behavior as expected in [1]. So unless Condition 2.3 signifies some significant and meaningful change in network behaviors, the diagram would not show new significant insights compared to the literature.*
> First of all, there is no contradiction in taking a ratio of two quantities of different dimension -  we just get a quantity which is not dimensionless as a result. In our case, a model increment is proportional to a kernel. This means that the ratio of a kernel and a logit is proportional to the ratio of a logit increment and the logit itself - these two quantities have the same physical dimension. The purpose of Condition 2.3 is to separate the band into two regions: the one for which the logit increment at the initialization is vanishing compared to the logit itself, and the one for which it is diverging. In the former case the learning process does not start: each step do not alter the model predictions in the limit. Hence only the latter case, as well as the boundary between the two cases, make sense.
>
> 12. *In the regime of small learning rate ($\hat{\eta}^\ast$ goes to 0), what we should obtain is a continuous time evolution with an expectation over data, not a discrete-time one with stochastically drawn data. Evidently Eq (14) as given in the paper completely contradicts all previous published reports on the MF limit. It does not make sense to claim, after Equation (19), that the prediction function diverges from iteration k=1 onwards. Firstly, as said, the analysis has to be done with respect to small learning rate and hence the iteration k has to be determined in relation with the learning rate. Secondly, that claim contradicts Figure 1: the prediction function does not seem to diverge at iteration 1.*
> In our case we have a discrete-time evolution for any $\hat\eta^* > 0$. The iteration $k$ does not depend on the learning rate - as the learning rate goes to zero, the effective time-step of a single iteration goes to zero as well. We note, however, that there is a mistake in equation (16): there should be $\hat\eta^*$ factors at the right-hand side - the correct equation is given in Appendix: see (95). In this case, $\Delta\theta$ goes to zero as $\hat\eta^*$ goes to zero. We do not claim that the prediction function should diverge for the mean-field limit.
>
> References:
> * [E et al., 2019] A Comparative Analysis of Optimization and GeneralizationProperties of Two-layer Neural Network and Random FeatureModels Under Gradient Descent Dynamics.
> * [Du et al., 2018] Gradient Descent Provably Optimizes Over-parameterized Neural Networks.
> * [Jacot et al., 2018] Neural Tangent Kernel: Convergence and Generalization in Neural Networks.

---

> ### Author Response · Authors · 2020-11-13
> **A response to Reviewer 2 - Part 3**
>
> Here is the final part of our response:
>
> 13. *The initialization-corrected mean field (IC-MF) limit lacks justification. Its aim is to correct the MF limit w.r.t. Condition 2.1, but there are very simple alternatives to do this correction. The first way is to add any fixed function, whose magnitude are independent of d in suitable sense, to the MF limit. The second way is to initialize the MF limit with non-zero mean distributions; in fact, it is known that non-zero mean initializations are more typical in MF limit. The paper argues that IC-MF limit can be good proxy for standard networks, giving only one simple experiment shown in Figure 1. There is no proof or mathematical heuristics provided. I think this IC-MF idea does not have anything to do with the theory in the previous sections, nor the binary classification problem with logistic loss. As such, it should be at least further tested on more complex experimental tasks, e.g. CIFAR-10. From a mathematical standpoint, I do not foresee a simple argument to show why it can be a good proxy for standard networks.*
> The way how we have corrected the MF limit corresponds to the first way mentioned above. Another property that we have aimed to achieve is that $d=d^*$ should correspond to a reference network trained with hyper-parameters $\sigma^*$ and $\eta^*$: see eq. (25). We agree that even adding this property will not restrict us to a single choice. We have not considered initializations with non-zero mean since in practice weights are always initialized with zero mean - the need for non-zero initialization is rather a specific feature of the MF limit. Assuming non-zero-mean initialization for the reference network seems unnatural. An experiment aimed to justify (or to refute) the IC-MF idea on CIFAR-10 is possible - we can do it for the upcoming revision.

---

> ### Comment · AnonReviewer2 · 2020-11-20
> **Another question**
>
> I thank the authors for the rebuttal. I have another clarifying question. From Appendix C, one may infer that in the dynamically stable band, all points correspond to either one of the two behaviors: constant and non-constant kernels. The constant kernel behavior differs from NTK essentially in the (possible) collapse of the initial prediction function and the logit. The non-constant kernel behavior also differs from MF in a similar fashion. Is this correct?

---

> > ### Author Response · Authors · 2020-11-23
> > **Clarification for the diagram on Figure 1 and Appendix C**
> >
> > Consider the constant kernel case (Appendix C.1).
> > Following the diagram in Figure 1, the initial prediction function, *as well as the initial tangent kernels*, can collapse or *diverge* or stay finite.
> > In contrast, both the initial prediction function and the tangent kernels are finite for the NTK limit.
> >
> > There is no contradiction with eq. (65-70) describing the evolution of model predictions driven by constant kernels in Appendix C.1.
> > Indeed, since logits $f_d(x)$ are allowed to vanish or diverge with $d$, in order to write evolution equations we have to normalize logits to make the finite: $\tilde f_d(x) = (d/d^*)^{-1-\tilde q -  2 q_\sigma} f_d(x)$, see eq. (55); we normalize the kernels in a similar manner, see eq. (52-53). These normalized logits do not diverge as $d \to \infty$ (that's why we have eq. (68)), for this reason, we are able to write the limit evolution equations for them (for $\tilde f_\infty(x)$): see eq. (65-70).
> >
> > We have similar behavior for the non-constant kernel case (Appendix C.2).
> > Again, following the diagram in Figure 1, the initial prediction function, as well as the initial tangent kernels, can collapse or diverge or stay finite, in contrast to the MF limit, for which kernels stay finite, while the initial logits vanish.
> > And again, we have to introduce the normalized prediction function $\tilde f_d(x)$, which is always finite, in order to write the limit evolution equations (89-93).

---

### Official Review · AnonReviewer4 · 2020-10-27
**Interesting story; but the stability conditions defined in this paper are not justified to be important**

**Rating:** 5
**Confidence:** 3

**Review:**

This paper proposed a general framework to derive different stable limiting behaviors of the dynamics of two-layers neural networks, under different parameterization of the hyper-parameters. For certain choices of hyper-parameters, this recovers the mean-field limit and the NTK limit. This paper also proposed certain properties of the limiting dynamics and showed that using these properties as the classification criteria, there are only a finite number of distinct models in the limit. This paper also proposed a novel initialization-corrected mean-field limit that satisfies all properties.
This paper tells an interesting story. The question is whether the problem solved by the story is important.
The main consideration of this paper is to find regimes of hyper-parameters such that the limiting dynamics are stable in some sense. For this purpose, the authors advocate the IC-MF regime, such that the limiting dynamics are stable with respect to all the conditions that the authors proposed. This story seems to be self-contained on its own. However, I believe that two more important criteria for good training algorithms are optimization and generalization efficiency, which are not discussed by the authors.
The optimization and generalization efficiency are much more important than the stability condition in practice. There could be some regimes of hyper-parameters that do not satisfy the stability condition, but has good optimization and generalization efficiency. The IC-MF regime proposed in this paper, although seems to satisfy additional stability conditions, but intuitively, it seems that its generalization efficiency is not as good as that of MF regime: it added an additional noisy function $f_{ntk, \infty}^{0)}$ to the mean-field prediction function, and this additional noise (intuitively will) hurt generalization.
I believe some of these stability properties should have some connection to optimization and generalization. For example, if some simple stability property is violated, the algorithm cannot generalize well. I feel that the authors should try to build connections of stability properties to optimization and generalization, to justify the importance of condition 1 and condition 2 defined in the paper.
Above all, I feel that this paper is interesting in its own criteria. However, it didn't justify that its criteria are important. So I feel that this paper is on the borderline.

Minor issues:
	1.	Some notations are easy to get readers confused. Eq. (7), $\sigma(d) = \sigma^*(d / d^*)^{q_\sigma}$. Here $\sigma$ is a function of $d$ while $\sigma^*$ is a scaler (not as a function of $d / d^*$). It takes me while to understand this.
	2.	Typos: page 18: in this case $1 + \tilde q + 2 q_\sigma$.
	3.	The notations of this paper looks very complicated, especially the superscripts and subscripts.

---

> ### Author Response · Authors · 2020-11-12
> **A response to Reviewer 4**
>
> We agree that the optimization and generalization efficiency of finite-width nets is more practically important than the stability properties of their infinite-width limits. However, if the stability condition, Condition 1, is violated, then either the model stops learning for large enough width (but, of course, it will learn for any finite $d$), or its relative output change becomes enormous for large width. We see that both pathologies that are ruled out by Condition 1 are optimization issues, hence Condition 1 is a necessary condition for the optimization efficiency of the model in the limit of large width. And of course, the above-mentioned issues rule out the generalization efficiency, hence Condition 1 is necessary for generalization of the limit model as well.
>
> As for Condition 2, there is no obvious connection between the above-mentioned properties and Condition 2. This condition aims a different question: "What model evolution can we get in the limit of large width and how much quantitatively the limit model evolution deviates from the evolution of a finite-width net?" This question is important on its own because each limit model can be used as an idealized proxy-model for realistic nets; this idealized model can in its turn be used for theoretical considerations. For instance, the NTK limit enjoys nice optimization and generalization guarantees, while obtaining similar guarantees for finite-width nets are much more complicated. However, a good proxy model should not only be analytically simple but also it should capture the behavior of finite-width models well. This is the purpose of properties listed in Condition 2. The fact that these properties divide the stability band by regions such that each region corresponds to a distinct unique limit model can be seen as a nice additional justification for considering exactly these properties. To sum up, while Condition 2 does not provide any optimization or generalization guarantees on its own, it points out what proxy-models potentially useful for theoretical analysis can we have, and which of them approximate finite-width nets better than others, and hence are more worse-studying. We advocate for the IC-MF limit model because it captures the behavior of finite-width models better than all other limits. Hence we argue that this particular limit model is the most worth-studying theoretically among other proxy-models previously considered (NTK and mean-field).
>
> The "noise" term we have added when defining the IC-MF limit is esentially a bias term for model predictions, and this bias term is fixed once after initialization. This is not a noise term that is chosen randomly at each optimization step; that is why we do not see why it should hurt generalization.
>
> We also thank the reviewer for pointing on notational issues and typos.

---

### Official Review · AnonReviewer3 · 2020-10-27
**An ambitions, potentially exciting paper, which is unfortunately impossible to follow**

**Rating:** 5
**Confidence:** 4

**Review:**


This paper investigates the dynamics of the training of fully connected neural networks with one hidden layer in the infinite-width limit, for classification. Starting from the observation [Golikov, 2020] that the mean-field limit and the NTK regime are only two special cases of continuous families of scalings, it attempts at identifying important or desirable features that an infinite-width limit should display in order to give good insight into the dynamics of large but finite neural networks.

This is an exciting program, and it looks like the ideas are very good. If / when laid out, they will really bring insight into the training of large neural nets.

Unfortunately, the way the paper is written makes it hard to accept in its present form. While the global structure is quite clear, the wording makes it very hard to extract information. For instance, it is not clear what is mean by "finite": does it mean there is a finite limit (what should be expected in principle), that it is bounded from above (what it seems to mean sometimes) or that it is strictly positive (what it appears to mean sometimes)? Also, there is a classification into 13 cases that are promised, but they are not clearly listed in the main, the fact that we are working with only one hidden layer is not clearly said in the abstract (and the discussion on how to extend to more hidden layers is not convincing). The grammar is somehow problematic (there is in particular a lot of missing "the" articles in front of nouns), and it sometimes makes it hard to follow. Also, it is not very clear what is proven, where assumptions are used (e.g. analyticity of the nonlinearity), the seemingly most interesting regimes are not defined in the main, etc.

I am tempted to think that the authors are very lucid about their understanding of what is happening and that they really have interesting something to convey, but the present version makes it very hard to get a reliable information. A (very significantly) revised version of this paper could, I believe, bring much insight to our understanding of neural nets. There is a lot of potential with this paper, just not realized in terms of exposition.

---

> ### Author Response · Authors · 2020-11-12
> **A response to Reviewer 3**
>
> We thank the anonymous reviewer for providing a useful external view of our manuscript.
>
> We shall try to clarify some points mentioned in the review:
>
> 1. *The meaning of the term "finite".*
> We say that $f(d)$ stays finite as $d\to\infty$ if both $f(d)$ and $1/f(d)$ stay bounded in some vicinity of infinity.
>
> 2. *The classification on 13 classes is not listed in the main.*
> The 13 limit models are listed in Appendix C. In particular, eqs. (65-70) define a class of models with constant limit kernels; there are 8 models in this class. At the same time, eqs. (89-93) define a class of models with non-stationary limit kernels; this class consists of 5 distinct limit models. We haven't listed all of these models explicitly in the main in order to save space, but we have listed the most interesting of them (non-dominated ones) in Section 3. However, we can add an explicit reference to them to the main of the revised version of the paper.
>
> 3. *The fact that we are working with only one hidden layer is not clearly said in the abstract.*
> We shall modify the abstract for the revised version accordingly.
>
> 4. *The discussion on how to extend to more hidden layers is not convincing.*
> The purpose of the discussion was not to convince the reader that our approach can be easily generalized to deep nets, but rather to give a sketch of a plan for actually doing it. We feel that the same program applied to deep nets can, in fact, be conceptually harder (at least, because the mean-field limit for deep nets is conceptually more involved as compared to shallow nets).
>
> 5. *The grammar is somehow problematic.*
> We apologize for the grammar issues. We shall do our best for fixing them.
>
> 6. *It is not very clear what is proven, where assumptions are used (e.g. analyticity of the nonlinearity).*
> Analyticity of the non-linearity is necessary to ensure that $\phi(\mathbf{\hat w^{(k)}} \mathbf{x})$ is not zero a.s. wrt $x \sim \mathcal{D}$ and the initialization. This is needed, for instance, to derive the exact power-law for exponent for $\Delta a^{(k)}$: see Lemma 1.3 in Section B.1.
>
> 7. *The seemingly most interesting regimes are not defined in the main.*
> We argue that the most interesting regimes are "non-dominated" ones: brief definitions for all of them are given in Section 3. The complete definitions are given in Appendix C. In particular, the NTK limit is defined in Section C.1.1, eqs. (71-75), the MF limit is defined in Section C.2.1, eqs. (94-96), the sym-default limit is defined in Section C.2.2, eqs. (97-101), and the IC-MF limit is defined in Section E, eqs. (131-134). We shall add explicit references to all of them to the main in the upcoming revision.

---

### Official Review · AnonReviewer1 · 2020-10-28
**Official Blind Review #1**

**Rating:** 7
**Confidence:** 3

**Review:**

Summary:

Main topic of the paper is to study various infinite width limits. The paper notices different scaling used in NTK limit and MF limit and proposes a general framework for studying the limit behavior depending on the scaling. This allows the authors to define more general dynamically stable models in the infinite width limit.

Stated contribution by authors are:

- Framework for reasoning about scaling that leads to ‘dynamically stable’ model evolution in the infinite width limit.
- Categorization of 13 distinct dynamically stable models in the infinite width limit.
- Characterization of “sym-default” model, which along with NTK and MF limit which shows properties most of finite-width network evolution.
- Model modification “Initialization-corrected mean-field limit”(IC-MF) that satisfy all identified property of finite-width network evolution
- Demonstration of IC-MF limit approximating finite-width network the best among all other possible models.

Reason for score:

The paper proposes an interesting theoretical framework to capture various infinite width limits. While settings are limited to make theory tractable, there is some empirical validation as well as capacity to broaden the study to different infinite width limits. I believe there are interesting novelties to be shared among ICLR participants who are interested in deep learning theory.

Pros:
General framework encapsulates widely studied infinite width limits and generalizes them which would be useful for theoretical study of large neural networks.

The paper makes predictions on scaling limits and finds a limit that could agree with finite networks better (IC-MF). For a variety of experiments the fact that this class matches with a reference finite width network is demonstrated well.

While the proposed analysis is limited to a single hidden layer case, the authors have described thoughtful possibilities how the framework could potentially generalize to deeper networks.
Authors shared code to reproduce their experiments in the paper which is useful for reproduction and clearly understand different proposed scalings.

Cons:

Being single hidden layer analysis is a drawback (while discussion on extension is appreciated and stated in the “pros”). Often multi-layer infinite width is more interesting since the number of hidden layer weights scales quadratically while only the input and output layer weight count scales linearly with width. This can induce different training dynamics and scaling, so limitation to single hidden layer analysis is a drawback.

To simplify proof non-linearity of the choice is “leaky softplus” which is not widely used in practice. Having general comments on extension or limitation to well-known activations is advised.

Assumption in p3 stating “gradients for a and W are estimated using independent data samples” may be too strong and not realistic. I cannot find no validation that the assumption is a reasonable one to make beyond making the theory simple. This may lead to inconsistencies raised in the “Questions” below.

Experiments demonstrating similar effect for more realistic setting (e.g. full CIFAR-10 classification task achieving reasonable performance) would have been better to convince the validity or generalizability of the theory. While one can see that IC-MF and reference agrees well for the training curve, for test-set since every curve essentially behaves the same it is hard to determine which limit is superior or not.

Questions:
In [Liu et al., 2020], they show non-constancy of NTK for non-linear output including the soft-max layer in the outputs. I believe this fact may be in conflict with this paper’s analysis on binary cross-entropy loss and still having constant NTK. I do suspect it may be due to assumptions on independent gradients for `a` and `W` in page 3. Do you think in a realistic training setting with softmax-cross entropy, you can generalize the analysis and reconcile the fact that NTK is non-constant?

In general, does the generalization of “dynamical stability” to admit infinite logits valid for multi-class cases? In multi-class simple “sign” can’t be used for classification and I wonder if authors have ideas on generalization to multi-class settings.

In eq (26), why is Gaussian distribution over logit a good way to measure KL divergence? I understand at initialization prior is distributed over Gaussian, however after training with softmax I expect the logit distribution is no-longer Gaussian.

Nits and additional feedback:

Nit: use \citep in places where appropriate.

Division (/) for denoting “or” is often confusing and would suggest other notation. Especially in the Figures and Condition 2 when sometimes it means “or” and sometimes it means division.

p4: ‘grows width’ -> ‘grows with’

[Sohl-Dickstein et al., 2020] show different scaling of weight and bias scaling extending standard parameterization to work well with NTK limit. While this paper mostly works with the NTK parameterization, it would be interesting to discuss how the improved parameterization in [Sohl-Dickstein et al., 2020] can be utilized. I suspect this is quite related to (9) / (10) where definition of NTK deviates from [Jacot et al., 2018].

In Section 3: When comparing performance on finite networks and NTK, it can be subtle depending on how one trains finite networks and there’s architecture dependence [Lee et al., 2020]
In Section 4: suggest citing [Chizat et al., 2019] for ‘lazy training’

Chizat et al., On Lazy Training in Differentiable Programming, NeurIPS 2019
Lee et al., Finite Versus Infinite Neural Networks: an Empirical Study, arXiv: 2007.15801
Liu et al., On the linearity of large non-linear models: when and why the tangent kernel is constant, arXiv:2010.01092
Sohl-Dickstein et al., On the infinite width limit of neural networks with a standard parameterization, arXiv:2001.07301

----
[post rebuttal]
I thank the authors for their revision and clarifying many of my questions and adding results based on that. I have read other reviewers concern and while I agree some room for improvement on clarity, I still think the paper brings in value. Unless there's technical flaws spotted by other reviewers that has not been resolved, I'm still leaning towards accepting (increased score from 6 to 7).

---

> ### Author Response · Authors · 2020-11-12
> **Authors' response for the official review #1**
>
> First of all, we thank the anonymous reviewer for thoroughly reading the manuscript and sharing a list of useful comments.
>
> We now respond to specific issues and questions raised by the reviewer:
>
> 1. *Being single hidden layer analysis is a drawback.*
> We absolutely agree that the multi-layer case is more realistic and is likely to enjoy a richer class of phenomena. However, we still feel our current analysis is a necessary starting point for extending it to a multi-layer case. We believe that at least, the shallow case can be a useful analogy for a multi-layered one. As an illustration of our arguments, we can mention the two papers on GD convergence via the kernel stability argument: [Du et al., 2018a] and [Du et al., 2018b]. The first deals with the shallow case and provides a good illustration of the approach, while the second deals with a multi-layer case, which is more realistic but more technically involved.
>
> 2. *“Leaky softplus” is not widely used in practice.*
> Our proof technique requires two properties: smoothness and asymptotic linearity. Relaxing any of it will likely introduce a lot of technicalities. We note, however, that our experiments were held for standard leaky ReLU nets and the results do not contradict our theoretical findings.
>
> 3. *Assuming that the gradients for a and W are estimated with independent data samples is not realistic.*
> Assuming that, we still have an unbiased estimate for true gradients, so, we feel that our modification of the standard SGD lies on the same level of approximation as the standard SGD itself. Note that in our experiments we have used either a full-batch GD, or the standard mini-batch SGD.
>
> 4. *Does the generalization of “dynamical stability” to admit infinite logits valid for multi-class cases?*
> Yes, in this case what matters is the argmax of logits, which is invariant to scaling of logits - they can be vanishing or diverging.
>
> 5. *Need an experiment for CIFAR10 with reasonable peformance.*
> Taking the previous comment into account, we do not see any conceptual obstacles in providing the same set of experiments in a multi-class setting. We can add them into the next revision.
>
> 6. *In [Liu et al., 2020], they show non-constancy of NTK for non-linear output including the soft-max layer in the outputs.*
> One of the results of this paper is that having an activation function at the end of a network makes its NTK non-constant. For cross-entropy loss, we can either view a softmax layer at the end of the network as its part, or view it as a part of the loss function. These cases differ in their definition of the NTK. In the former case the NTK is indeed non-constant, however, we have used the latter point of view in our paper - in this case the NTK is constant, and there is no contradiction with [Liu et al., 2020]. This phenomenon is not related to our assumptions of independent gradients wrt $a$ and $W$.
>
> 7. *Why is Gaussian distribution over logit a good way to measure KL divergence?*
> We cannot really guarantee that the logits are Gaussian for $k > 0$. However, we do not know their distribution, and cannot measure the KL-divergence solely using samples (at least, robustly). Assuming normality seemed the simplest way to approximate the divergence. We have also tried to apply sigmoid to our logits, assume that the resulting probabilities follow beta-distribution, and measure the KL between the two betas. The results were qualitatively similar -  we are ready to add them to Appendix in the next revision of the paper. We understand, however, that it can be a good idea to consider the Wasserstein distance between samples instead of the KL-divergence.
>
> 8. *“/” for “or” is confusing.*
> We are thinking about three alternatives: (1) just "or" (clear, but may be too long), (2) "v" (may be confusing), (3) "|" (short, but can be misinterpreted as conditioning). What do you think can be a good substitute?
>
> 9. *[Sohl-Dickstein et al., 2020] extend the NTK limit to a standard parameterization.*
> The improved standard parameterization of [Sohl-Dickstein et al., 2020] introduces a width-scaling factor $s$, and ensures that for $s=1$ it recovers a “baseline” network, while the NTK stays finite when $s$ goes to infinity. Essentially we apply the same way of scaling: see Appendix F. In our terminology a baseline network is called a “reference” and $s = d/d^*$. Indeed, for $d=d^*$ we recover the reference network with hyper-paramaters $\sigma^*$, $\eta^*$ (in standard parameterization), and for appropriate scaling exponents (e.g. for the NTK scaling $q_\sigma = -1/2$, $\tilde q = 0$), the kernels stay finite as $d$ (or $s$) goes to infinity. Moreover, the same thing was done previously in [Golikov, 2020].
>
> 10. *Cite [Chizat et al., 2019] for “lazy training”.*
> Sure, will add the citation.
>
> References:
> * [Du et al., 2018a] Gradient Descent Provably Optimizes Over-parameterized Neural Networks
> * [Du et al., 2018b] Gradient Descent Finds Global Minima of Deep Neural Networks

---

### Author Response · Authors · 2020-11-21
**A summary of the revision by 21 Nov**

In the current revision, we attempted to cover the issues noted by the reviewers. We are ready to add more revisions if necessary.

Here is the summary of changes:

1. We have explicitly mentioned in the main that while we assumed "leaky Softplus" activation to simplify our theoretical constructions, all experiments were held for standard leaky ReLUs.

2. Similarly, while we have assumed a non-traditional SGD variant in the theory, all experiments were held either for a full-batch GD, or for a standard mini-batch SGD --- we have mentioned this explicitly in the main.

3. We have added a citation of [Chizat et al., 2019] for "lazy training" in Section 4.

4. We have introduced \citep were appropriate.

5. We have replaced "a / w" meaning "a or w" with "a V w".

6. We have added a reference to a complete system of evolution equations for every limit model we discuss in Section 3.

7. We have explicitly stated that it is a single hidden layer analysis early in the abstract.

8. We have also mentioned in the abstract that we consider binary classification with zero-mean initialization.

9. We have added small vertical displacements to plots in the right panel of Figure 1 in order to make curves visually distinguishable.

10. We have fixed a confusing issue in eq. (7).

11. We have explicitly stated that the data distribution does not depend on width $d$.

12. In addition to measuring the KL-divergence between Gaussian fits for logits as in Figure 2, we have measured a KL-divergence between beta distribution fits for probabilities: see Appendix I.

13. We have also conducted experiments on CIFAR10: see Figure 8. In short, the IC-MF limit model still fits a reference best among other limit models in terms of the KL-divergence between logits, however, none of the limit models considered track the test accuracy well enough.

---

### Decision · Program_Chairs · 2021-01-07
**Final Decision**

**Decision:**

Reject

**Comment:**

This paper proposes a general framework to study the limit behavior of neural models with respect to the scaling of hyperparameters in terms of network width, which covers existing mean-field (MF) and neural tangent kernel (NTK) limits, as well as other new limit models that were not discovered before. While the reviewers agree that the study of limiting behavior of neural network models is of great importance and could be a good addition to the current understanding of NTK and MF, there is a technical flaw in the proof (regarding Condition 1) pointed by the reviewer. After reviewer discussion, all reviewers agree that this is a serious issue, and needs to be addressed before publication. I believe it could be a strong paper if the technical flaw can be fixed. I encourage the authors to revise the paper and resubmit it to the next conference.